# IDLM: Inverse-distilled Diffusion Language Models

David Li [*1]   Nikita Gushchin [*2 3]   Dmitry Abulkhanov   Eric Moulines [1 4]   Ivan Oseledets [3 2]   Maxim Panov [1]
Alexander Korotin [2 3]

## Abstract

Diffusion Language Models (DLMs) have recently achieved strong results in text generation. However, their multi-step sampling leads to slow inference, limiting practical use. To address this, we extend Inverse Distillation, a technique originally developed to accelerate continuous diffusion models, to the discrete setting. Nonetheless, this extension introduces both theoretical and practical challenges. From a theoretical perspective, the inverse distillation objective lacks uniqueness guarantees, which may lead to suboptimal solutions. From a practical standpoint, backpropagation in the discrete space is non-trivial and often unstable. To overcome these challenges, we first provide a theoretical result demonstrating that our inverse formulation admits a unique solution, thereby ensuring valid optimization. We then introduce gradient-stable relaxations to support effective training. As a result, experiments on multiple DLMs show that our method, *Inverse-distilled Diffusion Language Models (IDLM)*, reduces the number of inference steps by $4\times$-$64\times$, while preserving the teacher model's generation quality. We provide the code, model checkpoints, and video tutorials on the project page:

https://david-cripto.com/idlm

## 1. Introduction

Generative modeling for discrete data, such as natural language, has seen widespread adoption, driven by the rapid advancement of large language models (Touvron et al., 2023; Groeneveld et al., 2024; Lozhkov et al., 2024). Recently, Diffusion Language Models (DLMs; Sohl-Dickstein et al., 2015; Austin et al., 2021; Campbell et al., 2022; Lou et al.,

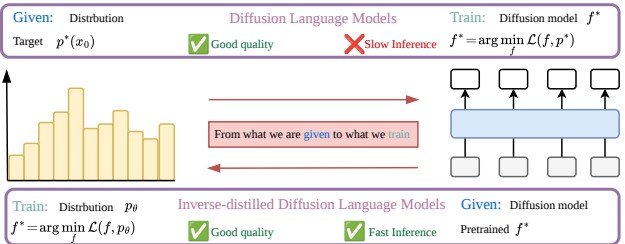

*Figure 1.* **Diffusion training vs inverse distillation.** DLMs learn a slow teacher $f^*$ from data. IDLM fixes $f^*$ and learns $p_\theta$ for fast few-step generation.

2024) have emerged as a promising framework for modeling the distribution of the text data. A general formulation for a broad class of DLMs was introduced by (Lou et al., 2024). However, the general formulation is often intractable in practice and introduces additional theoretical complexity. To address this, subsequent work (Sahoo et al., 2024; 2025; Schiff et al., 2025; Shi et al., 2024) has focused on specific types of diffusion processes, enabling simplified theoretical analyses and more tractable implementations. As a result, DLMs have achieved competitive performance in text generation, comparable to that of autoregressive language models (Nie et al., 2025). However, the inherently iterative nature of reverse diffusion requires hundreds or even thousands of sampling steps at inference time, resulting in considerable latency that limits their practicality.

On the other hand, in the continuous domain, the issue of slow inference in diffusion models (Ho et al., 2020; Song et al., 2021b; Karras et al., 2022) has been the focus of extensive research (Song et al., 2021a; Luhman & Luhman, 2021; Zhang & Chen, 2023; Lu et al., 2022). Among the various approaches, *distillation* has emerged as one of the most effective solutions (Song et al., 2023; Salimans & Ho, 2022; Luhman & Luhman, 2021; Berthelot et al., 2023; Xie et al., 2024). However, directly applying such techniques in the discrete domain is challenging due to the non-differentiable nature of discrete data. Recent efforts have begun to address this limitation for DLMs. Notably, SDTT (Deschenaux & Gulcehre, 2025) and Duo-DCD (Sahoo et al., 2025) propose adaptations of consistency-based distillation methods (Song et al., 2023; Song & Dhariwal, 2024; Kim et al., 2024) to the discrete setting, demonstrating initial progress toward enabling fast inference in DLMs.

[*]Equal contribution  [1]Mohamed Bin Zayed University of AI, Abu Dhabi, UAE [2]Applied AI Institute, Moscow, Russia [3]AXXX, Russia [4]EPITA, Laboratoire Recherche de l'EPITA, Paris, France. Correspondence to: David Li <David.Li@mbzuai.ac.ae>, Alexander Korotin <iamalexkorotin@gmail.com>.

*Proceedings of the 43rd International Conference on Machine Learning*, Seoul, South Korea. PMLR 306, 2026. Copyright 2026 by the author(s).

In this work, we investigate an alternative distillation paradigm by extending the theoretical framework of *Inverse Distillation* (Gushchin et al., 2025; Kornilov et al., 2026; Selikhanovych et al., 2026), originally developed for continuous diffusion models, to the discrete domain. However, this direct extension introduces several nontrivial challenges arising from the intrinsic properties of discrete data. First, theoretical concerns emerge regarding the validity of the training objective, particularly due to the potential non-uniqueness of its minimizers. Second, practical challenges arise from the non-smoothness of the discrete space, which complicates the optimization.

**Contributions.** To address the above-mentioned issues, we present *Inverse-distilled Diffusion Language Models (IDLM)*, a principled framework for accelerating a broad class of discrete diffusion models. In summary, our core contributions are as follows:

1. We theoretically establish the uniqueness of the global optimum for the IDLM objective, validating the correctness of the proposed optimization procedure (§3.1).
2. We propose gradient-stable relaxations for effective training in the non-smooth discrete domain (§3.2).
3. We empirically demonstrate that IDLM achieves competitive generation quality, matching the performance of 1024-step teacher models while requiring up to $64\times$ fewer sampling steps (§5).

## 2. Preliminaries

**Notations.** We represent scalar discrete random variables over $N$ categories as one-hot column vectors, with $\mathcal{V} = \{x \in \{0,1\}^N \colon \sum_{i=1}^N x_i = 1\}$ denoting the set of such vectors. We further define the probability simplex $\Delta := \{z \in \mathbb{R}^N \colon z \geq 0, \langle z, \vec{1}\rangle = 1\}$ as the convex hull of $\mathcal{V}$, where $\vec{1} \in \mathbb{R}^N$ denotes the all-ones vector.

### 2.1. Diffusion Language Models

Diffusion Language Models (DLMs; Austin et al., 2021; Campbell et al., 2022; Lou et al., 2024) generate samples from a data distribution $p^*(x_0)$ by learning to reverse a prescribed forward noising process. In this work, we focus on two tractable process families that are widely used in language modeling: the *absorbing* process, which underlies MDLM, and the *uniform* process, which underlies UDLM and Duo. We defer the general CTMC formulation and its SEDD score-entropy instantiation to Appendix A. The process-specific details for masked diffusion and uniform diffusion are expanded in Appendices B and C, respectively.

#### 2.1.1. ABSORBING PROCESS

**Forward process.** The absorbing process corrupts tokens by replacing them with the mask token $m$. Its terminal distribution is concentrated on $m$, and under the usual schedule $\alpha_t \in [0, 1]$, the forward transition from a clean token $x_0$ is

$$q_t(x_t \mid x_0) = \mathrm{Cat}\big(x_t; \alpha_t x_0 + (1 - \alpha_t)m\big).$$

Thus, a token is either still equal to $x_0$ or has already become masked.

**Training loss.** MDLM (Sahoo et al., 2024; Shi et al., 2024) parameterizes the denoiser as $f \colon \mathcal{V} \times [0, 1] \to \Delta$ and trains it with the weighted masked prediction objective

$$\mathcal{L}_{\mathrm{MDLM}}(f, p^*) := -\mathbb{E}_{p^*(x_0), t, q_t}\big[\lambda_t \langle x_0, \log f(x_t, t)\rangle\big]. \quad (1)$$

where $\lambda_t$ is a positive weighting function and the logarithm is applied element-wise.

**Sampling procedure.** The SUBS parameterization of MDLM encodes two useful properties: unmasked tokens are copied forward, and the model assigns zero probability to predicting the mask token as clean data. For $0 \leq s < t \leq 1$, the learned reverse transition is the absorbing posterior with the clean token replaced by the model prediction, define $f_t := f(x_t, t)$:

$$p_{s|t}^f(x_s \mid x_t) = \begin{cases} \mathrm{Cat}(x_s; x_t), & x_t \neq m, \\ \mathrm{Cat}\left(x_s; \dfrac{(1 - \alpha_s)m + (\alpha_s - \alpha_t)f_t}{1 - \alpha_t}\right), & x_t = m. \end{cases}$$

Sampling starts from a fully masked sequence $x_1 \sim \delta_m(\cdot)$ and repeatedly draws $x_s \sim p_{s|t}^f(\cdot \mid x_t)$ along a decreasing time grid. Once a token is revealed, it is carried over in later denoising steps. The masked-diffusion sampling process is visualized in Figure 8. The full MDLM formulation is given in Appendix B.1.

#### 2.1.2. UNIFORM PROCESS

**Forward process.** The uniform process corrupts tokens toward the uniform distribution $\frac{1}{N}\vec{1}$. With the same notation $\alpha_t$, its forward transition can be written compactly as

$$q_t(x_t \mid x_0) = \mathrm{Cat}\big(x_t; \alpha_t x_0 + (1 - \alpha_t)\tfrac{1}{N}\vec{1}\big).$$

Unlike absorbing diffusion, an intermediate token can be changed again at later reverse steps. This makes the process self-correcting: generation must decide both *which* token to move to and *when* to move.

**Training loss.** UDLM (Schiff et al., 2025) uses the clean-token predictor as MDLM. Duo (Sahoo et al., 2025) keeps the same uniform discrete process but evaluates the model on a Gaussian-relaxed input, which gives smoother gradients during training while recovering the hard uniform process in the low-temperature limit, which makes the following

parameterization of the model $f\colon \Delta \times [0,1] \to \Delta$. We write the objective as

$$\mathcal{L}_{\text{Duo}}(f, p^*) := \mathbb{E}_{p^*(x_0), t, q_t}\big[g_{\text{Duo}}\big(x_t, x_0, f(x_t, t)\big)\big]. \quad (2)$$

All details of $g_{\text{Duo}}$ are deferred to Appendix C.

**Sampling procedure.** Sampling starts from a sequence of tokens sampled independently from the uniform distribution, $x_1 \sim \text{Cat}(\vec{1}/N)$. Then the sequence is iteratively refined by the learned uniform reverse posterior. For $0 \leq s < t \leq 1$, define $\alpha_{t|s} := \alpha_t/\alpha_s$ and $f_t := f(x_t, t)$. The ancestral sampler uses

$$p^f_{s|t}(x_s \mid x_t) = \text{Cat}\Big(x_s; \frac{N\alpha_t\, x_t \odot f_t + (\alpha_{t|s} - \alpha_t)x_t}{N\alpha_t\langle x_t, f_t\rangle + 1 - \alpha_t}$$
$$+ \frac{(\alpha_s - \alpha_t)f_t + (1 - \alpha_{t|s})(1 - \alpha_s)\vec{1}/N}{N\alpha_t\langle x_t, f_t\rangle + 1 - \alpha_t}\Big).$$

In the sampling procedure each reverse step can revise the whole sequence. The uniform-diffusion sampling process is visualized in Figure 12. The full UDLM/Duo formulation is given in Appendix C.

The Duo paper also introduces an additional greedy sampling procedure. In the experiments, we report both variants and denote ancestral sampling by superscript ([a]) and greedy sampling by ([g]).

### 2.1.3. UNIFIED NOTATIONS & SEQUENCE LEVEL LOSS

**Unified token-level objective.** SEDD (12), MDLM (1), and Duo (2) use different losses, but they share the same structure. We write $\mathcal{L} \in \{\mathcal{L}_{\text{SEDD}}, \mathcal{L}_{\text{MDLM}}, \mathcal{L}_{\text{Duo}}\}$ in order to generalize our distillation scheme. Each loss can be expressed by the integrand $g$ of respective loss, so for any distribution $p$ we define:

$$\mathcal{L}(f, p) = \mathbb{E}_{p(x_0), t, q_t}\big[g\big(x_t, x_0, f(x_t, t)\big)\big].$$

**Sequence-level objective.** In practical applications, we are interested in generating sequences $x_0^{1:L}$ of length $L$. To avoid an exponentially large state space, DLMs usually apply the forward corruption independently across sequence positions (Lou et al., 2024). Under this assumption, the forward distribution factorizes as

$$q_t(x_t^{1:L} \mid x_0^{1:L}) = \prod_{l=1}^{L} q_t(x_t^l \mid x_0^l).$$

The corresponding sequence-level diffusion loss becomes

$$\mathcal{L}^{1:L}(f, p^*) = \mathbb{E}_{p^*(x_0), t, q_t}\Big[\sum_{l=1}^{L} g\big(x_t^l, x_0^l, f^l(x_t^{1:L}, t)\big)\Big].$$

## 2.2. Inverse Distillation

**Main goal.** The goal of inverse distillation is to directly train a student distribution $p_\theta(x_0)$, represented by a fast generator, so that it approximates the true data distribution $p^*(x_0)$. We use the pretrained diffusion model $f^*$, which was trained on samples from $p^*$. The good $p_\theta$ should be one for which the same diffusion training procedure would recover $f^*$.

**Diffusion training.** In usual diffusion training problem: given samples from $p^*$, we train the diffusion model by solving

$$f^* = \arg\min_f \mathcal{L}(f, p^*). \quad (3)$$

Thus, the pretrained model $f^*$ is the diffusion model whose predictions are optimal for the real data distribution, we call this model the *teacher*.

**Inverse viewpoint.** Inverse distillation reverses the roles of the known and unknown objects. We are given the pretrained diffusion model $f^*$ and seek a student distribution $p_\theta(x_0)$, such that training the same diffusion objective on samples from $p_\theta$ would recover the teacher:

$$f^* = \arg\min_f \mathcal{L}(f, p_\theta). \quad (4)$$

In words, diffusion training asks which diffusion model fits a distribution, while inverse distillation asks which distribution would make the fixed pretrained diffusion model optimal.

**Inverse distillation loss.** The optimality condition in (4) is not directly feasible as a training objective, since it requires optimizing parameters $\theta$ through an $\arg\min$. Following prior work on inverse distillation (Gushchin et al., 2025; Kornilov et al., 2026; Selikhanovych et al., 2026), we use tractable inverse loss

$$\mathcal{L}_{\text{inv}}(\theta) := \mathcal{L}(f^*, p_\theta) - \min_f \mathcal{L}(f, p_\theta). \quad (5)$$

The inner minimization defines a *fake* diffusion model trained on samples from $p_\theta$. The term $\mathcal{L}(f^*, p_\theta)$ measures the teacher denoiser's performance on samples from $p_\theta$, while $\min_f \mathcal{L}(f, p_\theta)$ is the best achievable denoising loss for a diffusion model trained on $p_\theta$. The generator is therefore optimized by minimizing this gap. By construction, $\mathcal{L}_{\text{inv}}(\theta) \geq 0$, and if $p_\theta = p^*$ then $\mathcal{L}_{\text{inv}}(\theta) = 0$. The converse is not guaranteed for this general objective.

**Connection to this work.** In *continuous* space, inverse distillation was extensively explored by prior works (Gushchin et al., 2025; Kornilov et al., 2026; Selikhanovych et al., 2026). Our main goal is to extend this idea to the *discrete* domain. However, it presents several challenges, stemming from the inherent characteristics of discrete spaces. These include theoretical concerns regarding the validity of the training objective, due to the non-uniqueness of its minimizers, as well as practical difficulties related to the non-differentiability of discrete sampling.

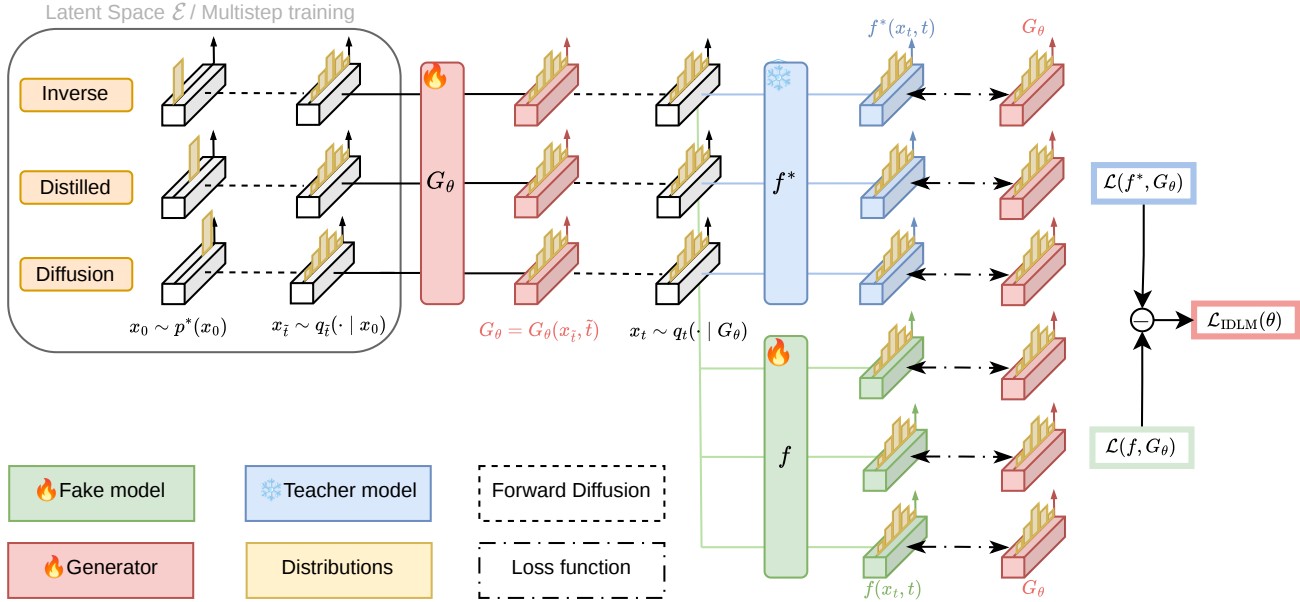

*Figure 2.* **IDLM training loop.** The student generator $G_\theta$ maps noised data to a clean sample, which is forward-diffused and passed as input to the frozen teacher $f^*$ and the learnable fake model $f$. Training alternates between fitting the fake model to the current student distribution and updating the generator with the IDLM loss. This loss pushes $G_\theta$ toward samples supported by the teacher more than by the fake model.

## 3. Inverse-distilled Diffusion Language Models

In this section, we present our core methodology for distilling a broad class of Diffusion Language Models into a few-step generator. We begin by providing the theoretical foundations supporting the extension of the Inverse Distillation framework to the discrete domain (§3.1). We then address the practical challenges arising from the non-smooth nature of discrete spaces (§3.2). Lastly, we outline key implementation details and introduce the sequence-level scaling of the proposed objective (§3.4). Appendices B and C give the masked- and uniform-diffusion formulations used below.

### 3.1. Theoretical extension of Inverse Distillation

**Inverse distillation for DLMs.** To address the issue of slow inference, we extend the Inverse Distillation framework, originally developed for continuous diffusion models, to the domain of Diffusion Language Models. Then, following the logic of the continuous case, described in (§2.2), we propose to consider the following objective for the discrete diffusion:

$$\mathcal{L}_{\text{IDLM}}(\theta) = \mathcal{L}(f^*, p_\theta) - \min_f \mathcal{L}(f, p_\theta). \quad (6)$$

**Theoretical validity of the training objective.** However, simply formulating this objective in a manner analogous to the distillation loss used in continuous diffusion models does not guarantee the validity of its solutions. The absence of a uniqueness guarantee implies that the optimization procedure may converge to suboptimal solutions. To address this, we present the following theorem.

**Theorem 3.1** (Unique solution). *For the SEDD* (12)*, MDLM* (1)*, and Duo* (2) *(in the limit as $\tau \to 0^+$) objectives the IDLM loss defined in equation* (6) *satisfies*

$$\mathcal{L}_{\text{IDLM}}(\theta) \geq \mathcal{D}_{KL}(p_\theta \parallel p^*) \geq 0$$

*and achieves its minimum (zero) if and only if the model distribution matches the target distribution*

$$\mathcal{L}_{\text{IDLM}}(\theta) = 0 \iff p_\theta = p^*.$$

We give the proof and discuss the sequence-level case in Appendix D.2.

**Probabilistic intuition.** The IDLM loss has a simple probabilistic interpretation: it matches full diffusion trajectories. If $\mathbb{P}^\theta$ and $\mathbb{P}^*$ denote the trajectory distributions induced by $p_\theta$ and $p^*$, respectively, then the proof shows

$$\mathcal{L}_{\text{IDLM}}(\theta) = \mathcal{D}_{\text{KL}}(\mathbb{P}^\theta \parallel \mathbb{P}^*).$$

Because the final clean sample is part of the trajectory, this KL controls $\mathcal{D}_{\text{KL}}(p_\theta \parallel p^*)$, which is exactly the inequality in Theorem 3.1. Hence zero IDLM loss can occur only when the student recovers the data distribution.

### 3.2. Addressing Discrete Nonsmoothness

We now explain the main optimization problems that arise when applying IDLM in a discrete domain. For clarity, we use MDLM as an example because the absorbing process gives the simplest setting in which these bottlenecks can be seen explicitly. The same problems also arise for uniform diffusion. Appendix B.2 expands the MDLM-specific inverse distillation simplification used below. Appendix C.2

gives the corresponding uniform-diffusion specialization. Specializing the IDLM objective to the MDLM loss gives

$$\mathcal{L}_{\text{IDLM}-\text{MDLM}}(\theta) = \mathcal{L}_{\text{MDLM}}(f^*, p_\theta) - \min_f \mathcal{L}_{\text{MDLM}}(f, p_\theta).$$

Here we use the same MDLM loss as in the Equation (1), replacing the data distribution $p^*$ by the student distribution $p_\theta$. This is the quantity through which the distribution $p_\theta$ must be optimized:

$$\mathcal{L}_{\text{MDLM}}(f, p_\theta) := -\mathbb{E}_{p_\theta(x_0), t, q_t}\big[\lambda_t\langle x_0, \log f(x_t, t)\rangle\big]. \quad (7)$$

In practice, we parametrize ditribution $p_\theta$ by *student* model $G_\theta \colon \mathcal{E} \to \mathcal{V}$, which transforms latent variable $\epsilon \in \mathcal{E}$ to a discrete output $x_0 \in \mathcal{V}$. The latent space $\mathcal{E}$ is equipped with a tractable prior distribution $p_\mathcal{E}$, inducing a distribution $p_\theta$ over $\mathcal{V}$ via the mapping $G_\theta$:

$$\epsilon \sim p_\mathcal{E}, \qquad x_0 = G_\theta(\epsilon).$$

In the equations below, we highlight with red quantities whose values depend on the generator parameter $\theta$. Substituting the generator sample into (7) gives the loss:

$$\mathcal{L}_{\text{MDLM}}(f, p_\theta) = -\mathbb{E}_{\epsilon, t, x_t \sim q_t(\cdot|G_\theta(\epsilon))}\big[\lambda_t\langle G_\theta(\epsilon), \log f(x_t, t)\rangle\big].$$

This formulation reveals two gradient bottlenecks. First, if $G_\theta$ is forced to output a one-hot token in $\mathcal{V}$, then the generator output is not a smooth function of its logits. Second, $x_t$ is sampled from $q_t(\cdot \mid G_\theta(\epsilon))$, hence the noised token depends on $\theta$ through the probabilities of this categorical distribution. Backpropagating through the objective would therefore require passing gradients through the sampling of $x_t$. This is not handled by automatic differentiation, and common alternatives such as hard Gumbel-Softmax can be unstable.

**Simplex relaxation.** To remove the first bottleneck, we relax the generator range from $\mathcal{V}$ to the simplex $\Delta$:

$$G_\theta(\epsilon) \in \Delta.$$

This relaxation is compatible with both ingredients of the MDLM loss: the cross-entropy term remains a valid soft-label loss, and the absorbing conditional distribution $q_t(\cdot \mid G_\theta(\epsilon))$ remains a categorical distribution when $G_\theta(\epsilon) \in \Delta$. Figures 7, 9, and 10 in Appendix B.2 visualize why this remains a valid update and how it changes a one-hot token loss into a weighted token loss. Appendix C.2 uses the same relaxation for uniform diffusion.

**MDLM: SUBS parameterization.** For MDLM, the SUBS parameterization copies already unmasked tokens: if $x_t \neq m$, then $f^*(x_t, t) = f(x_t, t) = x_t$ for any model. Consequently, the corresponding token contribution to $\mathcal{L}_{\text{IDLM}-\text{MDLM}}$ is zero. Hence the IDLM update is nonzero only when the noised token is the mask token, $x_t = m$.

The absorbing-process details are given in Appendix B. Appendix B.2 and Figure 11 show this mask-only cancellation explicitly. Therefore, for the generator update, the MDLM loss reduces to

$$\mathcal{L}_{\text{MDLM}}(f, p_\theta) = -\mathbb{E}_{\epsilon, t}\big[(1 - \alpha_t)\lambda_t\langle G_\theta(\epsilon), \log f(m, t)\rangle\big].$$

The sampled token $x_t$ no longer appears in the generator update. The remaining dependence on $\theta$ is only through the differentiable simplex-valued output $G_\theta(\epsilon)$. Appendix B.2 explains why this can be viewed as an implicit stop-gradient through the intermediate state for masked diffusion.

**Duo: Gaussian reparameterization.** Duo provides a differentiable path by introducing an auxiliary Gaussian variable $\xi$ and using the relaxed state

$$x_t = \text{softmax}\left(\frac{\tilde{\alpha}_t G_\theta(\epsilon) + \sqrt{1 - \tilde{\alpha}_t^2}\,\xi}{\tau}\right), \qquad \xi \sim \mathcal{N}(0, I),$$

where the red terms indicate the path through the generator parameters. The map $G_\theta(\epsilon) \mapsto x_t$ is differentiable. This is possible in the Duo parameterization because the denoiser is trained on simplex-valued states rather than only on discrete one-hot tokens. More details are given in Appendix C.2.

### 3.3. Loss Intuition

**Alternating Optimization.** The IDLM objective contains an inner optimization over the fake diffusion model $f$. In practice, we approximate this nested problem by alternating two updates, following prior work (Yin et al., 2024b;a; Zhou et al., 2024; Gushchin et al., 2025; Kornilov et al., 2026).

First, we fix the student generator $G_\theta$ and train $f$ on samples from the current student distribution $p_\theta$. This approximates the inner minimization in the IDLM objective:

$$\begin{aligned} &\textit{Update: } f; \quad \textit{Fix: } \theta \qquad\qquad (8) \\ &\mathcal{L}_f = \mathcal{L}(f, p_\theta). \end{aligned}$$

Second, we fix $f$ and update $G_\theta$ using the IDLM gap:

$$\begin{aligned} &\textit{Update: } \theta; \quad \textit{Fix: } f \qquad\qquad (9) \\ &\mathcal{L}_{\text{IDLM}}(\theta) = \mathcal{L}(f^*, p_\theta) - \mathcal{L}(f, p_\theta). \end{aligned}$$

These two updates are summarized visually in Figure 2, which depicts the fake model and the student generator as the two learnable components of the IDLM training loop.

**IDLM intuition.** For MDLM, the SUBS parameterization makes the generator update especially transparent: only masked positions contribute. Substituting the mask-only MDLM loss into the IDLM gap gives

$$\mathcal{L}_{\text{IDLM}-\text{MDLM}}(\theta) = -\mathbb{E}_{\epsilon, t}\big[\lambda_t\langle G_\theta(\epsilon), a_t\rangle\big].$$

Here $a_t = \log f^*(m, t) - \log f(m, t)$, and $f$ is the current fake model. Thus each masked position gives the generator a teacher-over-fake advantage vector $a_t$. If $G_\theta(\epsilon) = \text{softmax}(z_\theta)$, then for a fixed $(\epsilon, t)$

$$\frac{\partial \mathcal{L}_{\text{IDLM-MDLM}}}{\partial z_{\theta,i}} = -\lambda_t G_{\theta,i}(\epsilon) \Big( a_{t,i} - \langle G_\theta(\epsilon), a_t \rangle \Big).$$

Hence gradient descent increases token $i$'s logit exactly when its advantage is above the generator's current average advantage. In words, the student does not simply chase the teacher's most likely token, it promotes tokens that the teacher prefers more than the fake model does. When the fake model catches up to the teacher on student samples, this relative advantage vanishes.

| Case | Token behavior | Intuition |
|---|---|---|
| $a_{t,i} > \langle G_\theta, a_t \rangle$ | logit increases | $f^*$ prefers token $i$ more than $f$ does |
| $a_{t,i} < \langle G_\theta, a_t \rangle$ | logit decreases | token $i$ is below the current advantage |
| $a_t \approx 0$ | no relative signal | $f$ has matched $f^*$ on student samples |

For uniform diffusion, the same idea applies to a noised transition rather than only to a masked clean-token prediction. The local advantage becomes

$$a_t = g_{\text{Duo}}(x_t, G_\theta(\cdot), f^*(x_t, t)) - g_{\text{Duo}}(x_t, G_\theta(\cdot), f(x_t, t)),$$

Appendix C gives $g_{\text{Duo}}$, and Appendix C.2 expands the advantage. Since uniform diffusion can revise tokens repeatedly, this advantage guides both which token to choose and when the reverse path should jump.

**Why few-step distillation is difficult.** Standard discrete diffusion samplers usually draw the next state conditionally independently across output positions, even though each factor can attend to the whole current sequence. With many reverse steps, the composition of these local transitions can still represent a rich sequence distribution. In the few-step regime, however, each reverse step must replace many small teacher updates, making this factorization a bottleneck. IDLM addresses this bottleneck by learning a mixture of sequence-level distributions, as described next.

**Sequence-level mixture.** A full distribution over $\mathcal{V}^L$ would require $N^L$ probabilities, which is intractable to parameterize directly. The key is that the student uses a structured mixture: it samples a shared variable $\epsilon \sim p_{\mathcal{E}}$ and outputs $G_\theta^{1:L}(\epsilon) = (G_\theta^1(\epsilon), \ldots, G_\theta^L(\epsilon)) \in \Delta^L$. For this fixed $\epsilon$, the component may factorize over positions,

$$p_\theta(x_0^{1:L} \mid \epsilon) = \prod_{i=1}^L \text{Cat}\big(x_0^i; G_\theta^i(\epsilon)\big),$$

$$p_\theta(x_0^{1:L}) = \mathbb{E}_{\epsilon \sim p_{\mathcal{E}}}\big[p_\theta(x_0^{1:L} \mid \epsilon)\big].$$

The product describes one component, while the expectation mixes many such components. Since the same $\epsilon$ controls all positions, a component can encode a sentence-level choice. Although we use simplex-valued outputs for differentiable training, in theory the generator should produce vocabulary tokens. Hence, during training, each $G_\theta^i(\epsilon)$ should concentrate near a one-hot token, so the component degenerates to a point mass on one full sequence. Thus the generator represents text as a mixture over coherent sequence-level atoms, allowing $p_\theta$ to capture semantic correlations without explicitly enumerating the full sequence space. A similar latent-mixture intuition appears in VADD (Xie et al., 2026) and Di4C$^2$ (Hayakawa et al., 2025), but with different objectives.

### 3.4. Technical Aspects

This section specifies what latent space we use in practice and clarifies the remaining implementation details. The training update is summarized in Algorithm 1, the few-step sampler in Algorithm 2, and the full pipeline in Figure 2.

**Model initialization.** We initialize both the student generator $G_\theta$ and the auxiliary fake model $f$ from the pretrained teacher $f^*$. Since the teacher has already learned the semantic structure of the token space, this provides a strong initialization and lets the student refine the teacher's predictions rather than learn them from scratch.

**Multistep distillation.** We found that directly distilling the model into a one-step generator is difficult in practice, as the student must map terminal noise to a clean sample in a single pass. We therefore allow $G_\theta$ to produce samples in the few-step regime. Concretely, we set the generator latent input to $\epsilon = (x_t, t)$. During training, these inputs are obtained by corrupting real data:

$$x_0 \sim p^*(x_0), \qquad t \sim \mathcal{U}[0, 1], \qquad x_t \sim q_t(\cdot \mid x_0),$$

and then feeding $(x_t, t)$ to the student. This multistep parameterization allows the generator to accept intermediate noised states as input, so at inference we can use the same few-step reverse sampling procedure as the teacher model, see Algorithm 2.

**Sequence-level loss.** All practical updates use the sequence-level loss introduced in the (§2.1.3). That is, the token-level objective $\mathcal{L}(f, p)$ is replaced by $\mathcal{L}^{1:L}(f, p)$. The forward corruption factorizes across positions, but the teacher, fake model, and student are evaluated on the full noised sequence.

## 4. Related Works

**Continuous diffusion distillation.** Continuous distillation has two main branches: consistency-like methods (Song et al., 2023; Song & Dhariwal, 2024), which train students to jump along teacher trajectories, and distribution-matching

**Algorithm 1 IDLM: training.**

One branch is active per update: fit the fake model or distill the student.

```
# teacher f*, fake model f, and student generator net
# mode in {"fake", "student"} selects the updated
    component

x_0 = sample_data(p_data)
t = uniform(0, 1)
x_t = corrupt(x_0, t)   # q_t(· | x_0)
x_hat = net(x_t, t)

if mode == "fake":
    loss = L_seq(f, x_hat)
    update(f, loss)
else:
    loss = L_seq(f*, x_hat) - L_seq(f, x_hat)
    update(net, loss)
```

**Algorithm 2 IDLM: few-step sampling.**

The reverse transition is chosen by the diffusion process.

```
# grid = [t_K, ..., t_0]
# masked or uniform terminal distribution
x_t = sample_terminal(process)
for k in range(K, 0, -1):
    t = grid[k]
    t_prev = grid[k - 1]
    x_hat = net(x_t, t)

    # masked or uniform reverse step
    x_prev = sample_reverse(x_t, x_hat, t, t_prev)
    x_t = x_prev
return x_t
```

methods (Yin et al., 2024b;a; Zhou et al., 2024; Huang et al., 2024; Gushchin et al., 2025; Kornilov et al., 2026; Selikhanovych et al., 2026), which directly optimize the student distribution, usually with an auxiliary or fake model. IDLM follows the distribution-matching view and extends it to discrete diffusion.

**Inverse distillation.** Inverse distillation directly optimizes a student generator to approximate the target distribution, placing it within the family of distribution-matching distillation methods. This idea was introduced by IBMD for diffusion bridge models (Gushchin et al., 2025), then generalized to continuous-space flow and diffusion models in UID framework (Kornilov et al., 2026). RSD further extended this view to discrete-time diffusion and clarified its relation to DMD: a DMD-like update appears after a stop-gradient reduction (Selikhanovych et al., 2026). IDLM adopts the same inverse view, but extends it to discrete space.

**Consistency-style distillation for DLMs.** The consistency branch has discrete analogues such as SDTT (Deschenaux & Gulcehre, 2025) and Duo-DCD (Sahoo et al., 2025). However, the objective mainly teaches trajectory skipping. In practice, this preserves a conditional-independence factorization of the teacher's denoising process, which can miss the full joint distribution and favor common modes. Thus, especially in the one-step limit, these methods lack guaranteeing recovery of the data distribution and remain largely empirical. In contrast, IDLM represents the student as a sequence-level mixture, see (§3.3), and Theorem 3.1 shows that its ideal objective is minimized only by the data distribution.

**Distribution matching in discrete space.** Discrete distribution-matching methods such as Di[M]O (Zhu et al., 2025), D-MMD (Hoogeboom et al., 2026), and DiDi-Instruct (Zheng et al., 2026) match teacher and auxiliary signals at sampled noising times, corresponding to KL match-

ing of noised marginals,

$$\mathcal{L}_{\mathrm{DMD}}(\theta) = \int_0^1 w(t)\, D_{\mathrm{KL}}\big(p_t^\theta \,\|\, p_t^*\big)\; dt.$$

Here $p_t^\theta$ and $p_t^*$ are the student and target noised marginal distributions at time $t$. IDLM instead matches full diffusion path measures,

$$\mathcal{L}_{\mathrm{IDLM}}(\theta) = D_{\mathrm{KL}}\big(\mathbb{P}^\theta \,\|\, \mathbb{P}^*\big),$$

so the comparison is trajectory-level rather than marginal-only. A DMD-like update appears only after a stop-gradient reduction (Selikhanovych et al., 2026), so in masked diffusion, because of the SUBS parameterization, the marginal and path-based losses coincide. In uniform diffusion, this distinction matters in practice, as our experiments show.

**Continuous-space language flows.** A parallel line changes the state space: FLM/FMLM (Lee et al., 2026) use one-hot flows, while ELF (Hu et al., 2026) uses embedding-space flows. These models can obtain strong GenPPL, but their Entropy can drop relative to other baselines. This matters because lower GenPPL may trade off Entropy, for example when sweeping the sampling temperature. Moreover, FLM underperforms on recent benchmarks (Deschenaux & Gulcehre, 2026). Thus, it remains unclear whether a continuous or discrete state space is preferable for language modeling.

## 5. Experiments

This section demonstrates the applicability of our IDLM distillation method across several types of DLMs. We conduct the main experiments for MDLM and Duo. Although we focus on these two models, our theoretical formulation also covers the general SEDD setting. Therefore, we report an additional IDLM-SEDD evaluation in Appendix A.3. Representative generations are shown in Figure 5. Further implementation details, additional experimental results, and

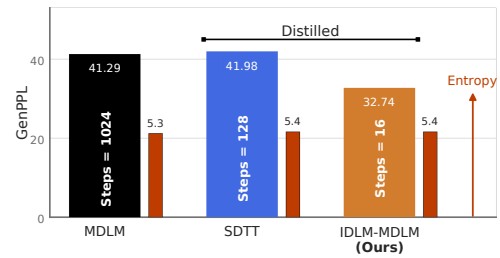

## Masked Diffusion

| | Steps | GenPPL ↓ | MAUVE ↑ | Entropy ↑ | GM ↓ |
|---|---|---|---|---|---|
| MDLM (Teacher) (Sahoo et al., 2024) | 1024 | 41.29 | 0.89 ± 0.01 | 5.28 | 0.672 |
| SDTT (Deschenaux & Gulcehre, 2025) | | 41.45 | 0.91 ± 0.03 | **5.31** | – |
| SDTT + Di4C$^2$ (Hayakawa et al., 2025) | | 31.51 | **0.93** ± 0.02 | 5.28 | – |
| DiDi-Instruct (Zheng et al., 2026) | 32 | 24.97 | – | 5.18 | – |
| D-MMD (Hoogeboom et al., 2026) | | 72.1 | – | **5.57** | 0.578 |
| IDLM-MDLM (**Ours**) | | **20.45** ± 1.88 | 0.91 ± 0.01 | 5.23 ± 0.06 | 0.629 |
| SDTT (Deschenaux & Gulcehre, 2025) | | 61.34 | 0.88 ± 0.03 | 5.36 | – |
| SDTT + Di4C$^2$ (Hayakawa et al., 2025) | | 44.82 | 0.90 ± 0.01 | 5.34 | – |
| DiDi-Instruct (Zheng et al., 2026) | 16 | 38.19 | – | 5.21 | – |
| D-MMD (Hoogeboom et al., 2026) | | 67.7 | – | **5.57** | 0.558 |
| IDLM-MDLM (**Ours**) | | **32.75** ± 1.62 | **0.93** ± 0.01 | 5.42 ± 0.06 | **0.527** |
| SDTT (Deschenaux & Gulcehre, 2025) | | 121.31 | 0.89 ± 0.03 | 5.39 | – |
| SDTT + Di4C$^2$ (Hayakawa et al., 2025) | 8 | 84.15 | 0.90 ± 0.04 | 5.38 | – |
| DiDi-Instruct (Zheng et al., 2026) | | **62.24** | – | 5.17 | – |
| IDLM-MDLM (**Ours**) | | 79.75 ± 5.61 | **0.91** ± 0.01 | **5.61** ± 0.04 | 0.835 |

## Uniform & Continuous Diffusion

| | Steps | GenPPL ↓ | MAUVE ↑ | Entropy ↑ |
|---|---|---|---|---|
| FLM (Lee et al., 2026) | 1024 | 62.23 | - | **5.33** |
| Duo$^g$ (Teacher) (Sahoo et al., 2025) | | 71.72 | 0.90 ± 0.02 | 5.22 |
| ELF (Hu et al., 2026) | | **24.08** ± 0.16 | – | 5.14 |
| FMLM (Lee et al., 2026) | | 45.09 | – | 5.25 |
| Duo-DCD$^g$ (Sahoo et al., 2025) | 32 | 46.31 | **0.96** ± 0.01 | **5.38** |
| IDLM-DCD$^g$ (**Ours**) | | 39.34 ± 8.51 | 0.92 ± 0.01 | 5.35 ± 0.11 |
| ELF (Hu et al., 2026) | | **33.66** ± 1.09 | – | 5.16 |
| FMLM (Lee et al., 2026) | | 63.63 | – | 5.29 |
| Duo-DCD$^g$ (Sahoo et al., 2025) | 16 | 54.11 | **0.96** ± 0.01 | 5.37 |
| IDLM-DCD$^g$ (**Ours**) | | 43.36 ± 3.76 | 0.94 ± 0.01 | **5.41** ± 0.04 |
| ELF (Hu et al., 2026) | | 67.32 ± 2.25 | – | 5.14 |
| FMLM (Lee et al., 2026) | | 86.5 | – | 5.36 |
| Duo-DCD$^g$ (Sahoo et al., 2025) | 8 | 69.58 | 0.93 ± 0.01 | 5.30 |
| IDLM-DCD$^g$ (**Ours**) | | **53.61** ± 3.08 | **0.96** ± 0.01 | **5.41** ± 0.03 |
| FMLM (Lee et al., 2026) | | 111.31 | – | 5.26 |
| Duo-DCD$^g$ (Sahoo et al., 2025) | 4 | 96.24 | 0.69 ± 0.02 | 4.93 |
| IDLM-DCD$^g$ (**Ours**) | | **77.47** ± 4.43 | **0.89** ± 0.01 | **5.28** ± 0.06 |

*Figure 3.* **OWT distillation.** IDLM preserves the quality/diversity tradeoff while using far fewer reverse steps for both masked and uniform diffusion. The strongest gains appear in the low-step regime.

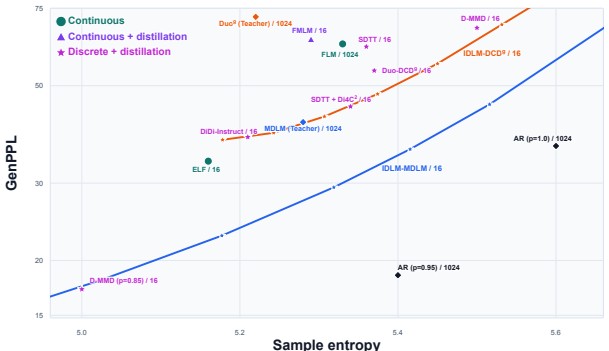

*Figure 4.* **GenPPL–Entropy tradeoff on OWT.** Each point denotes a Method/Sampling steps. Lower GenPPL and higher entropy are preferred.

uncurated text samples are provided in Appendices E, F, and G, respectively.

### 5.1. Unconditional Generation on OpenWebText

We use the suffix to indicate the teacher: IDLM-MDLM distills MDLM, IDLM-Duo distills the original Duo model, and IDLM-DCD distills the Duo-DCD model. We describe the calculation of all evaluation metrics used in this section in Appendix E.1.

**Teacher acceleration.** Figure 3 summarizes how well IDLM distills teachers. For masked diffusion MDLM,

IDLM-MDLM reduces the 1024-step teacher to 16 reverse steps, giving a (64×) acceleration while preserving GenPPL and Entropy metrics. For Duo with the greedy sampler (§2.1.2), IDLM-Duo also reaches 16 steps (64×), while IDLM-DCD$^g$ reaches 4 steps (256×). The corresponding ancestral Duo results are reported in Appendix F.1, where IDLM-DCD$^a$ is evaluated down to 4 steps and gives its best speed-quality tradeoff at 8 steps (128×). Also, GenPPL and Entropy vs sampling-step curves are provided in Figure 13.

**Comparison with baselines.** The lower panels of Figure 3 compare IDLM against other baselines with GenPPL, MAUVE, Entropy, and GM where available. For absorbing-state distillation, IDLM-MDLM gives the best overall low-step tradeoff at 16 steps, improving GenPPL while preserving diversity. For uniform-state distillation, IDLM-DCD$^g$ is strongest in the 8–4 step regime. The ancestral step curve in Figure 13 and the detailed results in Table 2 show the same trend.

**GenPPL–Entropy tradeoff.** Recent evaluations of flow and diffusion language models show that GenPPL should be interpreted together with sample entropy: lowering entropy can substantially improve GenPPL without necessarily improving the underlying generative distribution. Figure 4 visualizes this tradeoff on OWT. IDLM-MDLM improves GenPPL at comparable entropy to the MDLM teacher, and IDLM-DCD remains competitive.

**Scaling.** We additionally test IDLM on an 0.9B MDLM checkpoint from SDTT (Deschenaux & Gulcehre, 2025),

---

📝 **Unconditional Generation (OpenWebText)**

- - - - - - - - - - - - - - - - - - - - - - - - - - - - - - - - - - - - - - - - - - - - - - - - - - - - - - - - - - -

**IDLM-MDLM / 16 Sampling steps**

**Gen. PPL: 34.1    H: 5.43**

... After this short presentation, let me briefly describe the layout of the grid. I initially intended to drag the page away from the phone screen and place it onto the grid platform. ...

**IDLM-DCD$^g$ / 4 Sampling steps**

**Gen. PPL: 77.5    H: 5.28**

... The thing, programs are a language, and structures are highly hierarchical. With the complexity of a programming language, the underlying structure grows, and you improve that structure. ...

---

📄 **Conditional Generation (TinyGSM)**

- - - - - - - - - - - - - - - - - - - - - - - - - - - - - - - - - - - - - - - - - - - - - - - - - - - - - - - - - - -

**IDLM-MDLM / 128 Sampling steps**

**Accuracy: 19.9%**

**Prompt.** A baseball coach buys 9 baseballs for \$3 each, and a basketball coach buys 8 basketballs for \$14 each. How much more did the basketball coach spend?

**Generation.**

```
def simple_math_problem() -> int:
    baseball_cost = 9 * 3
    basketball_cost = 8 * 14
    difference = basketball_cost - baseball_cost
    result = difference
    return result
```

**IDLM-Duo / 64 Sampling steps**

**Accuracy: 19.0%**

**Prompt.** The red car is 40% cheaper than the blue car. The price of the blue car is \$100. How much do both cars cost?

**Generation.**

```
def simple_math_problem() -> int:
    blue_car = 100
    red_car = blue_car * 0.60
    total_cost = blue_car + red_car
    result = total_cost
    return result
```

---

*Figure 5.* **Qualitative examples.** Representative IDLM samples. Additional qualitative samples are provided in Appendix G.

the full scaling comparison with SDTT is reported in Table 3. The same pattern persists at larger scale: IDLM improves the low-step GenPPL/Entropy tradeoff over SDTT.

**Ablation study.** We ablate the main IDLM design choices in Appendix F.4. The results indicate that the original masked-diffusion update gives the best early GenPPL–Entropy trade-off, while the full IDLM-Duo update is more stable than its stop-gradient variant. Adding simple Gaussian input noise to IDLM-MDLM does not improve performance (Table 4).

## 5.2. Conditional Generation on TinyGSM

GenPPL and entropy measure unconditional sample quality, but they do not directly test whether a model preserves sequence-level correctness. We therefore follow the TinyGSM/GSM8K protocol used by Deschenaux & Gulcehre (2026). The results in Table 1 show that IDLM improves steadily with more reverse steps and recovers teacher-level conditional generation with far fewer sampling steps. IDLM-MDLM reaches teacher-level accuracy 19.86% with only 128 steps, matching and slightly surpassing the 1024-step MDLM teacher at 18.0%, corresponding to an (8×) reduction in sampling steps. For Duo, IDLM-Duo already reaches comparable accuracy 19.03% at 64 steps relative to the 1024-step Duo teacher at 17.2%, giving a (16×) reduction. Overall, IDLM retains conditional generation with far fewer steps.

*Table 1.* **Conditional generation on GSM8K.** Baseline accuracies are reproduced from Deschenaux & Gulcehre (2026).

| Model | Steps | Accuracy (%) |
|---|---|---|
| *Autoregressive* | | |
| Sample | | 53.9 |
| Greedy | 512 | **63.3** |
| *Diffusion* | | |
| MDLM (Teacher) (Sahoo et al., 2024) | | **18.0** |
| Duo (Teacher) (Sahoo et al., 2025) | 1024 | 17.2 |
| FLM (Lee et al., 2026) | | 0.3 |
| S-FLM (Deschenaux & Gulcehre, 2026) | | **18.0** |
| *Distillation* | | |
| IDLM-MDLM (**Ours**) | | 19.86 |
| IDLM-Duo (**Ours**) | 128 | **21.38** |
| IDLM-MDLM (**Ours**) | | 14.94 |
| IDLM-Duo (**Ours**) | 64 | **19.03** |
| IDLM-MDLM (**Ours**) | | 12.81 |
| IDLM-Duo (**Ours**) | 32 | **15.39** |

## 6. Discussion and Conclusion

We extend Inverse Distillation to pretrained Diffusion Language Models (DLMs), deriving the IDLM loss (§3.1) and practical discrete relaxations for non-smooth token spaces (§3.2). Experiments (§5) show that IDLM accelerates teacher sampling by 4×–64× across DLM families while preserving generation quality.

### 6.1. Limitations and Future Work

Our experiments use relatively small-scale DLMs; a natural next step is to test whether IDLM's few-step gains persist for larger models such as LLaDA (Nie et al., 2025) and on standardized downstream benchmarks.

## Acknowledgements

The work was supported by the grant for research centers in the field of AI provided by the Ministry of Economic Development of the Russian Federation in accordance with the agreement 000000C313925P4F0002 and the agreement №139-10-2025-033.

## Impact Statement

This paper presents work whose goal is to advance the field of Machine Learning. There are many potential societal consequences of our work, none which we feel must be specifically highlighted here.

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

# Contents

## Main Paper

## Appendix

# A. SEDD Formulation, Theory, and Experiments

This appendix collects the material needed to use SEDD within the IDLM framework. Appendix A.1 gives the CTMC score-entropy formulation and its sequence factorization. Appendix A.2 specializes the IDLM objective to SEDD and explains the resulting teacher–fake advantage. Appendix A.3 reports the SEDD experimental validation. We use the notation of Section 2.1: $x_0, x_t, y \in \mathcal{V}$ are one-hot tokens, $p^*$ is the data distribution, $p_\theta$ is the student distribution, and $q_t(x_t \mid x_0)$ is the forward noising kernel. For a matrix $A$, write $\langle u, v \rangle_A := u^\top A v$; when the subscript is omitted, $\langle u, v \rangle = u^\top v$.

## A.1. SEDD Formulation and Theory

**Forward process.** As stated in Section 2.1, the main text focuses on absorbing and uniform diffusion and defers the general CTMC formulation to this appendix. A SEDD forward process defines noised data marginals $p_t \in \Delta$ through

$$\frac{dp_t}{dt} = Q_t p_t, \quad p_0 = p^*, \tag{10}$$

where each diffusion matrix $Q_t \in \mathbb{R}^{N \times N}$ has nonnegative off-diagonal entries and columns summing to zero. The off-diagonal entries are transition rates between tokens. The zero column-sum condition preserves total probability in Eq. (10), because probability mass that leaves a token is balanced by mass assigned to outgoing transitions. In the common parameterization $Q_t = \sigma_t Q$, the fixed matrix $Q$ determines the diffusion process, including which token-to-token transitions are allowed, while $\sigma_t$ controls the noise rate. Let $\bar{\sigma}_t := \int_0^t \sigma_s ds$. Then $\exp(\bar{\sigma}_t Q)$ is the matrix exponential of $\bar{\sigma}_t Q$, and for a fixed clean token $x_0$ we write

$$q_t(x_t \mid x_0) := \mathrm{Cat}\big(x_t; \exp(\bar{\sigma}_t Q) x_0\big). \tag{11}$$

The corresponding noised data marginal is $p_t(x_t) = \int q_t(x_t \mid x_0) \, p^*(dx_0)$. Absorbing and uniform diffusion are specific choices of $Q$ and are described in Appendices B and C.

**Training loss.** In Section 2.1.3, SEDD is one instance of the unified token-level objective $\mathcal{L}(f, p)$. SEDD (Lou et al., 2024) learns a positive score model $f : \mathcal{V} \times [0, 1] \to \mathbb{R}_+^N$ that estimates reverse-process probability ratios. For a fixed clean token $x_0$, define the conditional score ratio

$$\langle s(x_t, t \mid x_0), y \rangle := \frac{q_t(y \mid x_0)}{q_t(x_t \mid x_0)}.$$

For any clean-token distribution $p$, the score-entropy objective has the same distribution-level form as the MDLM and Duo losses:

$$\mathcal{L}_{\mathrm{SEDD}}(f, p) := \mathbb{E}_{p(x_0), t, q_t}\big[g_{\mathrm{SEDD}}\big(x_t, x_0, f(x_t, t)\big)\big],$$

$$g_{\mathrm{SEDD}}\big(x_t, x_0, f(x_t, t)\big) := \sum_{y \neq x_t} \langle x_t, y \rangle_{Q_t} \left( \langle f(x_t, t), y \rangle - \langle s(x_t, t \mid x_0), y \rangle \log \langle f(x_t, t), y \rangle \right). \tag{12}$$

Here $\langle x_t, y \rangle_{Q_t}$ weights the loss by the forward transition rate from $y$ to $x_t$, so transitions not allowed by $Q_t$ do not contribute. For a generic training distribution $p$, we use the same notation $p_t$ for the noised marginal induced by $p$:

$$p_t(x) = \int q_t(x \mid x_0) \, p(dx_0).$$

At a minimizer of Eq. (12), the model satisfies

$$\langle f(x_t, t), y \rangle = \mathbb{E}_{p(x_0 \mid x_t)}\big[\langle s(x_t, t \mid x_0), y \rangle\big] = \frac{p_t(y)}{p_t(x_t)}, \qquad y \neq x_t. \tag{13}$$

In particular, when $p = p^*$ we denote the trained teacher by $f^*$, which gives

$$\langle f^*(x_t, t), y \rangle = \frac{p_t(y)}{p_t(x_t)}, \qquad y \neq x_t. \tag{14}$$

Thus, the trained SEDD model provides the marginal ratios required by the reverse process.

**Sampling procedure.**  Sampling uses the learned score model to instantiate a reverse-time CTMC. The reverse process starts from the tractable terminal distribution $\pi$ and evolves backward in time:

$$\frac{dp_{1-t}}{dt} = \overline{Q}_{1-t}^f p_{1-t}, \quad p_1 = \pi. \tag{15}$$

For $y \neq x_t$, the reverse transition rate induced by $f$ is

$$\langle y, x_t \rangle_{\overline{Q}_t^f} := \langle f(x_t, t), y \rangle \langle x_t, y \rangle_{Q_t}, \qquad \langle x_t, x_t \rangle_{\overline{Q}_t^f} = -\sum_{y \neq x_t} \langle y, x_t \rangle_{\overline{Q}_t^f}. \tag{16}$$

With the teacher $f^*$, Eq. (16) becomes the exact reverse rate

$$\langle y, x_t \rangle_{\overline{Q}_t^{f^*}} = \frac{p_t(y)}{p_t(x_t)} \langle x_t, y \rangle_{Q_t}.$$

Therefore, SEDD learns the probability ratios that convert the forward CTMC rates into reverse CTMC rates. Equivalently, for a small reverse step $h > 0$, these rates define the transition probabilities

$$\mathbb{P}(X_{t-h} = y \mid X_t = x_t) = \delta_{y,x_t} + h\langle y, x_t \rangle_{\overline{Q}_t^f} + o(h),$$

where $\delta_{y,x_t}$ is the Kronecker delta. For $y \neq x_t$, this reduces to

$$\mathbb{P}(X_{t-h} = y \mid X_t = x_t) = h\langle y, x_t \rangle_{\overline{Q}_t^f} + o(h),$$

while the diagonal term gives the probability of staying at $x_t$ up to order $h$. Since $X_{t-h}$ is a discrete random vector, its expected one-step displacement is

$$\mathbb{E}[X_{t-h} - x_t \mid X_t = x_t] = \sum_{y \in \mathcal{V}} (y - x_t)\mathbb{P}(X_{t-h} = y \mid X_t = x_t) = \sum_{y \neq x_t} (y - x_t)\mathbb{P}(X_{t-h} = y \mid X_t = x_t),$$

where the two sums are equal because the $y = x_t$ term is zero. Substituting the small-step probabilities yields

$$\mathbb{E}[X_{t-h} - x_t \mid X_t = x_t] = h\sum_{y \neq x_t} (y - x_t)\langle y, x_t \rangle_{\overline{Q}_t^f} + o(h).$$

The delta and diagonal terms do not contribute to the displacement because they correspond to $y = x_t$, where $y - x_t = 0$. Thus the reverse CTMC rates determine both the probabilities of individual jumps and the average local reverse update. We use this quantity below to interpret IDLM-SEDD. For a fixed corrupted token $x_t$, a SEDD model does not choose a single denoised token directly; it assigns rates to possible reverse jumps $x_t \to y$. The expectation $\mathbb{E}[X_{t-h} - x_t \mid X_t = x_t]$ compresses these local jump rates into the average one-step denoising motion induced by the model. Thus, it gives a compact summary of how the model locally moves $x_t$ toward cleaner tokens; related local-move views are used in discrete flow matching (Gat et al., 2024).

**Loss for sequences.**  For language modeling, directly comparing all token pairs in a full sequence is expensive for large vocabularies. SEDD avoids this by using a coordinate-wise CTMC: each transition changes one position, while the forward conditional distribution factorizes across positions,

$$q_t(x_t^{1:L} \mid x_0^{1:L}) = \prod_{\ell=1}^{L} q_t(x_t^\ell \mid x_0^\ell).$$

Consequently, the conditional score ratio also factorizes. If $y^{1:L}$ differs from $x_t^{1:L}$ only at position $\ell$, then all unchanged-position factors cancel:

$$\frac{q_t(y^{1:L} \mid x_0^{1:L})}{q_t(x_t^{1:L} \mid x_0^{1:L})} = \prod_{j=1}^{L} \frac{q_t(y^j \mid x_0^j)}{q_t(x_t^j \mid x_0^j)} = \frac{q_t(y^\ell \mid x_0^\ell)}{q_t(x_t^\ell \mid x_0^\ell)}, \qquad y^j = x_t^j \text{ for } j \neq \ell.$$

The sequence-level objective follows the unified notation of Section 2.1.3:

$$\mathcal{L}_{\text{SEDD}}^{1:L}(f, p) = \mathbb{E}_{p(x_0^{1:L}), t, q_t}\left[\sum_{\ell=1}^{L} g_{\text{SEDD}}\left(x_t^\ell, x_0^\ell, f^\ell(x_t^{1:L}, t)\right)\right],$$

where the score at position $\ell$ may condition on the full noised sequence. This factorization makes the general CTMC formulation practical for sequence models.

## A.2. Inverse Distillation for SEDD

We now specialize the IDLM objective from Section 3 to the SEDD loss above. The teacher $f^*$ is fixed and estimates reverse ratios for the data distribution. The fake model $f$ is trained on the current student distribution $p_\theta$ and estimates reverse ratios for student samples. IDLM-SEDD updates the student by comparing these two SEDD losses on the same student-generated examples.

**Student samples.** The expectations in $\mathcal{L}_{\mathrm{SEDD}}(f, p_\theta)$ are taken over

$$x_0 \sim p_\theta(x_0), \qquad t \sim \mathcal{U}[0,1], \qquad x_t \sim q_t(\cdot \mid x_0).$$

Equivalently, when $p_\theta$ is induced by the generator used in Section 3.4, we may write $x_0 = G_\theta(\epsilon)$. The real data distribution affects this update only through the fixed teacher $f^*$; the fake model and the student are evaluated on samples from $p_\theta$.

**IDLM-SEDD objective.** Using Eq. (6), the inverse-distillation objective for SEDD is

$$\mathcal{L}_{\mathrm{IDLM\text{-}SEDD}}(\theta) := \mathcal{L}_{\mathrm{SEDD}}(f^*, p_\theta) - \min_f \mathcal{L}_{\mathrm{SEDD}}(f, p_\theta). \tag{17}$$

In practice, as in the main IDLM algorithm, we alternate between fitting the fake model and updating the student. With $\theta$ fixed, the fake model minimizes $\mathcal{L}_{\mathrm{SEDD}}(f, p_\theta)$. With $f$ fixed, the student minimizes

$$\mathcal{L}_{\mathrm{IDLM\text{-}SEDD}}(\theta) := \mathcal{L}_{\mathrm{SEDD}}(f^*, p_\theta) - \mathcal{L}_{\mathrm{SEDD}}(f, p_\theta). \tag{18}$$

Substituting Eq. (12) into both terms gives

$$\mathcal{L}_{\mathrm{IDLM\text{-}SEDD}}(\theta) = \mathbb{E}_{p_\theta(x_0),\, t,\, q_t} \left[ \sum_{y \neq x_t} \langle x_t, y \rangle_{Q_t} \left( \langle f^*(x_t, t), y \rangle - \langle s(x_t, t \mid x_0), y \rangle \log \langle f^*(x_t, t), y \rangle \right) \right]$$
$$- \mathbb{E}_{p_\theta(x_0),\, t,\, q_t} \left[ \sum_{y \neq x_t} \langle x_t, y \rangle_{Q_t} \left( \langle f(x_t, t), y \rangle - \langle s(x_t, t \mid x_0), y \rangle \log \langle f(x_t, t), y \rangle \right) \right]. \tag{19}$$

Since both expectations use the same $p_\theta(x_0)$, $t$, and $q_t(x_t \mid x_0)$, this simplifies to

$$\mathcal{L}_{\mathrm{IDLM\text{-}SEDD}}(\theta) = \mathbb{E}_{p_\theta(x_0),\, t,\, q_t} \left[ \sum_{y \neq x_t} \langle x_t, y \rangle_{Q_t} \left( \langle f^*(x_t, t) - f(x_t, t), y \rangle - \langle s(x_t, t \mid x_0), y \rangle \log \frac{\langle f^*(x_t, t), y \rangle}{\langle f(x_t, t), y \rangle} \right) \right]. \tag{20}$$

Equation (20) is the fully substituted IDLM-SEDD loss with fixed fake model $f$.

**Advantage intuition.** Following the local-advantage notation of Section 3.3, for a sampled pair $(x_0, x_t)$ define

$$a_t := g_{\mathrm{SEDD}}\big(x_t, x_0, f^*(x_t, t)\big) - g_{\mathrm{SEDD}}\big(x_t, x_0, f(x_t, t)\big)$$
$$= \sum_{y \neq x_t} \langle x_t, y \rangle_{Q_t} \left( \langle f^*(x_t, t) - f(x_t, t), y \rangle - \langle s(x_t, t \mid x_0), y \rangle \log \frac{\langle f^*(x_t, t), y \rangle}{\langle f(x_t, t), y \rangle} \right).$$

In SEDD, $a_t$ compares teacher and fake local score-entropy losses on the same corrupted student sample. The comparison is made over the same jumps that define the reverse CTMC motion in Appendix A.1: each candidate jump $x_t \to y$ contributes according to its forward rate $\langle x_t, y \rangle_{Q_t}$ and the teacher–fake difference in the score assigned to that jump. Equivalently, the teacher $f^*$ and the fake model $f$ induce two average local reverse updates,

$$d_{f^*}(x_t, t) = \sum_{y \neq x_t} (y - x_t)\langle y, x_t \rangle_{\overline{Q}_t^{f^*}}, \qquad d_f(x_t, t) = \sum_{y \neq x_t} (y - x_t)\langle y, x_t \rangle_{\overline{Q}_t^{f}}.$$

The quantity $a_t$ should be read as a local teacher-minus-fake loss, not as a Euclidean distance between $d_{f^*}$ and $d_f$. Since IDLM-SEDD minimizes $a_t$, the generator is encouraged to produce samples $x_0$ whose corrupted versions $x_t$ are better explained by the teacher reverse CTMC than by the fake reverse CTMC. In jump terms, a low teacher loss means that the teacher assigns plausible rates to reverse moves $x_t \to y$ that are compatible with denoising $x_t$ toward $x_0$; equivalently, its

average local update $d_{f^*}(x_t, t)$ gives a useful teacher signal. The fake model contributes the opposite side of the comparison, so the student is pushed away from samples whose local reverse jumps are explained only by the fake model.

The sign of $a_t = g_{\text{SEDD}}(f^*) - g_{\text{SEDD}}(f)$ gives the corner cases. If $a_t < 0$, then the teacher has lower local SEDD loss than the fake model: the teacher likes the reverse jumps around this corrupted sample more, and the generator receives an attractive signal for such samples. If $a_t > 0$, then the fake model has lower local loss: the sample is better explained by the current student-trained dynamics than by the teacher, so minimizing the objective discourages the generator from producing it. If $a_t \approx 0$, the two models give nearly the same local explanation, and the relative teacher-versus-fake signal is weak.

At the sequence level, we use the factorized SEDD loss from Appendix A.1:

$$\mathcal{L}_{\text{IDLM-SEDD}}^{1:L}(\theta) := \mathcal{L}_{\text{SEDD}}^{1:L}(f^*, p_\theta) - \min_f \mathcal{L}_{\text{SEDD}}^{1:L}(f, p_\theta).$$

### A.3. SEDD Experimental Validation

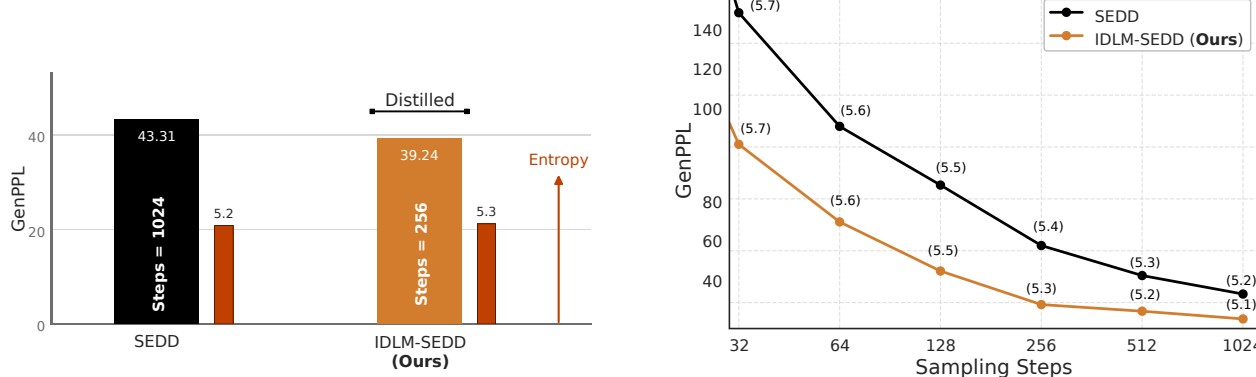

*Figure 6.* **IDLM-SEDD comparison with SEDD.** *Left:* IDLM-SEDD matches SEDD generation quality and diversity while reducing sampling from 1024 to 256 steps ($4\times$). *Right:* IDLM-SEDD improves GenPPL across sampling steps while preserving high entropy.

We use SEDD to evaluate IDLM under the general CTMC score-entropy formulation. Since only checkpoints for the absorbing-process SEDD variant are publicly available, we distill this variant and refer to it simply as SEDD for clarity.

As shown in Figure 6, IDLM-SEDD improves GenPPL over SEDD across the evaluated sampling-step range while preserving high sequence entropy at low step counts. In particular, it achieves a $4\times$ reduction in sampling steps, accelerating generation from 1024 to 256 steps while maintaining both GenPPL and output diversity. Uncurated samples are provided in Appendix G.

# B. Masked Diffusion

This appendix expands the masked diffusion process used by MDLM. The section has two parts. First, in Appendix B.1, we describe the MDLM forward process, training objective, and reverse sampler. Second, in Appendix B.2, we explain why this process gives a particularly simple inverse distillation signal. Masked diffusion is a process-specific instance of the general continuous-time Markov chain (CTMC) formulation in Appendix A, where ordinary tokens move to the mask state.

## B.1. Masked Diffusion Formulation and Theory

Masked diffusion corrupts text by replacing clean tokens with a special mask token. Reverse sampling then gradually reveals the masked positions. This process is also called absorbing diffusion because, once a token reaches the mask state in the forward process, it stays there.

**Forward process.** Let $m \in \mathcal{V}$ be the mask token. We use column-vector conventions throughout: probability vectors are columns, and each diffusion matrix has columns that sum to zero. The absorbing process sends every non-mask token to $m$ and leaves $m$ fixed. Hence its terminal distribution is the point mass $\pi = m$.

The base generator of this continuous-time Markov chain is

$$Q_{\mathrm{abs}} = \begin{bmatrix} -1 & 0 & \cdots & 0 & 0 \\ 0 & -1 & \cdots & 0 & 0 \\ \vdots & \vdots & \ddots & \vdots & \vdots \\ 0 & 0 & \cdots & -1 & 0 \\ 1 & 1 & \cdots & 1 & 0 \end{bmatrix}. \tag{21}$$

With time-dependent rate $\sigma_t$, the forward generator is

$$Q_t = \sigma_t Q_{\mathrm{abs}} = \begin{bmatrix} -\sigma_t & 0 & \cdots & 0 & 0 \\ 0 & -\sigma_t & \cdots & 0 & 0 \\ \vdots & \vdots & \ddots & \vdots & \vdots \\ 0 & 0 & \cdots & -\sigma_t & 0 \\ \sigma_t & \sigma_t & \cdots & \sigma_t & 0 \end{bmatrix}. \tag{22}$$

If $\bar{\sigma}_t := \int_0^t \sigma_s ds$, then the cumulative transition matrix is

$$\exp(\bar{\sigma}_t Q_{\mathrm{abs}}) = \begin{bmatrix} \exp(-\bar{\sigma}_t) & 0 & \cdots & 0 & 0 \\ 0 & \exp(-\bar{\sigma}_t) & \cdots & 0 & 0 \\ \vdots & \vdots & \ddots & \vdots & \vdots \\ 0 & 0 & \cdots & \exp(-\bar{\sigma}_t) & 0 \\ 1 - \exp(-\bar{\sigma}_t) & 1 - \exp(-\bar{\sigma}_t) & \cdots & 1 - \exp(-\bar{\sigma}_t) & 1 \end{bmatrix}. \tag{23}$$

Equivalently, with $\alpha_t := \exp(-\bar{\sigma}_t)$, a clean token $x_0$ is preserved with probability $\alpha_t$ and absorbed into the mask with probability $1 - \alpha_t$:

$$q_t(x_t \mid x_0) = \mathrm{Cat}\big(x_t; \alpha_t x_0 + (1 - \alpha_t)m\big).$$

Thus, at any time $t$, a token has only two possible forward states: it is still equal to the original clean token, or it has become the mask token. Figure 7 shows the same kernel visually. The important point is that the masked kernel is linear in the clean-token distribution: it can be applied either to a hard one-hot token or to the simplex-valued generator output used later in IDLM.

**Training loss.** MDLM (Sahoo et al., 2024; Shi et al., 2024) uses this absorbing process with a clean-token predictor $f(x_t, t) \in \Delta$. Training is weighted masked-token prediction: when a position has been absorbed into the mask, the model predicts its original clean token.

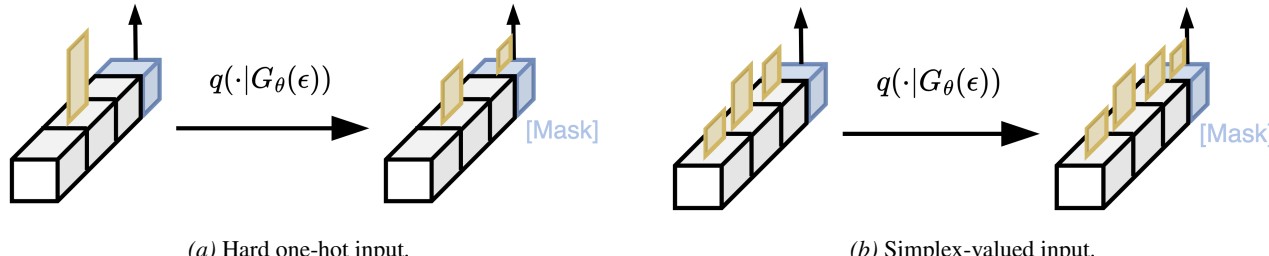

*(a)* Hard one-hot input.

*(b)* Simplex-valued input.

*Figure 7.* Masked forward process. The transition $q_t(\cdot \mid x_0)$ preserves the clean-token mass with probability $\alpha_t$ and moves the remaining mass to the mask token. Because this map is linear, the same operation applies to a one-hot token and to a differentiable simplex output $G_\theta(\epsilon)$.

**Sampling procedure.** Sampling starts from the terminal distribution, i.e. from a fully masked sequence, and then integrates the learned reverse process toward $t = 0$. For $0 \leq s < t \leq 1$, let $f_t := f(x_t, t)$. MDLM uses the absorbing posterior with the true clean token replaced by the model prediction:

$$p_{s|t}^f(x_s \mid x_t) = \begin{cases} \mathrm{Cat}(x_s; x_t), & x_t \neq m, \\ \mathrm{Cat}\left(x_s; \dfrac{1 - \alpha_s}{1 - \alpha_t} m + \dfrac{\alpha_s - \alpha_t}{1 - \alpha_t} f_t\right), & x_t = m. \end{cases}$$

Figure 8 visualizes this reverse sampling path. The two cases simply say that masked diffusion is one-way: a revealed token stays revealed, while a masked token can either stay masked or be revealed. This is the SUBS sampling rule used by MDLM. If the token has already been revealed, SUBS copies it unchanged. If the token is still masked, the reverse step either keeps it masked until time $s$ or reveals it according to the clean-token prediction $f_t$. The noise schedule controls how many tokens are still masked at each time. The model controls which token is chosen when a masked position is revealed.

This is the main contrast with uniform diffusion. In masked diffusion, revealed tokens stay fixed. In uniform diffusion, tokens can still change later, so the IDLM signal for uniform diffusion must track intermediate states; see Appendix C for the corresponding uniform-diffusion formulation.

### B.2. Inverse Distillation for Masked Diffusion

We now connect the masked diffusion process to the IDLM objective. The general inverse distillation loss, introduced in Section 2.2, asks for a student distribution $p_\theta$ on which the pretrained teacher $f^*$ is better than any diffusion model trained only on student samples:

$$\mathcal{L}_{\mathrm{IDLM}}(\theta) = \mathcal{L}(f^*, p_\theta) - \min_{\tilde{f}} \mathcal{L}(\tilde{f}, p_\theta).$$

In practice, the inner minimizer is represented by a fake model $f$ trained on the current student distribution, as in the alternating optimization in Section 3.3. Therefore, during a generator step we use the teacher-minus-fake objective

$$\mathcal{L}_{\mathrm{IDLM}}(\theta) = \mathcal{L}(f^*, p_\theta) - \mathcal{L}(f, p_\theta).$$

For masked diffusion, this gap has an especially transparent form because of the SUBS parameterization.

**Simplex relaxation.** The student generator should ultimately choose vocabulary tokens, but a hard one-hot output blocks ordinary backpropagation. We therefore allow the generator output to lie in the simplex, following Section 3.2,

$$G_\theta(\epsilon) \in \Delta.$$

This is still a valid token distribution: its entries are nonnegative and sum to one. Moreover, the masked forward kernel remains well-defined, as illustrated in Figure 7, because

$$\alpha_t G_\theta(\epsilon) + (1 - \alpha_t)m \in \Delta.$$

Thus the generator update can use a weighted sum over possible clean tokens instead of committing to one sampled token before gradients are computed. Figure 9 shows the relaxation: the discrete target is a one-hot token, but during optimization the generator can expose continuous token weights.

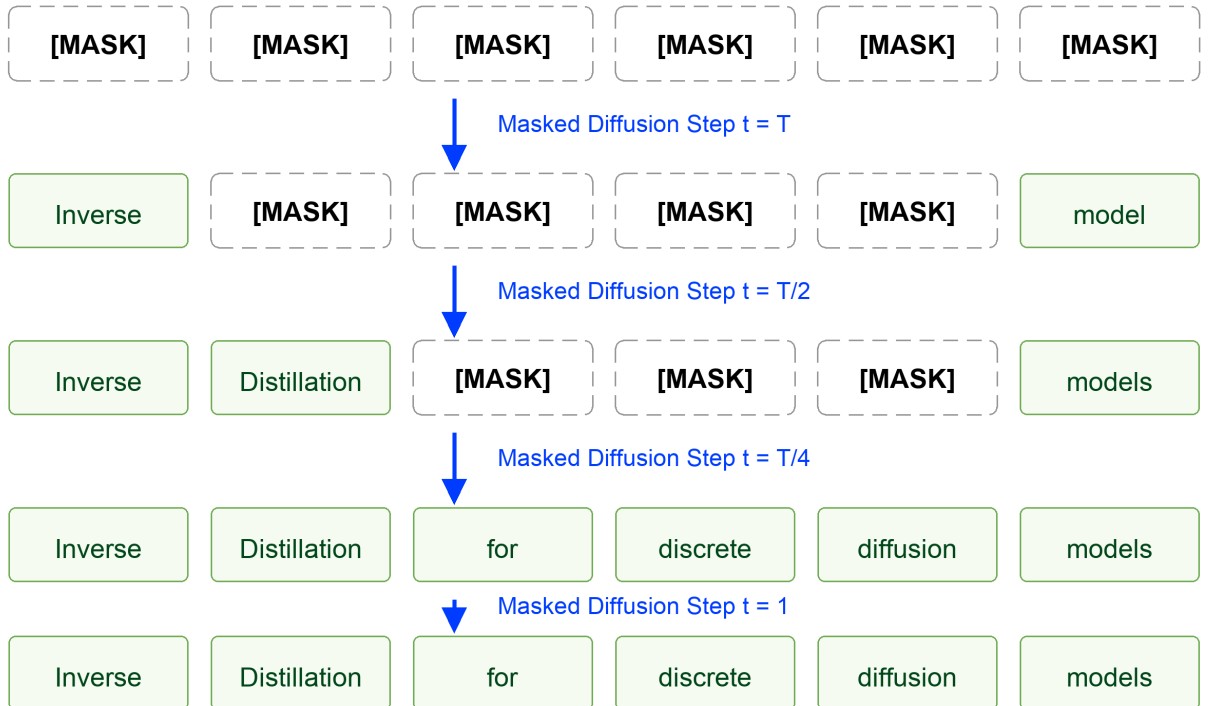

*Figure 8.* Masked-diffusion sampling procedure. The sampler starts from a fully masked sequence and reveals each position once; after a token is revealed, later reverse steps copy it. Thus the schedule decides when a position is revealed, and the model decides which token to put there.

This also changes how the token loss should be read. With a hard one-hot target, the inner product with $\log f(x_t, t)$ simply picks the log-probability of one token. With $G_\theta(\epsilon) \in \Delta$, the same inner product becomes a weighted sum over token log-probabilities, as shown in Figure 10.

**Mask-only cancellation.** The SUBS parameterization copies any already unmasked token:

$$x_t \neq m \quad \implies \quad f^*(x_t, t) = f(x_t, t) = x_t.$$

Therefore, for non-mask states, the teacher and fake model make the same prediction and their contributions cancel in the IDLM gap. Only masked states remain. Figure 11 gives the presentation view of this cancellation. If the forward process returns an unmasked state, the state is already the clean token, so both the teacher and the fake model copy it. If the forward process returns the mask state, the model must infer the clean token, and this is where the IDLM signal survives.

If $G_\theta(\epsilon)$ denotes the simplex-valued student output used in the generator update, the MDLM term reduces to the mask-only loss used in Section 3.2:

$$\mathcal{L}_{\mathrm{MDLM}}(f, p_\theta) = -\mathbb{E}_{\epsilon, t} \left[ (1 - \alpha_t)\lambda_t \left\langle G_\theta(\epsilon), \log f(m, t) \right\rangle \right].$$

**Teacher-over-fake advantage.** Consequently, the IDLM-MDLM generator update is driven by the relative advantage of the teacher over the fake model at the mask state:

$$\mathcal{L}_{\mathrm{IDLM-MDLM}}(\theta) = -\mathbb{E}_{\epsilon, t} \left[ (1 - \alpha_t)\lambda_t \left\langle G_\theta(\epsilon), \log f^*(m, t) - \log f(m, t) \right\rangle \right].$$

This is the mask-only signal used in Section 3.2. The sign structure gives the update an attraction–repulsion interpretation. This is the same teacher-over-fake view developed in Section 3.3. The teacher term attracts the student toward tokens supported by the pretrained model, while the fake-model term repels tokens that are already easy for a model trained on the current student samples. Therefore, the student is not simply pushed toward the teacher's most likely token. A token is promoted when the teacher assigns it higher log-probability than the fake model does. Once the fake model catches up with the teacher on student samples, this relative signal disappears.

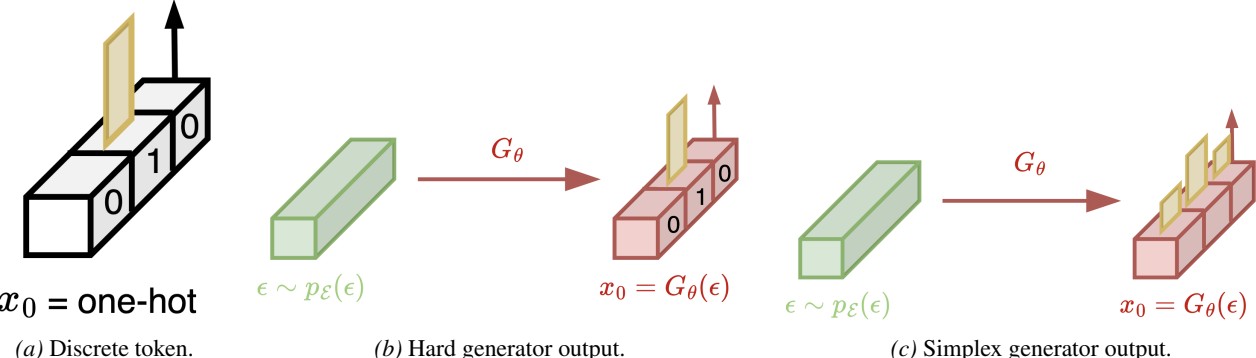

*(a) Discrete token.*  *(b) Hard generator output.*  *(c) Simplex generator output.*

*Figure 9.* Simplex relaxation for IDLM-MDLM, following Section 3.2. A generated token is ultimately interpreted as a one-hot vocabulary choice, but the generator update uses $G_\theta(\epsilon) \in \Delta$ so gradients can weight several candidate tokens before a hard sample is chosen.

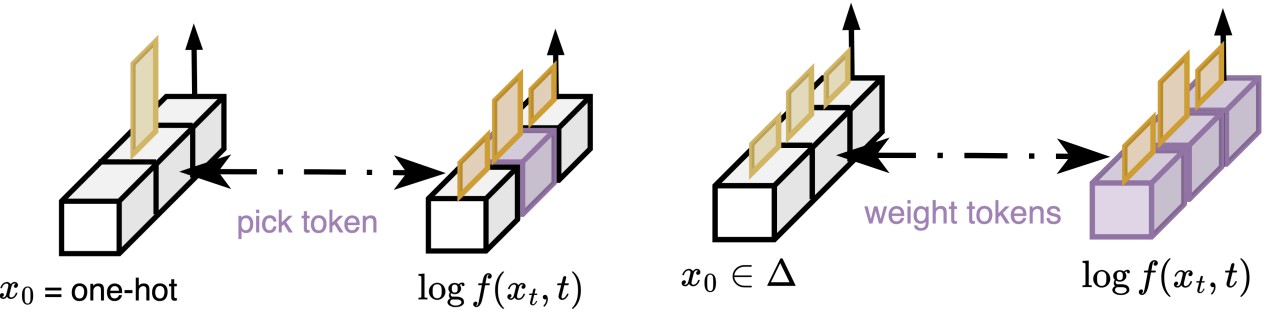

*(a) One-hot target: pick one coordinate.*  *(b) Simplex target: weight token coordinates.*

*Figure 10.* Loss evaluation under the simplex relaxation. The one-hot case selects a single log-probability from $\log f(x_t, t)$, while the simplex case evaluates the weighted sum $\langle G_\theta(\epsilon), \log f(x_t, t) \rangle$.

**Why SUBS is special.** The practical benefit is that the generator does not need to backpropagate through a sampled intermediate token $x_t$. The only generator-dependent object in the update is the differentiable simplex-valued output $G_\theta(\epsilon)$. In DMD-like reductions of inverse distillation, an explicit stop-gradient is usually needed to remove auxiliary gradients through intermediate states (Selikhanovych et al., 2026). For MDLM, the SUBS parameterization effectively handles this cancellation itself: non-mask states are copied and masked states use the fixed input $m$. In this sense, SUBS implicitly applies a stop-gradient to the intermediate state in the generator update. The resulting mask-only expression resembles the token-level distribution-matching loss used by discrete MMD distillation (Hoogeboom et al., 2026). The resemblance is structural rather than exact: D-MMD uses a different posterior for sampling the intermediate state. This is a special property of masked diffusion rather than a generic property of all discrete diffusion processes. In more revisable processes such as uniform diffusion, gradients through the intermediate state can change the training behavior substantially; this is tested in Section 5.1 and Appendix F.4.

**Beyond SUBS.** One can try to pass gradients through masked-diffusion intermediate states using Gumbel or Argmax relaxations, but this introduces a different problem: the teacher MDLM is trained on one-hot or mask-token inputs, whereas these relaxations may feed it simplex-valued states. The resulting teacher signal can therefore be weaker or less stable. This is why the main IDLM-MDLM update uses the mask-only SUBS signal and treats the other relaxations as ablations in Appendix F.4.

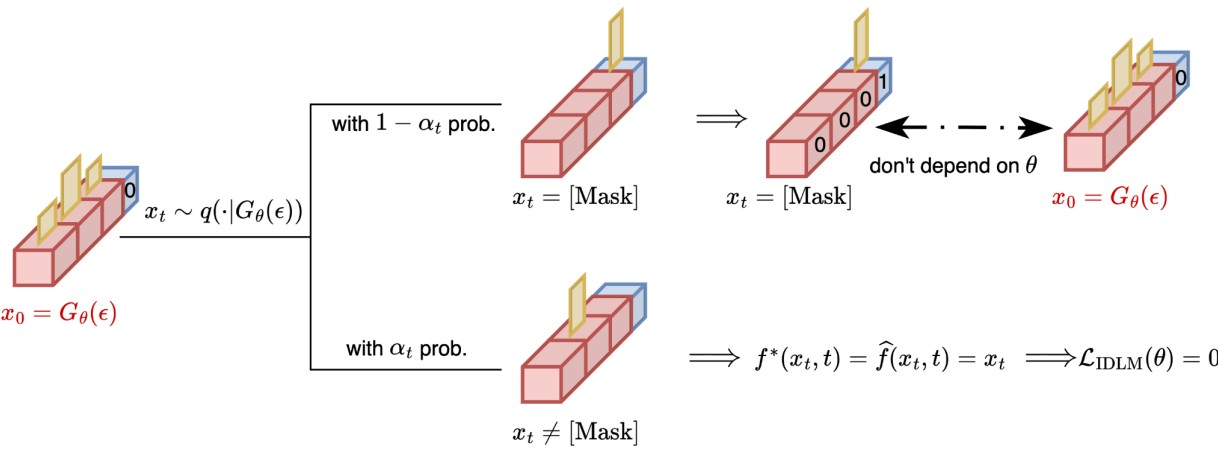

*Figure 11.* Mask-only cancellation under SUBS. The masked forward process produces either an unmasked token with probability $\alpha_t$ or the mask state with probability $1 - \alpha_t$. The unmasked branch is copied by SUBS, so teacher and fake predictions agree and its IDLM contribution cancels. The remaining generator signal comes from the masked branch.

## C. Uniform Diffusion

This appendix gives the process-specific details for uniform diffusion used by UDLM and Duo in Section 2.1.2, and then explains the corresponding IDLM update from Section 3.3.

### C.1. Uniform Diffusion Formulation and Theory

Section 2.1.2 introduces uniform diffusion in compact form. Here we unpack the same process: first the CTMC generator and forward kernel, then the UDLM/Duo objective, the Gaussian relaxation used by Duo, and the reverse sampler. This makes explicit which quantities are reused later by the inverse distillation framework in Appendix C.2.

**Forward process.** Following the general CTMC notation of Appendix A, the uniform process has terminal distribution $\pi = \frac{1}{N}\vec{1}$. Unlike masked diffusion in Appendix B, which only moves tokens toward the mask, the uniform process lets every token transition to every other token at the same rate. Its base matrix is

$$Q_{\text{uni}} = \begin{bmatrix} 1-N & 1 & \cdots & 1 \\ 1 & 1-N & \cdots & 1 \\ \vdots & \vdots & \ddots & \vdots \\ 1 & 1 & \cdots & 1-N \end{bmatrix}. \tag{24}$$

The time-dependent diffusion matrix is

$$Q_t = \sigma_t Q_{\text{uni}} = \begin{bmatrix} \sigma_t(1-N) & \sigma_t & \cdots & \sigma_t \\ \sigma_t & \sigma_t(1-N) & \cdots & \sigma_t \\ \vdots & \vdots & \ddots & \vdots \\ \sigma_t & \sigma_t & \cdots & \sigma_t(1-N) \end{bmatrix}. \tag{25}$$

Its cumulative transition matrix is

$$\exp(\bar{\sigma}_t Q_{\text{uni}}) = \begin{bmatrix} \exp(-N\bar{\sigma}_t) + \frac{1-\exp(-N\bar{\sigma}_t)}{N} & \frac{1-\exp(-N\bar{\sigma}_t)}{N} & \cdots & \frac{1-\exp(-N\bar{\sigma}_t)}{N} \\ \frac{1-\exp(-N\bar{\sigma}_t)}{N} & \exp(-N\bar{\sigma}_t) + \frac{1-\exp(-N\bar{\sigma}_t)}{N} & \cdots & \frac{1-\exp(-N\bar{\sigma}_t)}{N} \\ \vdots & \vdots & \ddots & \vdots \\ \frac{1-\exp(-N\bar{\sigma}_t)}{N} & \frac{1-\exp(-N\bar{\sigma}_t)}{N} & \cdots & \exp(-N\bar{\sigma}_t) + \frac{1-\exp(-N\bar{\sigma}_t)}{N} \end{bmatrix}. \tag{26}$$

With $\alpha_t := \exp(-N\bar{\sigma}_t)$, the conditional distribution used in the main text is therefore

$$q_t(x_t \mid x_0) = \text{Cat}\left(x_t; \alpha_t x_0 + (1-\alpha_t)\frac{1}{N}\vec{1}\right).$$

Thus $x_t$ either keeps the clean token or mixes with the uniform distribution; unlike masked diffusion, a trajectory can revise the same position multiple times.

**UDLM and Duo objective.** UDLM (Schiff et al., 2025) keeps the clean-token predictor $f(x_t, t)$ but uses the uniform noising law. Duo (Sahoo et al., 2025) uses the same discrete process while evaluating the denoiser on a relaxed input. The main text writes their shared training objective through the compact token-level function $g_{\text{Duo}}$ in Eq. (2). Below we write this function explicitly.

Let $q_t(\cdot \mid x)$ denote the uniform-process transition distribution at time $t$. We use $\tilde{x}_t$ for the input passed to the denoiser and $\bar{x}_t$ for the hard intermediate token whose probability is evaluated in the token-level loss. For UDLM these two objects coincide, $\tilde{x}_t = \bar{x}_t = x_t$. For Duo, $\tilde{x}_t$ is the relaxed model input and $\bar{x}_t$ is the hard token induced by that relaxation. For the corresponding model prediction, write

$$f_t := f(\tilde{x}_t, t).$$

For the uniform CTMC above, the scalar weight becomes (von Rütte et al., 2025)

$$\kappa_t(\bar{x}_t, x_0) := \frac{\sigma_t}{q_t(\bar{x}_t \mid x_0)}.$$

The pointwise Itakura–Saito divergence is

$$D_{\mathrm{IS}}(a\|b) := \frac{a}{b} - \log\frac{a}{b} - 1.$$

The integrand used in Eq. (2) is

$$g_{\mathrm{Duo}}(\bar{x}_t, x_0, f_t) := \lambda_t \kappa_t(\bar{x}_t, x_0)\Big[\mathcal{D}_{\mathrm{KL}}\big(q_t(\cdot \mid x_0)\|q_t(\cdot \mid f_t)\big) + D_{\mathrm{IS}}\big(q_t(\bar{x}_t \mid x_0)\|q_t(\bar{x}_t \mid f_t)\big)\Big]. \tag{27}$$

We keep $\lambda_t$ as the optional positive time reweighting used in the main notation. The KL term compares the true and predicted noised distributions. The Itakura–Saito term compares the probability assigned to the currently observed token $\bar{x}_t$, which is the term that later yields the holding-time intuition.

For comparison with the original CTMC notation, the UDLM objective can be written as

$$\mathcal{L}_{\mathrm{UDLM}}(f, x_0) = \int_0^1 \mathbb{E}_{x_t \sim q_t(\cdot|x_0)}\big[g_{\mathrm{UDLM}}\big(x_t, x_0, f(x_t, t)\big)\big]dt, \tag{28}$$

where $g_{\mathrm{UDLM}}$ is the hard-token version of Eq. (27): set $\tilde{x}_t = \bar{x}_t = x_t$ and $f_t = f(x_t, t)$. Here $q_t(\cdot \mid f_t)$ is obtained by applying the linear forward kernel to the predicted clean-token distribution.

**Duo Gaussian relaxation.** Duo exposes a Gaussian relaxation that makes the noised model input differentiable while keeping the same limiting discrete process. Following the notation of Section 3.3, introduce an auxiliary Gaussian noise variable and set

$$\xi \sim \mathcal{N}(0, I), \qquad w_t = \tilde{\alpha}_t x_0 + \sqrt{1 - \tilde{\alpha}_t^2}\,\xi,$$

where $\tilde{\alpha}_t$ is the rescaled Duo schedule. The hard intermediate token is obtained by

$$\bar{x}_t = x_t(w_t) := \arg\max(w_t), \qquad \bar{x}_t \sim q_t(\cdot \mid x_0),$$

where $\arg\max(w_t)$ denotes the one-hot vector at the largest coordinate. Instead of feeding this hard token to the model, Duo uses the soft approximation

$$\tilde{x}_t = x_t^\tau(w_t) = \mathrm{softmax}(w_t/\tau),$$

which gives the relaxed parameterization $f\colon \Delta \times [0, 1] \to \Delta$. With the notation of Eq. (2), Duo evaluates the same KL plus Itakura–Saito divergence objective at $g_{\mathrm{Duo}}(\bar{x}_t, x_0, f(\tilde{x}_t, t))$. As $\tau \to 0^+$, the soft input recovers the hard uniform-process token and the objective recovers the UDLM/NELBO limit.

**Sampling procedure.** Sampling starts from the uniform terminal distribution and repeatedly applies reverse transitions estimated from the clean-token predictor. For $0 \le s < t \le 1$, set $\alpha_{t|s} := \alpha_t/\alpha_s$ and $f_t := f(x_t, t)$. The learned reverse transition used in Section 2.1.2 is

$$p_{s|t}^f(x_s \mid x_t) = \mathrm{Cat}\left(x_s; \frac{N\alpha_t\, x_t \odot f_t + (\alpha_{t|s} - \alpha_t)x_t + (\alpha_s - \alpha_t)f_t + (1 - \alpha_{t|s})(1 - \alpha_s)\vec{1}/N}{N\alpha_t\langle x_t, f_t\rangle + 1 - \alpha_t}\right).$$

This reverse step can either keep the current token or move to another token, so the sampler has a timing decision in addition to a token-choice decision, as illustrated in Figure 12. This is the main structural difference from masked diffusion.

## C.2. Inverse Distillation for Uniform Diffusion

We now connect the uniform diffusion process to the IDLM objective. The ideal inverse-distillation loss compares the pretrained teacher $f^*$ with the best diffusion model that could be trained only on the current student distribution:

$$\mathcal{L}_{\mathrm{IDLM\text{-}Duo}}(\theta) := \mathcal{L}_{\mathrm{Duo}}(f^*, p_\theta) - \min_{\tilde{f}} \mathcal{L}_{\mathrm{Duo}}(\tilde{f}, p_\theta). \tag{29}$$

In practice, as in Section 3.3, we alternate between training the fake model and updating the generator. During the generator step, the fake model $f$ is fixed. Write

$$f_t^* := f^*(\tilde{x}_t, t), \qquad f_t := f(\tilde{x}_t, t)$$

for the teacher and fake predictions at the Duo input. The objective becomes

$$
\begin{aligned}
\mathcal{L}_{\text{IDLM-Duo}}(\theta) &:= \mathcal{L}_{\text{Duo}}(f^*, p_\theta) - \mathcal{L}_{\text{Duo}}(f, p_\theta) \\
&= \mathbb{E}_{\epsilon, t, \xi}\Big[\lambda_t \kappa_t(\bar{x}_t, G_\theta(\epsilon))\Big( \\
&\sum_{y \in \mathcal{V}} q_t(y \mid G_\theta(\epsilon)) \log \frac{q_t(y \mid f_t)}{q_t(y \mid f_t^*)} + q_t(\bar{x}_t \mid G_\theta(\epsilon))\Big(\frac{1}{q_t(\bar{x}_t \mid f_t^*)} - \frac{1}{q_t(\bar{x}_t \mid f_t)}\Big) + \log \frac{q_t(\bar{x}_t \mid f_t^*)}{q_t(\bar{x}_t \mid f_t)}\Big)\Big].
\end{aligned}
$$
(30)

Here $G_\theta(\epsilon)$ is the relaxed student sample, and $(\bar{x}_t, \tilde{x}_t)$ are the hard and relaxed Duo states defined below. This is the same teacher-minus-fake principle as in Appendix B.2, but the uniform process does not collapse to a mask-only term.

**Simplex relaxation.** We use the same simplex relaxation as in Section 3.2 and Appendix B.2. The generator output is allowed to be a token distribution,

$$G_\theta(\epsilon) \in \Delta,$$

which is the same relaxation illustrated in Figure 9. The uniform forward kernel is linear in the clean token, so it remains well-defined on this relaxed output:

$$q_t(\cdot \mid G_\theta(\epsilon)) = \alpha_t G_\theta(\epsilon) + (1 - \alpha_t)\frac{1}{N}\vec{1} \in \Delta.$$

Thus, as in the masked case, the generator can receive gradients through weighted token probabilities instead of through a hard one-hot choice.

**Gaussian reparameterization.** The second issue is the intermediate uniform token. Unlike masked diffusion, uniform diffusion can move from any token to any other token, so the sampled intermediate state cannot be removed by the SUBS cancellation. Duo resolves this by sampling a Gaussian latent variable whose noise is independent of the generator output:

$$\xi \sim \mathcal{N}(0, I), \qquad w_t = \tilde{\alpha}_t G_\theta(\epsilon) + \sqrt{1 - \tilde{\alpha}_t^2}\,\xi.$$

The hard token used in the token-level loss and the relaxed input passed to the denoiser are

$$\bar{x}_t = \arg\max(w_t), \qquad \tilde{x}_t = \text{softmax}(w_t/\tau).$$

Equivalently, the full differentiable transition from the generator output to the relaxed Duo input is

$$G_\theta(\epsilon) \mapsto \tilde{x}_t = \text{softmax}\left(\frac{\tilde{\alpha}_t G_\theta(\epsilon) + \sqrt{1 - \tilde{\alpha}_t^2}\,\xi}{\tau}\right), \qquad \xi \sim \mathcal{N}(0, I).$$

These definitions make the fixed-fake objective in Eq. (30) fully reparameterized and keep the intermediate-state path differentiable.

**Teacher-over-fake advantage.** Following the local-advantage notation of Section 3.3, define

$$a_t := g_{\text{Duo}}(\bar{x}_t, G_\theta(\epsilon), f_t^*) - g_{\text{Duo}}(\bar{x}_t, G_\theta(\epsilon), f_t).$$

Using Eq. (27), and dropping the common positive weight $\lambda_t \kappa_t(\bar{x}_t, G_\theta(\epsilon))$, the local advantage expands to

$$
\begin{aligned}
\big(\lambda_t \kappa_t(\bar{x}_t, G_\theta(\epsilon))\big)^{-1} a_t = &-\Big\langle q_t(\cdot \mid G_\theta(\epsilon)), \log \frac{q_t(\cdot \mid f_t^*)}{q_t(\cdot \mid f_t)}\Big\rangle \\
&+ q_t(\bar{x}_t \mid G_\theta(\epsilon))\Big(\frac{1}{q_t(\bar{x}_t \mid f_t^*)} - \frac{1}{q_t(\bar{x}_t \mid f_t)}\Big) + \log \frac{q_t(\bar{x}_t \mid f_t^*)}{q_t(\bar{x}_t \mid f_t)}.
\end{aligned}
$$

The first term is the distribution-level teacher-over-fake signal at time $t$. The remaining terms compare how much probability the teacher and fake model assign to the current intermediate token $\bar{x}_t$.

**Intuition.** Uniform diffusion can revise the same position more than once. At each reverse step, the sampler decides whether to keep the current token or replace it. The first term in the local advantage is the time-$t$ analogue of the distribution-level attraction–repulsion signal from masked diffusion (Appendix B.2):

$$-\left\langle q_t(\cdot \mid G_\theta(\epsilon)), \log \frac{q_t(\cdot \mid f_t^*)}{q_t(\cdot \mid f_t)} \right\rangle = \mathcal{D}_{\text{KL}}(q_t(\cdot \mid G_\theta(\epsilon))\|q_t(\cdot \mid f_t^*)) - \mathcal{D}_{\text{KL}}(q_t(\cdot \mid G_\theta(\epsilon))\|q_t(\cdot \mid f_t)).$$

Thus, it favors generator samples whose distribution after uniform noising at time $t$ is closer to the teacher than to the fake model. Compared with masked diffusion, this comparison is not made directly on clean-token distributions: the uniform CTMC first mixes each token toward the uniform stationary distribution, and the amount of mixing depends on $t$. The signal is therefore a difference between KL divergences of the time-dependent noised distributions. The holding-time intuition is controlled by the probability assigned to the current intermediate token: larger $q_t(\bar{x}_t \mid f_t^*)$ corresponds to a longer teacher holding time at $\bar{x}_t$, while smaller $q_t(\bar{x}_t \mid f_t^*)$ corresponds to a shorter one. This effect appears directly in the current-token part of Eq. (30):

$$q_t(\bar{x}_t \mid G_\theta(\epsilon)) \left( \frac{1}{q_t(\bar{x}_t \mid f_t^*)} - \frac{1}{q_t(\bar{x}_t \mid f_t)} \right) + \log \frac{q_t(\bar{x}_t \mid f_t^*)}{q_t(\bar{x}_t \mid f_t)}.$$

For a fixed intermediate token, the coefficient of $q_t(\bar{x}_t \mid G_\theta(\epsilon))$ is

$$c_t(\bar{x}_t) := \frac{1}{q_t(\bar{x}_t \mid f_t^*)} - \frac{1}{q_t(\bar{x}_t \mid f_t)}.$$

If $q_t(\bar{x}_t \mid f_t^*)$ is small, then $c_t(\bar{x}_t)$ is large and positive, so minimizing the loss pushes the generator to reduce the probability of clean samples whose noising lands at $\bar{x}_t$. This agrees with the holding-time view: the teacher reverse process has a large exit rate at such an intermediate state and would leave it quickly, so the generator is discouraged from producing samples whose uniform noising often visits that state. If the teacher probability is large while the fake-model probability is small, then $c_t(\bar{x}_t) < 0$. Since it minimizes the IDLM loss, the generator is pushed toward clean samples whose uniform noising visits such teacher-supported states more often; the small fake-model probability strengthens this preference through the negative term $-1/q_t(\bar{x}_t \mid f_t)$. This is the uniform-process analogue of the attraction–repulsion interpretation in Appendix B.2: teacher-supported states attract the generator, while the fake model's low probability makes this signal stronger. In the next alternating step, the fake model is trained on the updated generator distribution, so it adapts to these states and reduces the teacher–fake gap there. Thus, the uniform loss teaches both which token should be chosen and how long the sampler should stay in the current intermediate state.

**DMD-like stop-gradient for uniform diffusion.** The ablation in Section 5.1 and Appendix F.4 compares the full IDLM-Duo update with a stop-gradient variant. In the idealized objective, IDLM matches complete diffusion path measures,

$$\mathcal{L}_{\text{IDLM}}(\theta) = \mathcal{D}_{\text{KL}}(\mathbb{P}^\theta\|\mathbb{P}^*),$$

where $\mathbb{P}^\theta$ and $\mathbb{P}^*$ are the path measures induced by the student and data distributions. For any time $t$, let

$$p_t^\theta(x_t) = \int q_t(x_t \mid x_0) \, p_\theta(dx_0), \qquad p_t^*(x_t) = \int q_t(x_t \mid x_0) \, p^*(dx_0).$$

A DMD-like reduction, as shown by RSD (Selikhanovych et al., 2026), replaces the path-measure objective by noised-marginal matching,

$$\mathcal{L}_{\text{DMD-like}}(\theta) = \mathbb{E}_t\left[\mathcal{D}_{\text{KL}}(p_t^\theta\|p_t^*)\right],$$

and treats the intermediate state as fixed in the generator update. In the Duo parameterization this corresponds to replacing the full relaxed path

$$w_t = \tilde{\alpha}_t G_\theta(\epsilon) + \sqrt{1 - \tilde{\alpha}_t^2}\, \xi$$

by its stop-gradient version

$$w_t^{\text{sg}} = \tilde{\alpha}_t \text{sg}(G_\theta(\epsilon)) + \sqrt{1 - \tilde{\alpha}_t^2}\, \xi.$$

Thus, the DMD-like stop-gradient objective optimizes a marginal matching problem rather than the path-level IDLM-Duo objective. Empirically, this distinction is important: Figure 15 shows that IDLM-Duo with stop-gradient quickly enters a high-entropy, high-GenPPL regime, while the full IDLM-Duo update remains stable.

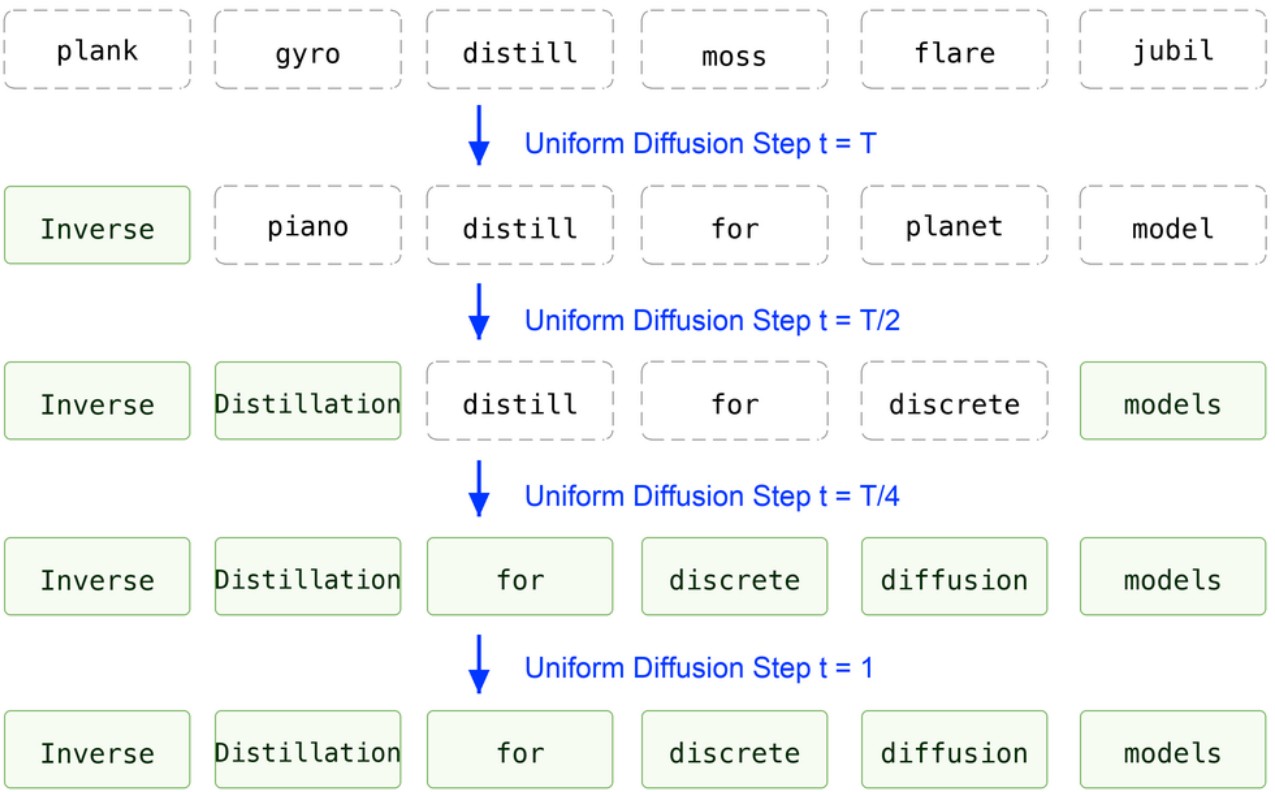

*Figure 12.* Uniform diffusion sampling. Unlike masked diffusion, a position can be revised multiple times, so each step must decide both whether to hold or change the current token and which token should replace it.

## D. Proofs

**Notation and idealized assumptions.** Recall that

$$\mathcal{V} = \{x \in \{0,1\}^N : \sum_{i=1}^{N} x_i = 1\}$$

is the set of one-hot tokens, and let $D([0,T],\mathcal{V})$ be the Skorokhod space of cadlag paths with values in $\mathcal{V}$. For a path measure $\mathbb{Q}$, write $\mathbb{Q}_t$ for its marginal at time $t$.

We assume that the teacher exactly minimizes the expected loss and therefore induces the true reverse path measure, $\mathbb{P}^{f^*} = \mathbb{P}^*$. We also assume that the minimization over $f$ is performed over a sufficiently rich model class, so the exact reverse predictor is available for every clean distribution considered. We assume that the student path does not use transitions that are impossible under the teacher path. If $\mathbb{P}^\theta \not\ll \mathbb{P}^*$, then $\mathcal{D}_{\mathrm{KL}}(\mathbb{P}^\theta \| \mathbb{P}^*) = +\infty$, so the main theorem becomes immediate.

**Time convention.** For a clean distribution $p$, let $\mathbb{P}^p$ denote the path measure associated with the fixed forward noising process, written in reverse-time coordinates so that the clean sample is the terminal marginal of $\mathbb{P}^p$. Its forward-time noised marginal at noise level $t$ is

$$p_t(x_t) = \int q_t(x_t \mid x_0) \, p(dx_0).$$

We write $\mathbb{P}^\theta = \mathbb{P}^{p_\theta}$ and $\mathbb{P}^* = \mathbb{P}^{p^*}$. Since time reversal is a deterministic reparameterization of paths, the KL divergence is unchanged by choosing either coordinate convention.

### D.1. Connection with the main text

DLM training can be viewed as fitting the reverse-time path measure associated with a fixed forward noising process. Following the main text, we write this using expected losses of the form $\mathcal{L}(f,p)$, where $p$ is a clean-sample distribution. For SEDD, Dynkin's formula (Hanson, 2007; Campbell et al., 2022) gives the following distribution-level objective, up to terms independent of the optimized model:

$$\mathcal{L}_{\mathrm{SEDD}}(f,p) = \mathbb{E}_{p(x_0),t,q_t} \sum_{y \neq x_t} \langle x_t, y \rangle_{Q_t} \big( \langle f(x_t,t), y \rangle - \langle s(x_t, t \mid x_0), y \rangle \log \langle f(x_t,t), y \rangle \big). \tag{31}$$

This is the SEDD instance of the unified loss $\mathcal{L}(f,p)$ used in Equation (12). For a generic clean distribution $p$, define

$$f^p = \arg\min_f \mathcal{L}_{\mathrm{SEDD}}(f,p).$$

At this minimizer,

$$\langle f^p(x_t,t), y \rangle = \mathbb{E}_{p(x_0|x_t)} \langle s(x_t, t \mid x_0), y \rangle. \tag{32}$$

In particular, $f^* = f^{p^*}$ denotes the exact teacher and $f^\theta = f^{p_\theta}$ denotes the exact fake model for the student distribution.

### D.2. Proof of the theorem

We prove the theorem in three steps. First, we show that the path KL equals the SEDD inverse-distillation gap. Second, we transfer the result to MDLM and the uniform objective, with Duo covered in the hard-token limit. Third, we use the chain rule to show that the path KL controls the clean-data KL.

**Lemma D.1** (SEDD path KL equals the inverse gap). *Under the population assumptions above,*

$$\mathcal{D}_{\mathrm{KL}}(\mathbb{P}^\theta \| \mathbb{P}^*) = \mathcal{L}_{\mathrm{SEDD}}(f^*, p_\theta) - \min_f \mathcal{L}_{\mathrm{SEDD}}(f, p_\theta). \tag{33}$$

*Proof.* The relative entropy between two continuous-time jump-process path measures can be written as an integral over the relative jump-rate divergence. In the SEDD parameterization this gives

$$\mathcal{D}_{\mathrm{KL}}(\mathbb{P}^\theta \| \mathbb{P}^*) = \int_0^T \mathbb{E}_{p_t^\theta(x_t)} \sum_{y \neq x_t} \langle x_t, y \rangle_{Q_t} \langle f^*(x_t,t), y \rangle \phi \left( \frac{\langle f^\theta(x_t,t), y \rangle}{\langle f^*(x_t,t), y \rangle} \right) dt, \tag{34}$$

where $\phi(r) = r \log r - r + 1$. Following the linearization trick of Kornilov et al. (2026), for the positive ratios used below,

$$\phi(r) = \max_l \{rl + 1 - \exp(l)\}.$$

Using the parameterization

$$l(y, x_t, t) = \log \frac{\langle f(x_t, t), y \rangle}{\langle f^*(x_t, t), y \rangle},$$

and taking the maximum over admissible $f$, Equation (34) becomes

$$\mathcal{D}_{\mathrm{KL}}(\mathbb{P}^\theta \| \mathbb{P}^*) = \max_f \int_0^T \mathbb{E}_{p_t^\theta(x_t)} \sum_{y \neq x_t} \langle x_t, y \rangle_{Q_t} \left[ \langle f^\theta(x_t, t), y \rangle \log \frac{\langle f(x_t, t), y \rangle}{\langle f^*(x_t, t), y \rangle} \right. \tag{35}$$
$$\left. + \langle f^*(x_t, t) - f(x_t, t), y \rangle \right] dt.$$

We now rewrite the expectation in Equation (35). First, apply the posterior identity (32) with $p = p_\theta$:

$$\langle f^\theta(x_t, t), y \rangle = \mathbb{E}_{p_\theta(x_0 | x_t)} \langle s(x_t, t \mid x_0), y \rangle.$$

After substituting this identity, the only term depending on $x_0$ is $\langle s(x_t, t \mid x_0), y \rangle$. The remaining term $\langle f^*(x_t, t) - f(x_t, t), y \rangle$ depends only on $x_t$, $t$, and $y$, so it can be placed inside the same conditional expectation:

$$\mathbb{E}_{p_t^\theta(x_t)} \sum_{y \neq x_t} \langle x_t, y \rangle_{Q_t} \left[ \langle f^\theta(x_t, t), y \rangle \log \frac{\langle f(x_t, t), y \rangle}{\langle f^*(x_t, t), y \rangle} + \langle f^*(x_t, t) - f(x_t, t), y \rangle \right]$$
$$= \mathbb{E}_{p_t^\theta(x_t)} \mathbb{E}_{p_\theta(x_0 | x_t)} \sum_{y \neq x_t} \langle x_t, y \rangle_{Q_t} \left[ \langle s(x_t, t \mid x_0), y \rangle \log \frac{\langle f(x_t, t), y \rangle}{\langle f^*(x_t, t), y \rangle} + \langle f^*(x_t, t) - f(x_t, t), y \rangle \right].$$

Finally, the joint law of $(x_0, x_t)$ can be written in two equivalent ways:

$$p_t^\theta(x_t) p_\theta(x_0 \mid x_t) = p_\theta(x_0) q_t(x_t \mid x_0),$$

so

$$\mathbb{E}_{p_t^\theta(x_t)} \mathbb{E}_{p_\theta(x_0 | x_t)}[\cdot] = \mathbb{E}_{p_\theta(x_0), \, x_t \sim q_t(\cdot | x_0)}[\cdot].$$

The maximization over $f$ stays outside this change of variables. Therefore we obtain

$$\mathcal{D}_{\mathrm{KL}}(\mathbb{P}^\theta \| \mathbb{P}^*) = \max_f \mathbb{E}_{p_\theta(x_0), t, q_t} \sum_{y \neq x_t} \langle x_t, y \rangle_{Q_t} \left[ \langle s(x_t, t \mid x_0), y \rangle \log \frac{\langle f(x_t, t), y \rangle}{\langle f^*(x_t, t), y \rangle} \right. \tag{36}$$
$$\left. + \langle f^*(x_t, t) - f(x_t, t), y \rangle \right].$$

The right-hand side is exactly the population SEDD loss difference:

$$\max_f \left\{ \mathcal{L}_{\mathrm{SEDD}}(f^*, p_\theta) - \mathcal{L}_{\mathrm{SEDD}}(f, p_\theta) \right\}.$$

Because the first term is independent of $f$, this equals

$$\mathcal{L}_{\mathrm{SEDD}}(f^*, p_\theta) - \min_f \mathcal{L}_{\mathrm{SEDD}}(f, p_\theta).$$

This proves the claim. Importantly, the maximum remains outside the population expectation throughout the argument. $\square$

**Lemma D.2** (Gap preservation under equivalent losses). *Suppose two losses satisfy*

$$\mathcal{L}_A(Tf, p) = \mathcal{L}_B(f, p) + C(p),$$

*where $C(p)$ is independent of the optimized model $f$, and $T$ maps the candidate class for $\mathcal{L}_B$ onto the corresponding candidate class for $\mathcal{L}_A$. Then their inverse gaps agree:*

$$\mathcal{L}_A(Tf^*, p) - \min_g \mathcal{L}_A(g, p) = \mathcal{L}_B(f^*, p) - \min_f \mathcal{L}_B(f, p).$$

*Proof.* The claim follows by cancellation of the model-independent constant:

$$\mathcal{L}_A(Tf^*, p) - \min_g \mathcal{L}_A(g, p) = \mathcal{L}_B(f^*, p) + C(p) - \min_f\{\mathcal{L}_B(f, p) + C(p)\}$$
$$= \mathcal{L}_B(f^*, p) - \min_f \mathcal{L}_B(f, p).$$

$\square$

**MDLM, UDLM, and Duo.** For the absorbing diffusion matrix $Q_{\mathrm{abs}}$ in Equation (22), the SEDD objective under the clean-token parameterization is equivalent to the MDLM loss $\mathcal{L}_{\mathrm{MDLM}}$ up to terms independent of the optimized predictor. Lemma D.2 therefore gives

$$\mathcal{D}_{\mathrm{KL}}(\mathbb{P}^\theta \| \mathbb{P}^*) = \mathcal{L}_{\mathrm{MDLM}}(f^*, p_\theta) - \min_f \mathcal{L}_{\mathrm{MDLM}}(f, p_\theta).$$

For the uniform diffusion matrix $Q_{\mathrm{uni}}$ in Equation (25), the same argument transfers the SEDD gap to the hard-token UDLM objective. In distribution-level notation,

$$\mathcal{L}_{\mathrm{UDLM}}(f, p) := \mathbb{E}_{p(x_0), t, x_t \sim q_t(\cdot | x_0)}\big[g_{\mathrm{UDLM}}\big(x_t, x_0, f(x_t, t)\big)\big].$$

Thus

$$\mathcal{D}_{\mathrm{KL}}(\mathbb{P}^\theta \| \mathbb{P}^*) = \mathcal{L}_{\mathrm{UDLM}}(f^*, p_\theta) - \min_f \mathcal{L}_{\mathrm{UDLM}}(f, p_\theta).$$

Duo evaluates the same uniform-process objective on a Gaussian-relaxed input and recovers the hard-token UDLM objective as $\tau \to 0^+$. Hence the theorem applies to Duo in this hard-token limit, as stated in the main text. This argument does not claim exact uniqueness for finite-temperature Duo.

**Theorem D.3** (Unique solution). *For SEDD, MDLM, and hard-token UDLM, and for Duo in the limit $\tau \to 0^+$, the IDLM loss satisfies*
$$\mathcal{L}_{\mathrm{IDLM}}(\theta) = \mathcal{D}_{\mathrm{KL}}(\mathbb{P}^\theta \| \mathbb{P}^*) \geq \mathcal{D}_{\mathrm{KL}}(p_\theta \| p^*) \geq 0.$$

*Moreover,*
$$\mathcal{L}_{\mathrm{IDLM}}(\theta) = 0 \quad \Longleftrightarrow \quad p_\theta = p^*.$$

*Proof.* Disintegrate the path measures with respect to their clean-sample marginal:

$$\mathbb{P}^\theta(dx, dw) = p_\theta(dx)\mathbb{P}_x^\theta(dw), \qquad \mathbb{P}^*(dx, dw) = p^*(dx)\mathbb{P}_x^*(dw).$$

The chain rule for relative entropy gives

$$\mathcal{D}_{\mathrm{KL}}(\mathbb{P}^\theta \| \mathbb{P}^*) = \mathcal{D}_{\mathrm{KL}}(p_\theta \| p^*) + \mathbb{E}_{x \sim p_\theta} \mathcal{D}_{\mathrm{KL}}(\mathbb{P}_x^\theta \| \mathbb{P}_x^*).$$

The conditional KL term is nonnegative, so

$$\mathcal{D}_{\mathrm{KL}}(\mathbb{P}^\theta \| \mathbb{P}^*) \geq \mathcal{D}_{\mathrm{KL}}(p_\theta \| p^*) \geq 0.$$

Now consider the zero-loss condition. From the previous part of the proof, $\mathcal{L}_{\mathrm{IDLM}}(\theta) = \mathcal{D}_{\mathrm{KL}}(\mathbb{P}^\theta \| \mathbb{P}^*)$. Therefore, if $\mathcal{L}_{\mathrm{IDLM}}(\theta) = 0$, then

$$\mathcal{D}_{\mathrm{KL}}(\mathbb{P}^\theta \| \mathbb{P}^*) = 0.$$

Substituting this into the chain-rule decomposition gives

$$0 = \mathcal{D}_{\mathrm{KL}}(p_\theta \| p^*) + \mathbb{E}_{x \sim p_\theta} \mathcal{D}_{\mathrm{KL}}(\mathbb{P}_x^\theta \| \mathbb{P}_x^*).$$

Both terms on the right are nonnegative, so each term must be zero. In particular,

$$\mathcal{D}_{\mathrm{KL}}(p_\theta \| p^*) = 0.$$

The KL divergence between two probability distributions is zero only when the two distributions are equal, hence $p_\theta = p^*$.

Conversely, assume $p_\theta = p^*$. The forward noising process is fixed: once the clean distribution $p$ is chosen, the whole path measure $\mathbb{P}^p$ is obtained by applying the same noising dynamics to samples from $p$. Therefore, if the clean distributions are equal, the induced path measures are also equal:

$$\mathbb{P}^\theta = \mathbb{P}^{p_\theta} = \mathbb{P}^{p^*} = \mathbb{P}^*.$$

Thus

$$\mathcal{D}_{\mathrm{KL}}(\mathbb{P}^\theta \| \mathbb{P}^*) = 0,$$

and using again $\mathcal{L}_{\mathrm{IDLM}}(\theta) = \mathcal{D}_{\mathrm{KL}}(\mathbb{P}^\theta \| \mathbb{P}^*)$, we get $\mathcal{L}_{\mathrm{IDLM}}(\theta) = 0$. □

**Sequence-level extension.** The proof above is for one token. For a full sequence, the same proof works in an ideal setting where the whole sequence is treated as one state and the models describe the full reverse process over sequences. In that case, the IDLM loss is the KL divergence between the full sequence path measures, so it also controls the difference between $p_\theta(x_0^{1:L})$ and $p^*(x_0^{1:L})$. This is not exactly the loss used in experiments. In practice, we sum token-level losses, which is convenient for training but does not by itself give the full sequence path KL, because tokens may still depend on each other. Thus, the proof covers the one-token objective and the ideal full-sequence objective, but not the practical factorized sequence loss. We leave a formal theoretical analysis of the factorized sequence-level IDLM objective as a promising direction for future research.

# E. Experimental Details.

## E.1. Unconditional Generation on OpenWebText

**Codebase**. Our implementation builds upon the original Duo repository (Sahoo et al., 2025) and the SEDD repository (Lou et al., 2024), which serve as the primary codebases for our experiments. We extend these repositories to incorporate our proposed training framework, with the main algorithmic details provided in Algorithm 1.

**Teacher checkpoints**. We use pretrained checkpoints for the distillation of MDLM and Duo models obtained from the Duo repository (Sahoo et al., 2025), and a pretrained checkpoint for SEDD distillation from the SEDD repository (Lou et al., 2024). Unfortunately, for SEDD, only a checkpoint corresponding to the absorbing process was available.

**Model architecture**. We follow (Sahoo et al., 2025) for distilling MDLM and Duo, and design the denoising model to operate on both continuous and discrete latents. We use a Transformer (Vaswani et al., 2017) in which the first-layer token representations are computed via a matrix multiplication with the embedding matrix. For discrete inputs, we perform standard embedding lookups. In contrast, continuous inputs correspond to "soft lookups," producing a convex combination of vocabulary embeddings. For SEDD distillation, we follow (Lou et al., 2024) and use a model that only accepts one-hot vectors. For all setups we use small models with 169M parameters, specifically with 12 layers, 12 heads, hidden size 768, sequence length 1024, and dropout 0.1.

**Training hyperparameters**. Across all experimental settings, we employed a per-device batch size of 8, resulting in a global batch size of 512. Optimization was performed using the AdamW optimizer (Loshchilov & Hutter, 2019), with a fixed learning rate of $1 \times 10^{-6}$ for MDLM and Duo, and $3 \times 10^{-7}$ for SEDD. A constant learning rate schedule was used with 2500 warmup steps for all configurations. We adopted the log-linear noise formulation from (Lou et al., 2024) in the loss function, and applied exponential moving average (EMA) with a decay rate of 0.9999 throughout training.

**Evaluation protocol**. As observed by (Zheng et al., 2025), the GenPPL metric is sensitive to floating-point precision. To ensure consistency and numerical stability, we follow the protocol of (Sahoo et al., 2025) and perform all sampling experiments using `float64` precision. All evaluation metrics are computed using the official codebase provided in the Duo (Sahoo et al., 2025) repository.

**Metric definitions**. Let $x_0^{1:L} \sim p_\theta$ denote generated samples and let $p^*$ denote the validation-data distribution. For a finite generated set $\mathcal{S}_\theta$, generative perplexity (GenPPL) is computed by scoring each generated sequence with a fixed GPT-2 Large autoregressive evaluator $p_{\mathrm{AR}}$ (Radford et al., 2019):

$$\mathrm{GenPPL} = \exp\left(-\frac{1}{M} \sum_{x_0^{1:L} \in \mathcal{S}_\theta} \sum_{i=1}^{L} \log p_{\mathrm{AR}}(x_0^i \mid x_0^{1:i-1})\right),$$

where $M$ is the total number of evaluated tokens. Lower GenPPL means that the evaluator assigns higher likelihood to generated samples, and we report it together with diversity metrics to avoid favoring overly repetitive text. Entropy is the empirical token entropy of generated samples:

$$\mathrm{Entropy} = -\sum_{v \in \mathcal{V}} \hat{p}_\theta(v) \log \hat{p}_\theta(v),$$

where $\hat{p}_\theta(v)$ is the generated-token frequency of token $v \in \mathcal{V}$; higher entropy means greater token-level diversity. MAUVE (Lou et al., 2024) compares samples from $p_\theta$ with validation samples from $p^*$ in language-model embedding space, higher values mean that the two distributions are closer. Finally, GM denotes the GPT-2 Gradient Moment metric (Hoogeboom et al., 2026):

$$\mathrm{GM}(p_\theta, p^*) = \left\| \mathbb{E}_{x_0^{1:L} \sim p_\theta} \nabla_\psi \log p_{\mathrm{AR},\psi}(x_0^{1:L}) - \mathbb{E}_{x_0^{1:L} \sim p^*} \nabla_\psi \log p_{\mathrm{AR},\psi}(x_0^{1:L}) \right\|_2^2,$$

estimated with independent minibatches. Lower GM means that generated samples induce reference-model gradients closer to those induced by validation data.

**Dataset preparation**. Following prior work (Sahoo et al., 2024; 2025; Lou et al., 2024), we preprocess the One Billion Words dataset using the detokenization procedure described by Lou et al. (2024) and Sahoo et al. (2024), with the official implementation available at this link. For the OpenWebText dataset, we utilize the GPT2 tokenizer and apply a similar

concatenation and wrapping procedure as in the previous works (Sahoo et al., 2024; 2025), targeting a sequence length of 1,024 tokens. During wrapping, eos tokens are inserted between consecutive sequences. As OpenWebText does not include an official validation split, we reserve the final 100,000 documents from the dataset as a validation set.

**Other baselines**. Our primary baselines are SDTT (Deschenaux & Gulcehre, 2025) and Duo-DCD (Sahoo et al., 2025). Performance metrics for both methods are reported using values provided in the Duo (Sahoo et al., 2025), corresponding to the best-performing results obtained after 5 rounds of distillation, described in (Sahoo et al., 2025). While a valid pretrained checkpoint was available for Duo-DCD, we were unable to identify a usable checkpoint for SDTT (Deschenaux & Gulcehre, 2025), which precluded a fair comparison or distillation of this model within our experimental framework.

### E.2. Conditional Generation on TinyGSM

**Dataset preparation**. We use TinyGSM for training and validation and tokenize examples with the HuggingFaceTB/SmolLM-135M tokenizer. Following the conditional-generation setup, examples are kept at sequence length 512 without document wrapping. We insert eos tokens, use a newline separator between the problem and solution fields, filter examples that exceed the maximum length, and reserve $1\%$ of the data for validation with seed 42. The problem prompt is treated as conditioning context and is excluded from the training loss, while padding tokens are included.

**Model and teacher checkpoint**. We use the same small denoising Transformer architecture as in the OpenWebText experiments: 12 layers, 12 attention heads, hidden size 768, sequence length 512, and dropout 0.1. IDLM-MDLM uses the MDLM SUBS parameterization with log-linear noise and is initialized from the TinyGSM MDLM teacher checkpoint. IDLM-Duo uses uniform diffusion with the mean parameterization, time conditioning, and adaptive layer normalization; the student is initialized from the TinyGSM Duo teacher, using the teacher EMA weights.

**Training hyperparameters**. Both IDLM-MDLM and IDLM-Duo are trained with adversarial distillation enabled at every update. We use a global batch size of 512, 4 GPUs, DDP training, and AdamW with learning rate $1 \times 10^{-6}$, weight decay 0, $\beta_1 = 0.9$, $\beta_2 = 0.999$, and $\epsilon = 10^{-8}$. IDLM-MDLM uses per-device batch size 32 and bfloat16 precision, while IDLM-Duo uses per-device batch size 16 and float32 precision. We train for 250,000 steps with a constant learning-rate schedule after 2,500 warmup steps, use antithetic time sampling, and maintain an EMA copy of the model with decay 0.9999.

**Evaluation protocol**. We follow the conditional-generation protocol of Deschenaux & Gulcehre (2026). Each GSM8K evaluation instance is formatted as a problem prompt followed by a solution region. During sampling, the problem prompt is clamped and the model generates only the solution region. We evaluate IDLM-MDLM and IDLM-Duo with 32, 64, and 128 reverse steps and compare them against the corresponding 1024-step teachers. IDLM-MDLM uses the cached ancestral predictor, IDLM-Duo uses the ancestral sampler, and both use float64 precision at sampling time.

**Metric**. For each problem, we generate one completion and use the execution-based TinyGSM/GSM8K scorer to extract the predicted numerical answer. We report exact-match accuracy against the GSM8K ground-truth answer in Table 1.

*Table 2.* **Duo[a] distillation comparison.** We report GenPPL ↓, MAUVE ↑, and Entropy ↑. Methods are grouped by Steps, with the best metrics bolded in each group.

| | Steps | GenPPL ↓ | MAUVE ↑ | Entropy ↑ |
|---|---|---|---|---|
| FLM (Lee et al., 2026) | 1024 | **62.23** | – | 5.33 |
| Duo[a] (Teacher) (Sahoo et al., 2025) | | 77.69 | **0.96** ± 0.01 | **5.55** |
| ELF (Hu et al., 2026) | | **24.08** ± 0.16 | – | 5.14 |
| FMLM (Lee et al., 2026) | 32 | 45.09 | – | 5.25 |
| Duo-DCD[a] (Sahoo et al., 2025) | | 61.31 | **0.97** ± 0.01 | **5.52** |
| IDLM-DCD[a] (**Ours**) | | 42.13 ± 3.65 | 0.94 ± 0.01 | 5.41 ± 0.05 |
| ELF (Hu et al., 2026) | | **33.66** ± 1.09 | – | 5.16 |
| FMLM (Lee et al., 2026) | 16 | 63.63 | – | 5.29 |
| Duo-DCD[a] (Sahoo et al., 2025) | | 75.24 | **0.96** ± 0.01 | **5.53** |
| IDLM-DCD[a] (**Ours**) | | 52.06 ± 4.23 | 0.95 ± 0.01 | 5.44 ± 0.05 |
| ELF (Hu et al., 2026) | | 67.32 ± 2.25 | – | 5.14 |
| FMLM (Lee et al., 2026) | 8 | 86.5 | – | 5.36 |
| Duo-DCD[a] (Sahoo et al., 2025) | | 111.88 | 0.94 ± 0.01 | **5.52** |
| IDLM-DCD[a] (**Ours**) | | **66.41** ± 4.59 | **0.95** ± 0.01 | 5.42 ± 0.03 |
| FMLM (Lee et al., 2026) | | 111.31 | – | 5.26 |
| Duo-DCD[a] (Sahoo et al., 2025) | 4 | 261.82 | 0.79 ± 0.02 | **5.50** |
| IDLM-DCD[a] (**Ours**) | | **111.29** ± 8.39 | **0.88** ± 0.02 | 5.32 ± 0.06 |

## F. Additional Experimental Results

### F.1. Ancestral Sampling Results

Table 2 reports the detailed Duo[a] results corresponding to the main OWT comparison. As mentioned above, Figure 13 shows the corresponding GenPPL and entropy curves together with the other step-scaling results. Under ancestral sampling, IDLM-DCD remains strongest in the aggressive low-step regime. In particular, the 8-step sampler gives the clearest speed-quality tradeoff: it stays close to the 1024-step Duo teacher while reducing the reverse-step budget by $128\times$.

The main comparison is with Duo-DCD under the same ancestral sampler. At 8 steps, IDLM-DCD improves GenPPL from 111.88 to 66.41 while keeping MAUVE and entropy close to the teacher. At 4 steps, Duo-DCD degrades sharply, whereas IDLM-DCD still preserves a usable quality-diversity tradeoff. Thus the benefit of IDLM is largest when very few reverse steps must replace the original 1024-step teacher trajectory.

### F.2. GenPPL and Entropy vs. Sampling Steps

Figure 13 provides the main GenPPL and entropy curves that complement the compact OWT summary in Figure 3. These plots show how generation quality and token-level diversity change as the number of reverse sampling steps is reduced. For MDLM, IDLM-MDLM gives the strongest low-step tradeoff relative to the teacher and SDTT baselines, improving GenPPL while maintaining comparable entropy. For Duo[g], IDLM-DCD is most effective in the aggressive low-step regime, reaching strong GenPPL at 4 steps while preserving competitive entropy. We also include the ancestral Duo curve here, so the greedy and ancestral step trends can be compared in one place.

### F.3. Scaling to Larger MDLM

To test whether IDLM's low-step behavior persists beyond the 169M setting, we distill an 863M MDLM (Sahoo et al., 2024) checkpoint from the SDTT (Deschenaux & Gulcehre, 2025) work. This is the most direct larger-scale comparison available because SDTT uses the same model family and provides matched low-step baselines. As shown in Table 3, IDLM-MDLM consistently improves over SDTT at 32, 16, and 8 sampling steps, improving both GenPPL and entropy in the low-step regime. In particular, the 16-step sampler preserves the $64\times$ reduction from the 1024-step teacher, indicating that the acceleration behavior observed in the 169M experiments also extends to the larger-scale regime.

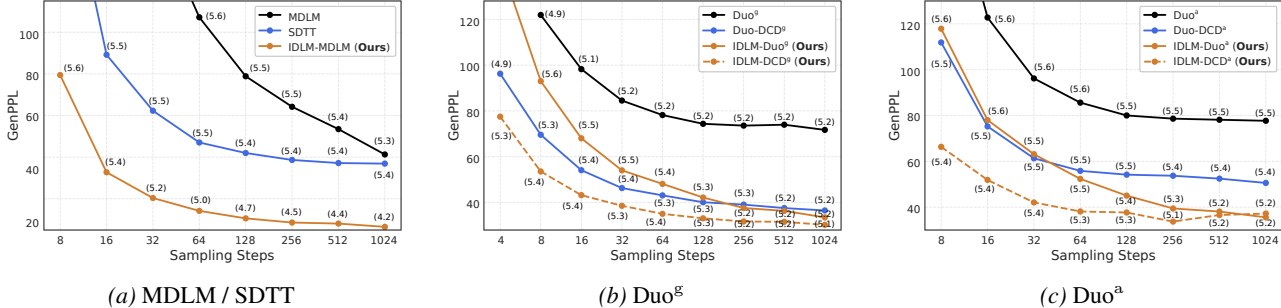

*(a) MDLM / SDTT*      *(b) Duo^g*      *(c) Duo^a*

*Figure 13.* **OWT GenPPL and entropy vs. sampling steps.** Each panel shows how generation quality and diversity change as the reverse sampling budget is reduced. GenPPL measures sample quality, where lower is better, while entropy tracks output diversity, where higher is better. The panels provide a unified comparison of step-scaling behavior for our IDLM-MDLM, IDLM-DCD^g, and IDLM-DCD^a models.

*Table 3.* **Scaling to an 863M MDLM teacher.** We report GenPPL ↓ and Entropy ↑.

| Method (863M params) | Steps | GenPPL ↓ | Entropy ↑ |
|---|---|---|---|
| MDLM (Teacher) (Sahoo et al., 2024) | 1024 | 34.99 | 5.24 |
| SDTT (Deschenaux & Gulcehre, 2025) | 32 | 31.16 | 5.12 |
| IDLM-MDLM **(Ours)** | | **13.71** $\pm$ 1.70 | **5.17** $\pm$ 0.10 |
| SDTT (Deschenaux & Gulcehre, 2025) | 16 | 52.13 | 5.20 |
| IDLM-MDLM **(Ours)** | | **24.07** $\pm$ 2.13 | **5.36** $\pm$ 0.04 |
| SDTT (Deschenaux & Gulcehre, 2025) | 8 | 121.51 | 5.25 |
| IDLM-MDLM **(Ours)** | | **67.37** $\pm$ 4.11 | **5.59** $\pm$ 0.03 |

## F.4. Ablation Study

This subsection expands the ablations summarized in Section 5.1. We keep the teacher and evaluation setup fixed, so the comparisons isolate how gradients are passed to the generator and how much latent stochasticity is available during masked-diffusion sampling.

**Beyond the SUBS parameterization.** In masked diffusion, the SUBS parameterization makes the IDLM-MDLM generator update mask-only: unmasked positions are copied by both the teacher and fake model, so their contribution cancels. As discussed in Section 3.2, this removes the sampled intermediate token from the generator gradient and behaves like an implicit stop-gradient through the absorbing transition. We therefore test whether explicitly differentiating through the intermediate state improves the update. Let $x_0 = G_\theta(\epsilon)$ and let $m$ denote the mask token. The Gumbel relaxation softens the full categorical draw from $q_t(\cdot \mid x_0)$, replacing the hard token by

$$\tilde{x}_t^{\mathrm{G}} = \mathrm{softmax}\left(\frac{\log q_t(\cdot \mid G_\theta(\epsilon)) + g}{\tau}\right), \qquad g_i \overset{\mathrm{i.i.d.}}{\sim} \mathrm{Gumbel}(0,1).$$

Its objective is

$$\mathcal{L}_{\mathrm{Gumbel}}(\theta) = -\mathbb{E}_{t,\epsilon,g}\left\langle G_\theta(\epsilon), \log f^*(\tilde{x}_t^{\mathrm{G}}, t) - \log \hat{f}(\tilde{x}_t^{\mathrm{G}}, t)\right\rangle.$$

The Argmax relaxation keeps the forward state closer to the hard masked process by using the Gumbel-max form of $q_t(x_t \mid x_0) = \alpha_t \delta_{x_0}(x_t) + (1 - \alpha_t)\delta_m(x_t)$:

$$b_{\mathrm{copy}} = \mathbf{1}\{\log \alpha_t + g_1 \geq \log(1 - \alpha_t) + g_2\}, \qquad b_{\mathrm{mask}} = 1 - b_{\mathrm{copy}},$$

$$\tilde{x}_t^{\mathrm{A}} = b_{\mathrm{copy}} G_\theta(\epsilon) + b_{\mathrm{mask}} m, \qquad g_1, g_2 \overset{\mathrm{i.i.d.}}{\sim} \mathrm{Gumbel}(0,1).$$

The corresponding objective is

$$\mathcal{L}_{\mathrm{Argmax}}(\theta) = -\mathbb{E}_{t,\epsilon,g_1,g_2}\left\langle G_\theta(\epsilon), \log f^*(\tilde{x}_t^{\mathrm{A}}, t) - \log \hat{f}(\tilde{x}_t^{\mathrm{A}}, t)\right\rangle.$$

Figure 14 shows that this additional gradient path does not improve the masked-diffusion tradeoff. The SUBS update reaches the useful low-GenPPL regime fastest while preserving substantially more entropy than Argmax. Gumbel is smoother, but it reduces GenPPL more slowly and gives a weaker quality–diversity profile near the shared training budget. This supports the main-text hypothesis that feeding soft or branch-switched intermediate states to an MDLM teacher can move the evaluation outside the one-hot training domain and weaken the distillation signal.

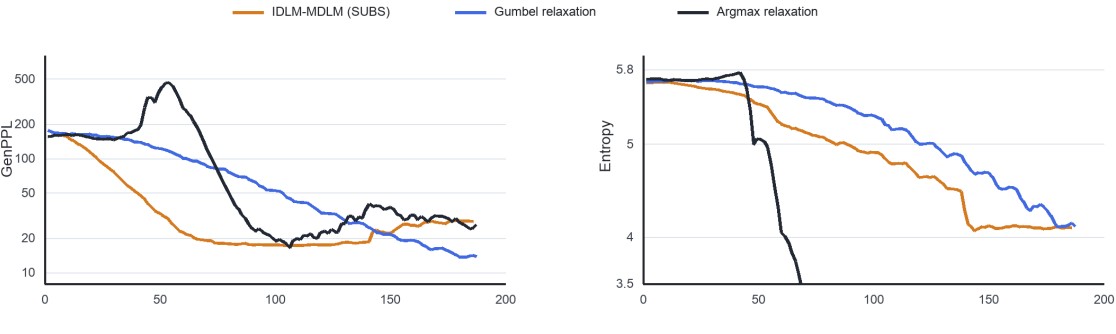

Training steps (k)

*Figure 14.* **Masked-diffusion parameterization ablation.** Validation GenPPL and entropy for IDLM-MDLM with the SUBS, Gumbel, and Argmax generator updates. The SUBS update gives the strongest early quality–diversity tradeoff under the shared training budget.

**DMD-like uniform distillation.** For uniform diffusion, the Duo parameterization evaluates denoisers on relaxed states, so the intermediate state remains differentiable. This makes the difference between full inverse distillation and a DMD-like stop-gradient reduction directly testable. Let

$$a_t(w, x) = g_{\mathrm{Duo}}\big(x_t(w), x, f^*(x_t^\tau(w), t)\big)$$
$$- g_{\mathrm{Duo}}\big(x_t(w), x, f(x_t^\tau(w), t)\big),$$

where $x_t(w) = \arg\max(w)$ and $x_t^\tau(w) = \mathrm{softmax}(w/\tau)$ are the hard and relaxed Duo states. The full IDLM-Duo objective and the DMD-like stop-gradient objective use the same teacher–fake Duo advantage, but differ in whether the relaxed intermediate state carries gradients:

$$
\begin{aligned}
w_t &= \tilde{\alpha}_t G_\theta(\epsilon) + \sqrt{1 - \tilde{\alpha}_t^2}\, \xi, \\
w_t^{\mathrm{sg}} &= \tilde{\alpha}_t \mathrm{sg}(G_\theta(\epsilon)) + \sqrt{1 - \tilde{\alpha}_t^2}\, \xi, \\
\mathcal{L}_{\mathrm{IDLM}}^{\mathrm{Duo}}(\theta) &= \mathbb{E}[a_t(w_t, G_\theta(\epsilon))], \\
\mathcal{L}_{\mathrm{DMD\text{-}like}}^{\mathrm{Duo}}(\theta) &= \mathbb{E}[a_t(w_t^{\mathrm{sg}}, G_\theta(\epsilon))].
\end{aligned}
\tag{37}
$$

Figure 15 shows that this distinction matters in the uniform case. The stop-gradient variant quickly enters a high-GenPPL, high-entropy regime, so its entropy reflects instability rather than useful diversity. In contrast, the full IDLM-Duo update remains controlled for most of training. Thus, the formal connection between IDLM and DMD-like matching does not imply that the two generator updates have the same optimization behavior for uniform diffusion; the pathwise gradient through the relaxed intermediate state is empirically important.

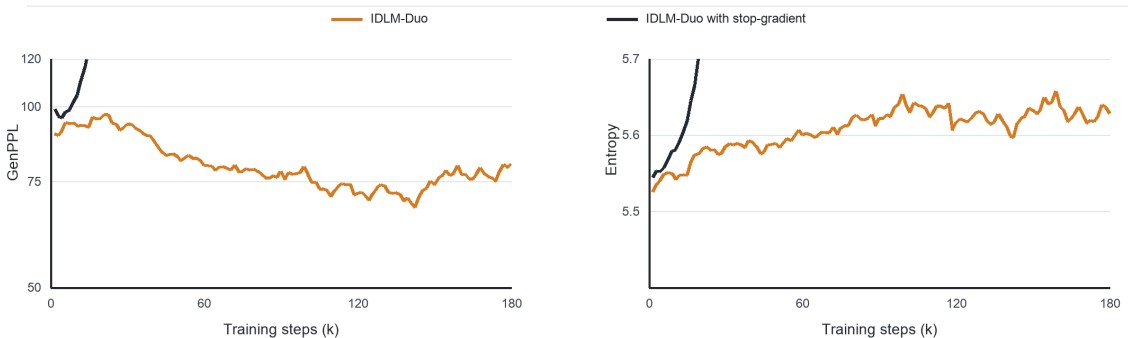

*Figure 15.* **Uniform-diffusion stop-gradient ablation.** Validation GenPPL and entropy for IDLM-Duo and its DMD-like stop-gradient variant. Removing the intermediate-state gradient leads to a divergent high-GenPPL regime, whereas the full IDLM update remains stable.

**Additional latent noise for masked diffusion.** Finally, we test whether extra latent randomness helps masked diffusion in the extreme low-step regime. Unlike uniform diffusion, masked diffusion starts from a deterministic all-mask terminal state, so a deterministic one-step generator has no rich latent source for sample diversity. We therefore inject Gaussian noise into the IDLM-MDLM input as a simple control. Table 4 shows that this modification does not improve GenPPL or entropy:

*Table 4.* **Additional latent noise for IDLM-MDLM.** Values are GenPPL with entropy in parentheses.

| Steps | IDLM-MDLM | IDLM-MDLM + Gaussian noise |
|---|---|---|
| 1 | 4056.12 (5.87) | 4096.54 (5.88) |
| 8 | 79.75 (5.61) | 81.61 (5.61) |
| 16 | 32.75 (5.42) | 35.76 (5.45) |
| 32 | 20.45 (5.23) | 21.03 (5.26) |

The negative result is consistent with Section 5.1: simple input-level Gaussian noise is insufficient for masked diffusion, and improving the low-step regime may require stochasticity that is matched more closely to the discrete absorbing process.

# G. Qualitative Samples

This appendix provides compact qualitative excerpts for the unconditional OpenWebText setting and the conditional TinyGSM setting. Samples are shortened for readability.

## G.1. Unconditional Generation (OpenWebText)

> **📑 Unconditional Generation (OpenWebText)**
>
> - - - - - - - - - - - - - - - - - - - - - - - - - - - - - - - - - - - - - - - - - - - - - - - - - - - - - - - - - - - - - - - -
>
> **SEDD**
> **IDLM-SEDD / 256 Sampling steps**
> **Gen. PPL: 39.24; H: 5.3**
>
> . . . We knew the injuries against Fulham and Newcastle prevented us from gaining victory, but we felt we deserved a big victory in a game which brought us back to the group stage. There were only two fifth division sides in the last eight games, and the Premier League came into the day in third place. . . .
>
> **SEDD / 1024 Sampling steps**
> **Gen. PPL: 43.31; H: 5.2**
>
> . . . I think we want to win every championship with these players, and we have given all I have to say. I know and I do not know that we will rush forward. In our view, we will not give any assurances that we are going to stay. . . .
>
> - - - - - - - - - - - - - - - - - - - - - - - - - - - - - - - - - - - - - - - - - - - - - - - - - - - - - - - - - - - - - - - -
>
> **Masked Diffusion**
> **IDLM-MDLM / 16 Sampling steps**
> **Gen. PPL: 32.75 $\pm$ 1.62; H: 5.42 $\pm$ 0.06**
>
> . . . In the wake of angry remarks from the media, Sam Kicks Radio decided to take a shot, claiming that he had only ever learned how to get away with something bad. I want to get over it, he says, while the exchange returns to the same public dispute. . . .
>
> **MDLM / 1024 Sampling steps**
> **Gen. PPL: 41.29; H: 5.28**
>
> . . . Google claims that Android is its own operating system and could soon be replaced by more complicated cloud services from Samsung and Mozilla. Several business applications on mobile devices bring many direct interaction options. . . .
>
> - - - - - - - - - - - - - - - - - - - - - - - - - - - - - - - - - - - - - - - - - - - - - - - - - - - - - - - - - - - - - - - -
>
> **Uniform Diffusion (Ancestral Sampling)**
> **IDLM-Duo[a] / 16 Sampling steps**
> **Gen. PPL: 78.00; H: 5.6**
>
> . . . I think it is important to worry about better credit security and better integrated trading of those metrics, says a professor of regenerative engineering at McMaster University. The resident range has grown to 7.4 million in the UAE's next 25 years. . . .
>
> **IDLM-DCD[a] / 8 Sampling steps**
> **Gen. PPL: 66.41 $\pm$ 4.59; H: 5.42 $\pm$ 0.03**
>
> . . . I am mostly an economist; for me, I am one of the co-creators of the culture of socialization and a simple person. A computer is enhanced with incredible energy by itself, while the passage moves through biography and explanation. . . .
>
> **Duo-DCD[a] / 16 Sampling steps**
> **Gen. PPL: 75.24; H: 5.53**
>
> . . . My favorite cuisine is pretty varied, changing from regular to country cooking. Some local dishes are fine when guests visit the hotel, and the Southern Japanese Thai restaurant is described as something to watch. . . .
>
> **Duo[a] / 1024 Sampling steps**
> **Gen. PPL: 77.69; H: 5.55**
>
> . . . The Germanic literature is based on a set of cultural expectations, while the English is largely freighted with fiction. Thus, it would be a useful contribution to this field for a shift away from the vernacular. . . .
>
> - - - - - - - - - - - - - - - - - - - - - - - - - - - - - - - - - - - - - - - - - - - - - - - - - - - - - - - - - - - - - - - -
>
> **Uniform Diffusion (Greedy Sampling)**
> **IDLM-Duo[g] / 16 Sampling steps**
> **Gen. PPL: 68.04; H: 5.6**
>
> . . . Strong performance in the NBN Co network will drive costs toward power parity, said Paul McCormick, chief economist at Business Economics. Strong growth of the network in Telstra Powerplows will drive costs in real time. . . .
>
> **IDLM-DCD[g] / 4 Sampling steps**
> **Gen. PPL: 77.47 $\pm$ 4.43; H: 5.28 $\pm$ 0.06**
>
> . . . This device is not the only way to use features in this article. People who conduct interviews in secure environments are attracted by a two-way monitor, making it an ideal candidate for an interviewee who can adapt to secure systems. . . .
>
> **Duo-DCD[g] / 8 Sampling steps**
> **Gen. PPL: 69.58; H: 5.30**
>
> . . . Craft beer from Wisconsin was based on a local foundation, while the market spread through misleading advertising and state-level pushes for ways to craft beer. The passage repeats the theme of freshness and local identity. . . .
>
> **Duo[g] / 1024 Sampling steps**
> **Gen. PPL: 71.72; H: 5.22**
>
> . . . Medical applications and patents are being struck down under the patent devices act. The patent does not include more than medical devices or the described holder, and readers are asked to note changes in size or functionality. . . .

## G.2. Conditional Generation (TinyGSM)

> 📄 **Conditional Generation (TinyGSM)**
>
> ------------------------------------------------------------------------
>
> **IDLM-MDLM / 32 Sampling steps**
> **Accuracy: 12.8%**
>
> **Prompt.** Marcus is half of Leo's age and five years younger than Deanna. Deanna is 26. How old is Leo?
>
> **Generation.**
>
> ```
> def simple_math_problem() -> int:
>     marcus = 26 - 5
>     leo = marcus * 2
>     result = leo
>     return result
> ```
>
> **IDLM-MDLM / 64 Sampling steps**
> **Accuracy: 14.9%**
>
> **Prompt.** Dan plants 3 rose bushes. Each rose bush has 25 roses. Each rose has 8 thorns. How many thorns are there total?
>
> **Generation.**
>
> ```
> def simple_math_problem() -> int:
>     thorns = 3 * 25 * 8
>     result = thorns
>     return result
> ```
>
> **IDLM-MDLM / 128 Sampling steps**
> **Accuracy: 19.9%**
>
> **Prompt.** John collects peaches for 3 hours. He can collect 2 peaches a minute. How many peaches does he collect?
>
> **Generation.**
>
> ```
> def simple_math_problem() -> int:
>     minutes = 3 * 60
>     peaches = 2 * minutes
>     result = peaches
>     return result
> ```
>
> ------------------------------------------------------------------------
>
> **IDLM-Duo / 32 Sampling steps**
> **Accuracy: 15.4%**
>
> **Prompt.** Ted starts with $200. He buys 3 books for 16 dollars each and 3 pencils for 6 dollars each. How much did he spend in total?
>
> **Generation.**
>
> ```
> def simple_math_problem() -> int:
>     total_cost = 3 * 16 + 3 * 6
>     result = total_cost
>     return result
> ```
>
> **IDLM-Duo / 64 Sampling steps**
> **Accuracy: 19.0%**
>
> **Prompt.** Sam ran 3 miles on Monday, Wednesday and Friday. On Tuesday and Thursday, Sam ran 5 miles. How many miles did Sam run this week?
>
> **Generation.**
>
> ```
> def simple_math_problem() -> int:
>     total_miles = (3 * 3) + 5 * 2
>     result = total_miles
>     return result
> ```
>
> **IDLM-Duo / 128 Sampling steps**
> **Accuracy: 21.4%**
>
> **Prompt.** James has to buy insurance. Since he had an accident, it was 60% more than normal. The normal cost is $120 a month. How much does he pay a year?
>
> **Generation.**
>
> ```
> def simple_math_problem() -> int:
>     yearly_cost = (120 * (1 + 0.60)) * 12
>     result = yearly_cost
>     return result
> ```

