# OpenReview forum: "IDLM: Inverse-distilled Diffusion Language Models"
_ICML.cc/2026/Conference — ICML 2026 regular_

### Official Review · Reviewer_d4S4 · 2026-03-06

**Soundness:** 2
**Presentation:** 4
**Significance:** 2
**Originality:** 3
**Overall Recommendation:** 4
**Confidence:** 4

**Summary:**

This paper extends Inverse Distillation of Diffusion Models to the Discrete Case. To do so several hurdles must be solved that do not appear in the continuous case. The first one is the generation of a sample from the learned distribution. Unlike the samples in continuous diffusion, in the discrete case, by default they lose the smooth dependency on model parameters. Another issue, is that the noising process also disables the gradient propagation, due to its discrete nature. This paper sidesteps the first issue by using the predicted probabilities as an output of the distilled model, and then utilizing diffusion duality in the uniform case, and the special properties of the masked case to deal with the second problem. Experiments show that the proposed method can produce low generative perplexity text, while maintaining a respectable entropy score, overall surpassing existing methods.

**Compliance With Llm Reviewing Policy:**

Affirmed.

**Final Justification:**

The authors’ latest response improves my view of the paper. What had previously been a concerning table of results is now meaningfully more encouraging. However, the MAUVE concern is not fully settled, since the evaluation is still based on only 50 generated tokens. A stronger resolution would have been to increase this to 1024 tokens and to show a clear benefit of the proposed method over Duo at 16 generation steps. That additional evidence would have moved me to a clear accept.

Nonetheless, after considering the latest response, the new results can be seen as positive enough that I continue to lean accept.

**Key Questions For Authors:**

Could the authors report the Mauve scores? Can also the perplexity bound results be provided? Any other metrics to estimate distribution coverage?

**Limitations:**

There is no specific section that highlights limitations.

**Strengths And Weaknesses:**

**Strengths**

The strengths of the paper are many. To mention the key ones:

1. The presentation of the paper is excellent. It is considerably clearer, cleaner and intuitive than its predecessors: SiG, IBMD, RealUID.

2. The solution of the unique issues in the discrete case are tackled cleverly.

3. The experimental results, given the metrics used, are promising.

**Weaknesses**

The main weaknesses are the following:

1. The extension of distillation from continuous to discrete diffusion while intuitive is theoretically not well supported. The paper mentions two challenges, but there is a third challenge present in all discrete diffusion distillation methods: In continuous diffusion/flows for every sample in the source distribution there exists a unique image, that is, $p(x_0|x_T)$ is a dirac delta. In the discrete case, $p(x_0|x_T)=p(x_0)$. This means that there is no natural one step sampling procedure. The paper states this in fact, and uses this as motivation to use few step sampling. Unfortunately, it is not certain this resolves the issue. In the case of self-distillation and self-consistency, it is straightforward to show that such an objective will lead to coverage degradation (a noisier state cannot adopt the prediction of all future stochastic states, so numerically it will prioritize the more frequently occurring ones, thus iteratively leading to vicious cycle of mode degradation). For the proposed method this is not certain, but still probable. Which raises the second closely related point.

2. The paper uses the common metrics: generative perplexity and entropy. However, it is trivial to construct a better model than the ones presented in the paper in terms of these two metrics. Indeed, we can construct a single sequence, which has an entropy of 5.7 but a generative perplexity of 20. Then we can define a model that only generates that single sequence. In this case our model would outperform all models in the paper. The issue is that these two metrics do not measure mode coverage. Thus could the authors please also report the MAUVE scores (for all methods) as well, as in the SEDD paper? Can the authors report the perplexity bounds? Since all other papers in this direction focus on generative perplexity and entropy I will not reduce my score regardless of the results, however, I think it is essential that such distillation methods are stress tested in this aspect and I believe this would significantly increase the value of the paper. In case that the proposed methods do not degrade performance as evaluated by such metrics, then such promising results would indicate that the paper deserves a higher score.

**Minor Weaknesses**

1. Entropy scores should report two digits after the decimal in the figures. I believe there is space. A change from 5.50 to 5.54 or 5.59 (depending on the rounding/truncating method) is significant.
2. While the presentation is excellent, I believe it could be further improved if the authors describe the token by token formulation of discrete flow/diffusion directly, instead of the general case. Also the prior work can be compressed a bit.

---

> ### Author Rebuttal · Authors · 2026-03-31
>
> Dear Reviewer d4S4, thank you for your comments. Please find below the answers to your questions and comments.
>
> **[W.1] Coverage degradation for the IDLM method.**
>
> We agree that self-consistency distillation often reduces distributional coverage, but we do not think this limitation applies to IDLM. Our intuition is that self-distillation learns to skip steps in the teacher’s denoising process while preserving a conditional independence factorization. This approximation can fail to capture the full joint distribution and instead favor the most common modes. By contrast, IDLM represents the generated distribution as a mixture of factorized models (see response [W.1] to reviewer J544). It is trained so that a fake diffusion model fitted to samples from this distribution matches the teacher diffusion model trained on the true data distribution. In this view, Theorem 3.1 is central because it guarantees recovery of the underlying data distribution without mode loss. To our knowledge, self-distillation methods lack theoretical guarantees and remain largely empirical.
>
> **[W.2] Can the authors report and MAUVE scores the perplexity bounds?**
>
> Following your suggestion, we computed these metrics. The results for absorbing-state diffusion distillation are presented in Table 1, and those for the uniform-state setting are shown in Table 2.
>
> (Methods are grouped by Steps, with the best metrics **bolded** in each group.)
>
> ## Table 1. Absorbing state distillation
>
> | Method | Steps | GenPPL ↓ | MAUVE ↑ | Entropy ↑ |
> |---|---:|---:|---:|---:|
> | MDLM (Teacher) |  | 41.29 | 0.89 ± 0.01 | 5.28 |
> | SDTT | 32 | 41.45 | 0.91 ± 0.03 | **5.31** |
> | SDTT + Di4C^2 | 32 | 31.51 | **0.93 ± 0.02** | 5.28 |
> | IDLM-MDLM (**Ours**) | 32 | **20.45 ± 1.88** | 0.91 ± 0.01 | 5.23 ± 0.06 |
> | SDTT | 16 | 61.34 | 0.88 ± 0.03 | 5.36 |
> | SDTT + Di4C^2 | 16 | 44.82 | 0.90 ± 0.01 | 5.34 |
> | IDLM-MDLM (**Ours**) | 16 | **32.75 ± 1.62** | **0.93 ± 0.01** | **5.42 ± 0.06** |
> | SDTT | 8 | 121.31 | 0.89 ± 0.03 | 5.39 |
> | SDTT + Di4C^2 | 8 | 84.15 | 0.90 ± 0.04 | 5.38 |
> | IDLM-MDLM (**Ours**) | 8 | **79.75 ± 6.81** | **0.91 ± 0.01** | **5.61 ± 0.04** |
>
> ## Table 2. Uniform state distillation
>
> ### Table 2a
>
> | Method | Steps | GenPPL ↓ | MAUVE ↑ | Entropy ↑ |
> |---|---:|---:|---:|---:|
> | Duo^a (Teacher) |  | 77.69 | 0.96 ± 0.01 | 5.55 |
> | Duo-DCD^a | 32 | 61.31 | **0.97 ± 0.01** | **5.52** |
> | IDLM-DCD^a (**Ours**) | 32 | **42.13 ± 3.65** | 0.94 ± 0.01 | 5.41 ± 0.05 |
> | Duo-DCD^a | 16 | 75.24 | **0.96 ± 0.01** | **5.53** |
> | IDLM-DCD^a (**Ours**) | 16 | **52.06 ± 4.23** | 0.95 ± 0.01 | 5.44 ± 0.05 |
> | Duo-DCD^a | 8 | 111.88 | 0.94 ± 0.01 | **5.52** |
> | IDLM-DCD^a (**Ours**) | 8 | **66.41 ± 4.59** | **0.95 ± 0.01** | 5.42 ± 0.03 |
> | Duo-DCD^a | 4 | 261.82 | 0.79 ± 0.02 | **5.50** |
> | IDLM-DCD^a (**Ours**) | 4 | **111.29 ± 8.39** | **0.88 ± 0.02** | 5.32 ± 0.06 |
>
> ### Table 2b
>
> | Method | Steps | GenPPL ↓ | MAUVE ↑ | Entropy ↑ |
> |---|---:|---:|---:|---:|
> | Duo^g (Teacher) |  | 71.72 | 0.90 ± 0.02 | 5.22 |
> | Duo-DCD^g | 32 | 46.31 | **0.96 ± 0.01** | **5.38** |
> | IDLM-DCD^g (**Ours**) | 32 | **39.34 ± 8.51** | 0.92 ± 0.01 | 5.35 ± 0.11 |
> | Duo-DCD^g | 16 | 54.11 | **0.96 ± 0.01** | 5.37 |
> | IDLM-DCD^g (**Ours**) | 16 | **43.36 ± 3.76** | 0.94 ± 0.01 | **5.41 ± 0.04** |
> | Duo-DCD^g | 8 | 69.58 | 0.93 ± 0.01 | 5.30 |
> | IDLM-DCD^g (**Ours**) | 8 | **53.61 ± 3.08** | **0.96 ± 0.01** | **5.41 ± 0.03** |
> | Duo-DCD^g | 4 | 96.24 | 0.69 ± 0.02 | 4.93 |
> | IDLM-DCD^g (**Ours**) | 4 | **77.47 ± 4.43** | **0.89 ± 0.01** | **5.28 ± 0.06** |
>
> Across all configurations, our method achieves strong performance, ranking **among the best across generation steps**. Notably, in the low-step generation regime, the performance of our model is **best**, which further empirically supporting the claim in [W.1] regarding mitigation of coverage degradation. We will highlight the MAUVE scores and perplexity bounds more clearly in the revised experimental section.
>
> **[W.3] Entropy scores should report two digits after the decimal in the figures.**
>
> We agree that one decimal place may hide meaningful differences. To address this, we added Table 1 in Appendix with double-precision results and will revise all figures to report entropy scores to two decimal places.
>
> **[W.4] Token by token formulation and compression of prior works.**
>
> We appreciate the positive feedback. The token-by-token formulation was omitted only due to the 8-page limit. In the revision, we will add it and compress the prior work section to make a room for MAUVE results.
>
> **[L.1] No Limitations section.**
>
> This section was omitted for the same reasons outlined in [W.4]. In the revised version of the article, we will provide a comprehensive description of the limitations of our method.
>
> **Concluding remarks.** We would be grateful if you could let us know if our explanations have been satisfactory. If so, we kindly ask that you consider increasing your rating. We are also open to discussing any other questions you may have.

---

> > ### Author Rebuttal · Reviewer_d4S4 · 2026-04-02
> >
> > I thank the authors for their detailed clarifications and additional experiments. I agree with the following intuition:
> >
> > *Our intuition is that self-distillation learns to skip steps in the teacher’s denoising process while preserving a conditional independence factorization. This approximation can fail to capture the full joint distribution and instead favor the most common modes.*
> >
> >
> > This is precisely why I had hoped that, in light of Theorem~3.1, the proposed method would mitigate this issue. However, I remain concerned about the MAUVE results: in several cases, particularly at lower numbers of steps (16 and 32), Duo appears to perform better. I did not expect this outcome. Given a sequence length of 1024, 16 denoising steps is already quite small, so I would have expected the proposed method to show a clearer advantage over Duo in this regime.
> >
> > Could the authors comment on this discrepancy? In particular, are they confident that the implementation and evaluation setup are fully correct?
> >
> >
> > **UPDATE after author's reply to the rebuttal acknowledgement.**
> >
> > The authors’ latest response improves my view of the paper. What had previously been a concerning table of results is now meaningfully more encouraging. However, the MAUVE concern is not fully settled, since the evaluation is still based on only 50 generated tokens. A stronger resolution would have been to increase this to 1024 (or 512) tokens and to show a clear benefit of the proposed method over Duo at 16 generation steps. That additional evidence would have moved me to a clear accept.
> >
> > Nonetheless, after considering the latest response, the new results can be seen as positive enough that I continue to lean accept.

---

> > > ### Author Response · Authors · 2026-04-03
> > >
> > > **Clarification on MAUVE results**
> > >
> > > We thank the reviewer for this thoughtful question. We agree that, in light of Theorem 3.1, one might expect our method to mitigate the mode-seeking behavior of self-distillation more clearly, particularly at moderately small numbers of denoising steps. The MAUVE results at 16 and 32 steps are therefore an important point to clarify.
> > >
> > > A key consideration is the evaluation protocol used for the MAUVE calculation, which follows the setups of SEDD (Appendix C.8) and SDTT (Section 4). Under this protocol, each sample consists of a continuation of only 50 generated tokens, conditioned on 50 tokens drawn from WebText. MAUVE is then computed by comparing the distribution of these generated continuations with that of the corresponding real continuations. Because only 50 tokens are generated in total, the number of generated tokens per denoising step remains relatively small at 32 and 16 steps. Specifically, this corresponds to approximately 1.6 tokens per step at 32 steps, 3.1 tokens per step at 16 steps, and 6.3 tokens per step at 8 steps.
> > >
> > > Our interpretation is that, under this evaluation regime, the advantage of our method becomes more apparent only in the more aggressively accelerated setting, where each denoising step must account for a larger portion of the overall generation process. In other words, the benefits predicted by our analysis are most clearly reflected at very small numbers of steps, such as 4 and 8, whereas at 16 and 32 steps Duo and our method remain closer in performance under MAUVE. We therefore do not view the 16- and 32-step results as contradicting the theory. Rather, they suggest that this particular metric and protocol are less informative in this setting.
> > >
> > > Regarding the correctness of the implementation and evaluation setup, we have carefully verified both. In particular, we use the same MAUVE evaluation protocol as SEDD and SDTT, and all Duo-based baselines were compared using the same codebase and evaluation pipeline. For this reason, we are confident that the reported comparison is fair and that the observed trend does not arise from an implementation or evaluation error.
> > >
> > > **Concluding remarks.** Thank you for the thoughtful discussion. We appreciate the opportunity to clarify this point, and we hope that our explanation has made the results and their interpretation clearer. If you find our response satisfactory, we would be grateful if you would consider raising your rating.

---

### Official Review · Reviewer_MEfP · 2026-03-10

**Soundness:** 3
**Presentation:** 3
**Significance:** 3
**Originality:** 3
**Overall Recommendation:** 3
**Confidence:** 3

**Summary:**

This paper proposes a new method called Inverse Distillation Diffusion Language Model (IDLM), aiming to address the slow inference speed issue of diffusion language models in text generation. The authors extend the inverse distillation framework originally used for continuous diffusion models to the discrete data domain. By training a few-step generator (student model) to approximate the pre-trained diffusion language model (teacher model), the method achieves a significant reduction in sampling steps while maintaining the generation quality and diversity. The method theoretically proves that the unique solution of the optimization objective can restore the real data distribution, and by introducing simplex relaxation and re-parameterization techniques, it overcomes the non-differentiability challenges in the discrete space. Experimental results show that IDLM achieves up to 256 times faster inference speed on various diffusion language models such as SEDD, MDLM, and Duo, and performs on par with or even better than the original model in terms of generation perplexity and diversity metrics.

**Compliance With Llm Reviewing Policy:**

Affirmed.

**Key Questions For Authors:**

1. What are the differences between the diffusion language model and the previous language models?
2. Table 1 is placed too far back, which might affect the readers' reading experience.
3. It is suggested to include a summary conclusion in the main text.

**Limitations:**

yes

**Strengths And Weaknesses:**

Soundness
The author addressed the issue that the multi-step sampling in diffusion language models leads to slow inference speed, which limits the practical application of DLMs. To this end, the author proposed the Inverse Distillation Diffusion Language Model (IDLM). Moreover, the author theoretically proved the uniqueness of the global optimal solution of the IDLM objective function and verified the correctness of the proposed optimization method. The methodological part detailed the specific implementation of this approach. Furthermore, IDLM achieved a significant 256-fold speedup in inference on various diffusion language models such as SEDD, MDLM, and Duo.
Presentation
The submitted article is written clearly and the overall narrative is easy to understand.
Significance
The paper addressed the issue of slow inference speed of diffusion language models. Solving this problem, it promoted the application of machine learning in the field of diffusion language models.
Originality
In response to the slow inference speed issue of diffusion language models, the authors provided a theoretical proof demonstrating the uniqueness of the global optimal solution of the IDLM objective function and verified the correctness of the proposed optimization method.

---

> ### Author Rebuttal · Authors · 2026-03-31
>
> Dear Reviewer MEfP, thank you for your comments. We were happy to see that **no major flaws** were mentioned in our work from your side. As you mentioned, the paper addresses the slow inference speed caused by multi-step sampling in diffusion language models by proposing the Inverse-distilled Diffusion Language Model (IDLM). We believe that our article has the following strengths: (1) a good theoretical basis, (2) practical solutions enabling effective training, and (3) strong empirical results.
>
> We appreciate your feedback that the article is clear, easy to follow, and makes an original and significant contribution toward making diffusion language models more practical. Please find below the answers to your questions and comments.
>
> **[Q.1] What are the differences between the diffusion language model and the previous language models?**
>
> As we understand your question, it concerns the distinction between previous language models, typically based on a dominant autoregressive approach, and diffusion-based language models. To address this clearly, we will (i) highlight the differences between autoregressive and diffusion-based approaches, and then (ii) explain how our IDLM method fits within the diffusion language model paradigm.
>
> (i) The primary difference between autoregressive and diffusion-based approaches lies in the generation paradigm and how the joint probability of a sequence is modeled.
>
> Standard previous language models (GPT-like) are autoregressive. They model the sequence strictly left-to-right:
>
> $$
> p_{\theta}(x_0) = \prod_{i=1}^L p_{\theta}(x_0^i \mid x_0^{<i}),
> $$
>
> where $p_{\theta}$ denotes the model, $x_0$ represents the sequence, $L$ is the sequence length, $x_0^i$ denotes the token at position $i$, and $x_0^{<i} = (x_0^1, \dots, x_0^{i-1})$ denotes the prefix of tokens preceding position $i$.
>
> In contrast, diffusion language models (e.g., MDLM, UDLM, Duo) are non-autoregressive. They model sequences by learning to reverse a Markovian corruption process (e.g., absorbing or uniform noise). In the discrete-time setting, this can be approximated as:
>
> $$
> p_{\theta}(x_0) \approx \sum_{x_1, x_2, \dots, x_T} p_{\theta}(x_0, x_1, x_2, \dots, x_T) = \sum_{x_1, x_2, \dots, x_T} \left( p(x_T) \prod_{t=0}^T p_{\theta}(x_{t-1} \mid x_t) \right).
> $$
>
> The key difference is that during generation, all tokens are updated simultaneously and can attend to the entire sequence at every step, rather than just a left-to-right prefix.
>
> This makes diffusion models a strong alternative to autoregressive approaches, offering competitive generation quality [1]. However, they suffer from slow inference due to the large number of required denoising steps, and this is exactly the limitation our method aims to address.
>
> (ii) A key practical limitation of diffusion language models is that their strong generation quality typically comes at the cost of many iterative denoising steps ($T \gg 1$), which makes inference slow and computationally expensive. This significantly limits their usability in real-world applications where latency and efficiency are critical. Our proposed IDLM directly addresses this bottleneck by preserving high-quality generation of the diffusion language models, but using only a few steps. This makes our IDLM method not only competitive in quality, but also practical and scalable for real-world use.
>
> **[Q.2] Table 1 is placed too far back, which might affect the readers' reading experience.**
>
> We agree with your observation and acknowledge that Table 1 was moved to the Appendix due to the 8-page limit and the need to compare a large number of baselines. However, following your advice, in the revised version we will incorporate the key parts of Table 1 into the main text by utilizing the additional page allowance, thereby ensuring that the readers’ experience is not adversely affected.
>
> **[Q.3] It is suggested to include a summary conclusion in the main text.**
>
> Thank you for the suggestion. We omitted the conclusion section in the initial submission for the reasons outlined in [Q.2]. In the revised version of the article, taking advantage of the additional page allowance, we will include a comprehensive conclusion that summarizes our contributions, highlights the effectiveness of IDLM, and outlines potential directions for future work.
>
> **Concluding remarks.** We would be grateful if you could let us know if our explanations have been satisfactory. If so, we kindly ask that you consider increasing your rating. We are also open to discussing any other questions you may have.
>
> **References:**
>
> [1] Large Language Diffusion Models

---

> > ### Author Rebuttal · Reviewer_MEfP · 2026-04-02
> >
> > I have no more questions about the author's refutation and reply.

---

### Official Review · Reviewer_qQmE · 2026-03-11

**Soundness:** 3
**Presentation:** 3
**Significance:** 3
**Originality:** 3
**Overall Recommendation:** 4
**Confidence:** 5

**Summary:**

The paper proposes IDLM, which adapts inverse distillation to diffusion language models. Instead of directly matching teacher trajectories as in consistency-style approaches, the student is trained so that the pretrained teacher remains optimal under the student-induced distribution, while a “fake” diffusion model is alternately trained on student samples. Experimentally, they evaluate on SEDD, MDLM, and Duo / Duo-DCD trained on OpenWebText, using GenPPL and sequence entropy, and report substantial step reductions, about 4× for SEDD, 64× for MDLM, and up to 128× / 256× relative to original Duo.

**Compliance With Llm Reviewing Policy:**

Affirmed.

**Final Justification:**

The rebuttal is thorough and the ablation study is convincing.

**Key Questions For Authors:**

See questions above.

**Limitations:**

See limitations above.

**Strengths And Weaknesses:**

# Strengths:
1. The paper attacks a very real bottleneck in diffusion LLMs: inference requires many denoising steps, which hurts latency and practicality. This paper explores a different family: inverse distillation with a fake model, which could beyond just skipping steps.

2. The most novel part is not the speedup itself, but the theorem showing that for the chosen discrete objectives, the inverse loss lower-bounds KL and has the correct unique optimum.

3. The method is reasonably unified across multiple DLM families.

4. Empirical results are strong in the low-step context.

# Weaknesses:
1. the experiments are limited in openwebtext. Could you have more verification on diverse downstream task.
2. No ablation study to test what actually matters.
3. Need to compare with more advanced methods like [1][2].
4. How much additional training instability does the fake model optimization introduce relative to SDTT / Duo-DCD.



[1] Ultra-fast language generation via discrete diffusion divergence instruct

[2] DLM-One: Diffusion Language Models for One-Step Sequence Generation

---

> ### Author Rebuttal · Authors · 2026-03-31
>
> Dear Reviewer qQmE, thank you for your insightful comments and for taking the time to review our work. We appreciate your high confidence and constructive feedback. Please find below the answers to your questions and comments.
>
> **[W.1] Verification on diverse downstream task.**
>
> Most downstream evaluations use benchmark validation, and many models are instruction-tuned for task following. By contrast, as noted in [W.3], we focus on unconditional generation, so downstream task evaluation is beyond this article’s scope.  Our main goal is to develop the theory and demonstrate a proof of concept for inverse distillation in discrete diffusion models, so rather than adapting the method to instruction-based settings, we test it on a wide range of existing diffusion models. To our knowledge, this is the first **unified distillation framework** for absorbing and uniform diffusion. We acknowledge the limitation and view extending IDLM to conditioned generation as an important direction for future work.
>
> **[W.2] No ablation study**
>
> Thank you for pointing this out. We believe IDLM is effective because of three core components, which we further ablate:
>
> - **Fake model** To show the importance of the fake model, we train an IDLM variant without it.
>
> - **Multistep** As noted in the main text (lines 299–301), distilling a single-step text generator remains practically difficult. To show the importance of multistep distillation, we also report results without it.
>
> - **Simplex relaxation** As discussed in lines 224–255, we use simplex relaxation to efficiently optimize the inverse distillation loss. The original loss requires a direct map $G_{\theta}: \mathcal{Z} \rightarrow \mathcal{V}$ from latent space to the vocabulary. To show the importance of simplex relaxation, we also test a direct-to-vocab variant using hard Gumbel-softmax to make the objective differentiable.
>
> We used the same IDLM-MDLM setup as in the main text, with results (GenPPL ↓, Entropy ↑) shown in the table below:
>
> |Steps|IDLM-MDLM|No-fake|No-multistep|No-simplex|
> |---:|---:|---:|---:|---:|
> | 1 | — | — | 2823 / 5.99 | — |
> | 8 | **79.75 / 5.61** | 146.08 / 5.44 | 1177 / 5.71 | 290.13 / 5.82 |
> | 16 | **32.75 / 5.42** | 52.43 / 5.27 | 964 / 5.61 | 106.06 / 5.66 |
> | 32 | **20.45 / 5.23** | 35.77 / 5.27 | 692 / 5.44 | 64.66 / 5.61 |
>
> We observe that, without the fake model, the GenPPL/Entropy frontier worsens and the model eventually collapses to a single-token output. Without multistep training, the student cannot generate meaningful outputs. Without simplex relaxation, performance also degrades and convergence is much slower. These results show that each core component is important to the final outcome.
>
> **[W.3] Need to compare with [1][2]**
>
> We thank the reviewer for highlighting these recent works. Table 1 compares our method with DiDi-Instruct [1].
>
> ## Table 1. Comparison with [1]
>
> Methods are grouped by Steps, with the best metrics **bolded** in each group.
>
> |Method|Steps|GenPPL↓|Entropy↑|
> |---|---:|---:|---:|
> |MDLM (Teacher)||41.29|5.28|
> |DiDi-Instruct [1]|32|24.97|5.18|
> |IDLM-MDLM (**Ours**)|32|**20.45**|**5.23**|
> |DiDi-Instruct [1]|16|38.19|5.21|
> |IDLM-MDLM (**Ours**)|16|**32.75**|**5.42**|
> |DiDi-Instruct [1]|8|**62.24**|5.17|
> |IDLM-MDLM (**Ours**)|8|79.75|**5.61**|
>
> As shown, IDLM **outperforms [1]** at nearly all generation steps. The only exception is at 8 steps, where it achieves comparable GenPPL but higher entropy. Overall, these results support the effectiveness of our approach relative to DiDi-Instruct.
>
> Regarding [2], we cannot provide a direct comparison because its method does not match our evaluation setup. Our work follows prior studies, such as MDLM and Duo, on unconditional generation, and both our theory and experiments are designed accordingly. In contrast, [2] focuses on instruction-conditioned text generation.
>
> **[W.4] Training instability associated with the additional optimization of the fake model**
>
> We found that fake-model optimization does *not* cause training instability. To show this, Table 2 reports the target GenPPL metric over training iterations for IDLM models with 32 sampling steps. We include randomly selected early iterations to show initial improvement, and late iterations after convergence to show low variance and stable performance. Together, these results suggest that training is stable and the metric does not diverge.
>
> ## Table 2. GenPPL over training iterations
>
> |Iteration|IDLM-MDLM|IDLM-Duo|IDLM-DCD|
> |---:|---:|---:|---:|
> |1,499|178.06|106.55|62.07|
> |7,499|137.47|95.26|62.59|
> |10,499|119.15|92.87|59.78|
> |...|...|...|...|
> |103,499|17.07|61.95|44.34|
> |104,999|16.66|61.57|42.48|
> |110,999|17.31|59.28|47.89|
> |116,999|17.28|65.61|48.44|
>
> **Concluding remarks.** We would be grateful if you could let us know if our explanations have been satisfactory. If so, we kindly ask that you consider increasing your rating. We are also open to discussing any other questions you may have.
>
> **References:**
>
> Same as in the review

---

> > ### Author Rebuttal · Reviewer_qQmE · 2026-04-03
> >
> > The rebuttal is thorough and the ablation study is convincing; we suggest adding a small-scale conditional generation experiment, qualitatively discussing methodological differences with DLM-One, and briefly explaining whether the late-stage IDLM-DCD training variance is within expected range. I'm willing to raise my score.

---

> > > ### Author Response · Authors · 2026-04-08
> > >
> > > **Conditional generation experiment.**
> > >
> > > Following your feedback, we added a small-scale conditional generation experiment. It was limited by three factors: (i) limited compute, (ii) the need to use a pretrained diffusion model, since training a high-quality one from scratch is too demanding under these constraints, and (iii) limited time to prepare this response. We therefore chose a fine-tuned diffusion language model (DLM) for conditional generation suited to this setting, `Qwen3-0.6B-diffusion-mdlm` from [1], which was trained using the pretrained autoregressive model Qwen and the MDLM training procedure. We follow the SFT setup of [1] and train IDLM on the Alpaca dataset. The GSM8K results across training iterations are shown in the table:
> > >
> > > |Method/Steps|256|128|64|32|16|
> > > |---|---:|---:|---:|---:|---:|
> > > |Teacher|31.01±1.27|28.35±1.24|15.92±1.01|11.68±0.88|3.64±0.52|
> > > |IDLM (5250)|-|-|-|11.75±0.89|-|
> > > |IDLM (6750)|-|-|-|12.28±0.90|-|
> > > |IDLM (7500)|-|-|**17.13±1.04**|**13.19±0.93**|**3.94±0.54**|
> > >
> > > As shown in the table, IDLM **improves accuracy** in the low-step regime for conditional generation. Entries marked with “-” indicate configurations that were not evaluated due to time constraints.
> > >
> > > **Please note that**, we trained the model for only 7,500 iterations with a batch size of 32 and did not tune hyperparameters. Nevertheless, we believe that these improvements over training iterations serve as a proof of concept that IDLM can be successfully applied in the conditional generation setting and we expect that longer training and a broader hyperparameter search could further improve the results.
> > >
> > > **Qualitative difference with DLM-One**
> > >
> > > Although both IDLM and DLM-One are distillation-based approaches for DLMs, they differ in their conceptual foundations and in the types of methods to which they are applied. To clarify this distinction, we (i) first introduce the two main categories of DLMs, and (ii) then explain how distillation approaches differ.
> > >
> > > (i) DLMs fall into two broad types: *continuous* and *discrete*. Discrete DLMs operate directly on tokens, matching the symbolic nature of language and enabling bidirectional generation. Recent models such as MDLM and Duo show this approach can be highly competitive. Continuous DLMs, by contrast, apply diffusion to embeddings or latent vectors using a standard Gaussian process [2]. Because language itself is discrete, these models must then map continuous states back to tokens. The **main difference** is where diffusion is applied: tokens in discrete models, latent representations in continuous ones.
> > >
> > > (ii) IDLM and DLM-One differ because they apply to different types of DLMs. IDLM adapts Inverse Distillation [3], originally developed for continuous diffusion models, to the discrete DLM setting. This adaptation was not straightforward, as it required addressing both **theoretical** and **practical** challenges: proving the loss function is valid (Theorem 3.1) and handling difficult backpropagation in discrete space through gradient-stable relaxations. DLM-One is likewise based on a distillation approach, specifically SiD [4], which was originally designed for continuous diffusion models. However, DLM-One transfers this idea directly to continuous DLMs, such as DiffuSeq [2].
> > >
> > > Thus, while both IDLM and DLM-One use distillation to make DLMs faster and more practical, they approach this challenge from different directions: IDLM distills in discrete token space, while DLM-One distills in continuous latent space.
> > >
> > > **Late-stage IDLM-DCD training variance**
> > >
> > > We believe the late-stage training variance of IDLM-DCD is within the expected range. GenPPL remains broadly stable near the end of training, with no sign of runaway variance or late-stage degradation. Across the last 20 logged points (iterations 98,999–127,499), the mean GenPPL is 45.87 with a standard deviation of 3.9. These measurements were taken during training using only 8 samples per estimate for efficiency, so some noise is expected. Even with this small sample size, the variance is modest, with an 8% coefficient of variation. Overall, these results suggest that late-stage IDLM-DCD training remains stable.
> > >
> > > **Conclusion.** We thank the reviewer for the constructive feedback. In response, we added a small-scale conditional generation experiment showing that IDLM improves accuracy in the low-step regime; clarified the distinction between IDLM and DLM-One; and provided further evidence that late-stage IDLM-DCD training remains stable. These additions strengthen both the empirical and conceptual contributions of the paper. We therefore respectfully hope the reviewer will consider raising their score in light of these new results and clarifications.
> > >
> > > **References:**
> > >
> > > [1] dLLM: Simple Diffusion Language Modeling
> > >
> > > [2] DiffuSeq: Sequence to Sequence Text Generation with Diffusion Models
> > >
> > > [3] Inverse Bridge Matching Distillation
> > >
> > > [4] Score identity Distillation

---

### Official Review · Reviewer_2suj · 2026-03-18

**Soundness:** 3
**Presentation:** 3
**Significance:** 3
**Originality:** 3
**Overall Recommendation:** 5
**Confidence:** 4

**Summary:**

Uniform-state diffusion models are highly compelling for language modeling because they enable few-step generation, unlike masked diffusion models or autoregressive (AR) models. In this paper, the authors present Inverse-Distilled Diffusion Language Models, which make significant progress over the discrete consistency models proposed in [1] for uniform diffusion and in [2] for masked diffusion models. Notably, the proposed approach yields a 4–64x speedup.


[1] Sahoo et al., ICML 2025 "The Diffusion Duality"

[2] Deschenaux et al., ICLR 2025 "Self distillation through time"

**Compliance With Llm Reviewing Policy:**

Affirmed.

**Final Justification:**

The authors addressed my concerns

**Key Questions For Authors:**

See above

**Limitations:**

yes

**Strengths And Weaknesses:**

### Strengths

Uniform-state diffusion are compelling because of few step generation [1] which isn't possible with masked dLLMs or AR models.
The method is technically sound and represents a significant improvement over prior distillation approaches [1, 2].

### Weaknesses

The experiments are conducted only at a relatively small scale (i.e., 110M parameters). Demonstrating similarly promising improvements at a larger scale, such as 1B+ parameters, would make the method even more compelling.

### References

[1] Sahoo et al., ICML 2025 "The Diffusion Duality"

[2] Deschenaux et al., ICLR 2025 "Self distillation through time"

---

> ### Author Rebuttal · Authors · 2026-03-31
>
> Dear Reviewer 2suj, thank you for your comments. Please find below the answers to your questions and comments.
>
> **[W.1] The experiments are conducted only at a relatively small scale (i.e., 110M parameters). Demonstrating similarly promising improvements at a larger scale, such as 1B+ parameters, would make the method even more compelling.**
>
> We fully agree that demonstrating our IDLM method at the 1B+ parameter scale is an important and natural question. However, we were constrained by two factors: (i) limited computational resources, and (ii) the need to rely on a pretrained diffusion model, since training a high-quality diffusion model from scratch is itself computationally demanding under such constraints. Accordingly, we framed this work as a theoretical foundation for inverse distillation in discrete diffusion models and aimed to provide a proof of concept. We therefore chose the 169M-parameter scale, as it was computationally feasible and offered access to several pretrained diffusion models (e.g., MDLM and Duo) as well as relevant distillation baselines (e.g., SDTT and Duo-DCD). Thus, rather than focusing on scaling our method, we chose to cover a broad range of existing diffusion models. To the best of our knowledge, our work is the first to provide a **unified distillation framework** for both absorbing and uniform diffusion.
>
> However, following your feedback, we found a checkpoint for MDLM model with 0.9B parameters from SDTT work. To the best of our knowledge, the only concurrent baseline available in this setup is SDTT. The corresponding results and comparison are presented in Table 1.
>
> ## Table 1. Scaling experiments
>
> Methods are grouped by Steps, with the best metrics **bolded** in each group.
>
> | 169M parameters | Steps | GenPPL ↓ | Entropy ↑ | 863M parameters | Steps | GenPPL ↓ | Entropy ↑ |
> |---|---:|---:|---:|---|---:|---:|---:|
> | MDLM (Teacher) |  | 41.29 | 5.28 | MDLM (Teacher) |  | 34.99 | 5.24 |
> | SDTT | 32 | 41.45 | **5.31** | SDTT | 32 | 31.16 | 5.12 |
> | IDLM-MDLM (**Ours**) | 32 | **20.45 ± 1.88** | 5.23 ± 0.06 | IDLM-MDLM (**Ours**) | 32 | **13.71 ± 1.70** | **5.17 ± 0.10** |
> | SDTT | 16 | 61.34 | 5.36 | SDTT | 16 | 52.13 | 5.20 |
> | IDLM-MDLM (**Ours**) | 16 | **32.75 ± 1.62** | **5.42 ± 0.06** | IDLM-MDLM (**Ours**) | 16 | **24.07 ± 2.13** | **5.36 ± 0.04** |
> | SDTT | 8 | 121.31 | 5.39 | SDTT | 8 | 121.51 | 5.25 |
> | IDLM-MDLM (**Ours**) | 8 | **79.75 ± 6.81** | **5.61 ± 0.04** | IDLM-MDLM (**Ours**) | 8 | **67.37 ± 4.11** | **5.59 ± 0.03** |
>
> As shown in the table, our method scales favorably and continues to deliver strong performance at larger model sizes. In particular, it achieves **superior results** compared to the SDTT baseline in the 0.9B-parameter setting. We believe these findings provide encouraging evidence that IDLM remains effective beyond the 169M scale and has strong potential for further improvements at even larger scales.
>
> **Concluding remarks.** We would be grateful if you could let us know if our explanations have been satisfactory. If so, we kindly ask that you consider increasing your rating. We are also open to discussing any other questions you may have.

---

> > ### Author Rebuttal · Reviewer_2suj · 2026-04-03
> >
> > Thanks for running the extra experiments. I understand the there are no pre-trained Uniform-state diffusion LLM available at scale. But the authors should consider adding some evals at 1B+ scale to make the paper more compelling.
> >
> > I'm increasing my score to 5. Good luck.

---

### Official Review · Reviewer_J544 · 2026-03-19

**Soundness:** 2
**Presentation:** 1
**Significance:** 3
**Originality:** 4
**Overall Recommendation:** 4
**Confidence:** 4

**Summary:**

This paper proposes inverse-distilled language models (IDLM), a new method for discrete diffusion distillation. Discrete diffusion models can in principle support few-step generation, but their performance is often poor. To preserve performance, several distillation methods have been proposed, such as SDTT. This paper introduces a new method called inverse distillation. The method can be applied to both absorbing-state discrete diffusion models and uniform-state discrete diffusion models. The authors also prove that the optimum is achieved when the teacher distribution and the student distribution are the same. In experiments, IDLM performs better than Duo-DCD and SDTT.

**Compliance With Llm Reviewing Policy:**

Affirmed.

**Final Justification:**

My major concern was the validity of Theorem 1. In the authors’ rebuttal, however, I found two points reasonably convincing: (1) the student distribution in Theorem 1 does not correspond to a specific modeling choice, and (2) the additional experiments with injected noise provided some empirical support. Based on these clarifications, I have raised my score.

I understand the paper’s core contribution, and the empirical results are also strong. Overall, I support this paper. At the same time, as I already noted in W1, the writing remains a major weakness, and I hope the following points will be addressed in the final version: 1) The paper should focus less on presenting the theory too strongly and instead provide more intuition for readers. As the authors explained in their response to W1, the paper would benefit from a clearer and more intuitive explanation of why existing discrete diffusion distillation methods are problematic, and why IDLM can overcome those issues. 2) Theorem 3.1 is stated too strongly. The proof is not for a multivariate setting such as text sequences, and as the authors themselves clarified in the rebuttal, the student distribution does not correspond to a specific modeling choice. Nevertheless, both the statement and the surrounding paragraph are written in a way that sounds overly strong. This could be problematic if future researchers accept the theorem at face value without fully understanding its limitations. In my view, the theorem is better understood as a proof in a toy setting that provides supporting intuition for the validity of the loss. That is still valuable, but the paper should make this limitation explicit.

The authors may feel somewhat negative about my review, but I hope they understand that my comments were intended as a detailed review to help the discrete diffusion field itself progress. I also hope these points will be reflected in the final version.

**Key Questions For Authors:**

1. What is the intuitive reason why IDLM performs well?

2. For absorbing-state diffusion that models multivariate sequences, does an optimal solution exist, and is it actually attainable?

**Limitations:**

The paper includes an impact statement, but it does not discuss its limitations. For limitations, please refer to W2.

**Strengths And Weaknesses:**

**Strengths**

1. To the best of my knowledge, this is the first method that can be applied to both absorbing-state discrete diffusion models and uniform-state discrete diffusion models.
2. Despite this broader applicability, it outperforms SDTT and Duo-DCD, which were designed for specific settings.
3. It is also the first paper to introduce inverse distillation to discrete diffusion.

**Weaknesses**

1. **The main issue is the writing.** More specifically, the paper focuses too much on theoretical convergence and on how the theoretical challenges are addressed.
   - It is hard to see why this method should work well in practice. The core problem here is few-step distillation. In that case, the paper should first explain what makes few-step distillation difficult and how the proposed method addresses those difficulties. Instead, the paper mainly discusses the theory needed to apply inverse distillation to discrete diffusion. Theorem 1 is provided, but it is still unclear why this should lead to a practically strong algorithm. Even if inverse distillation has been studied in image diffusion distillation, the paper gives very little intuition for why it should improve performance here.
   - It is also unclear why IDLM should outperform prior methods such as SDTT and Duo-DCD. The introduction briefly mentions these methods, but does not explain what they do, what their limitations are, or why IDLM should be better. I read both the preliminaries and related work sections, but they spend much more space on image diffusion distillation and inverse distillation than on the discrete diffusion literature most relevant to this paper.
   - **Why is there no conclusion section?**

2. Although the paper is written with a strong theoretical focus, **Theorem 3.1 does not seem very meaningful.**
   - Theorem 3.1 is proved only for the single-token case, but the target of discrete diffusion language modeling is language, not a single-token vocabulary.
   - Even if the result could be extended to multivariate sequences, it would have limited meaning unless the model is based on uniform discrete diffusion. Discrete diffusion generally represents a joint probability distribution as a product of independent probability distributions. Because of this, many prior works [1, 2, 3] have discussed that absorbing-state discrete diffusion models cannot fully match the joint distribution in few-step generation. If a few-step student cannot theoretically match a multi-step teacher distribution, then what is the practical meaning of showing that IDLM has a unique solution there? In such a setting, the desired solution does not even exist in the first place.

3. **The related work discussion is incomplete.**
   - Di4C [1] is also a well-known distillation method for absorbing-state discrete diffusion models, but it is not discussed in the paper. This is especially important because Di4C addresses the limitation mentioned above, namely the inability of absorbing-state discrete diffusion to represent the full joint distribution, and it also provides theoretical analysis. Both theoretically and empirically, it seems important to discuss this paper here.

The experimental section also feels somewhat limited, especially since the low-scale discrete diffusion models are all below 200M parameters and trained on OWT. Still, I can understand this choice because the paper evaluates both absorbing-state and uniform-state discrete diffusion models. My main concern is that **the paper is written with a very strong theoretical emphasis, yet it is still not clear whether the theory is well justified** or practically meaningful. If the authors can clarify these points, I would be open to raising my score.

[1] Hayakawa et al., "Distillation of Discrete Diffusion through Dimensional Correlations," ICML'25.

[2] Liu et al., "Discrete Copula Diffusion," ICLR'25.

[3] Xie et al., "Variational Autoencoding Discrete Diffusion with Enhanced Dimensional Correlations Modeling," ICLR'26.

---

> ### Author Rebuttal · Authors · 2026-03-31
>
> Dear Reviewer J544, thank you for your comments. Please find below the answers to your questions and comments.
>
> **[W.1] The main issue is the writing. ...**
>
> We agree that the concerns in your first weakness are connected and address them together below.
>
> **Why few-step distillation is hard.** Discrete diffusion typically models reverse transitions in a factorized way:
> $$p_{\theta}(x_{t-1}|x_t)=\prod_{i=1}^Lp_{\theta}(x_{t-1}^i|x_t).$$
> With many denoising steps, the composition of many such transitions can approximate the data distribution.
> $$p_{\text{data}}(x_0)\approx\sum_{x_1,x_2,\dots,x_T}\left(p(x_T)\prod_{t=0}^T\prod_{i=1}^L p_{\theta}(x_{t-1}^i|x_t)\right).$$
> In the few-step regime, however, this factorized parameterization becomes the bottleneck, especially for absorbing processes where $p(x_T)$ is concentrated on one masked sequence. This is why few-step distillation is difficult.
>
> **How IDLM addresses this bottleneck.** To overcome this limitation, we parametrize $p_{\theta}$ using a generator $G_{\theta}: \mathcal{Z} \rightarrow \mathcal{V}$ or $G_{\theta}: \mathcal{Z} \rightarrow \Delta$ for simplex relaxation (see lines 220--231), where $\mathcal{Z}$ has a tractable posterior $p_{\mathcal{Z}}$. Thus, we implicitly use a mixture of models similar to [1, 3], and $p_{\theta}$ is given by:
> $$p\_{\theta}(x_0) = \mathbb{E}\_{p_{\mathcal{Z}}(z)}[p\_{\theta}(x_0|z)], $$
> We then learn this student distribution with our inverse distillation loss in Eq.~10, extended to the multistep case in lines 300--320.
>
> **Why Theorem 1 matters.** Theorem 1 justifies this objective: our inverse distillation loss for a general discrete diffusion is zero iff $p_{\theta}(x_0) = p^*(x_0)$. Under an ideal teacher, minimizing it is therefore aligned with recovering the target data distribution.
>
> **Why IDLM should outperform prior methods such as SDTT and Duo-DCD.** Prior methods such as SDTT and Duo-DCD are based on the consistency/self-distillation idea: they compress several teacher denoising steps into one student step. In that formulation, the student is still a factorized few-step model, so it remains limited by the same bottleneck discussed above.
>
> By contrast, our approach changes the optimization target itself: instead of only matching a shortened teacher trajectory, it directly optimizes the student distribution $p_{\theta}(x_0)$ through inverse distillation. This is why we expect IDLM to improve when few-step factorized students become too restrictive. In addition, SDTT and Duo-DCD are justified primarily empirically, whereas our method is accompanied by Theorem 3.1, which shows that the inverse-distillation objective is minimized exactly at $p_{\theta}(x_0)=p^*(x_0)$.
>
> **[W.2] Although the paper is written with a strong theoretical focus, Theorem 3.1 does not seem very meaningful. ... If a few-step student cannot theoretically match a multi-step teacher distribution, then what is the practical meaning of showing that IDLM has a unique solution there?**
>
> You are correct that a *standard* factorized absorbing-state model cannot match the joint distribution in a few steps. However, IDLM does not use a standard factorized model. As discussed in W.1 and prior works [3], a student parameterized as a mixture of factorized models can model the data distribution in a few steps. **Hence, there is no contradiction: the desired solution exists**, and Theorem 3.1 proves uniqueness.
>
> **[W.3] The related work discussion is incomplete. ... Di4C [1] is also a well-known distillation method for absorbing-state discrete diffusion models, ...**
>
> We agree that Di4C [1] also addresses the limitation of factorized models using a mixture parametrization with a latent variable. Specifically, it uses an idea similar to SDTT and DUO, but unlike them, it proposes a mixture model parametrization. Di4C [1] learns such a model with the approximation $\log \mathbb{E}\_{q(z)}\left[p(x_{t-1}|x_t, z)\right] \approx \mathbb{E}\_{q(z)}\left[\log p(x_{t-1}|x_t, z)\right]$.
>
> Thus, Di4C is similar in overcoming the factorization problem using mixtures, but differs in how such a model is learned. Following your feedback, we also empirically compare our model with Di4C$^2$ in the answer **[W.2]** for reviewer d4S4 and show that **IDLM outperforms Di4C$^2$**.
>
> **[W.4] Why is there no conclusion section?**
>
> We omitted the conclusion section in the initial submission due to the 8-page limit. We will add a short conclusion in the revision.
>
> **[W.5] The experimental section also feels somewhat limited, especially since the low-scale discrete diffusion models are all below 200M parameters and trained on OWT**
>
> Following your feedback, we provided results for distilling the 0.9B MDLM model from the SDTT paper; see the answer **[W.1]** for reviewer 2suj.
>
> **Concluding remarks**. We would be grateful if you could let us know if our explanations have been satisfactory. We are also open to discussing any other questions you may have.

---

> > ### Author Rebuttal · Reviewer_J544 · 2026-04-03
> >
> > Thank you for the detailed rebuttal. My concerns regarding W.1 and W.3 have been addressed.
> >
> > However, I still do not agree about the justification for what I referred to as W.2 (which the authors describe as W.3), namely Theorem 3.1. Di4C [1] and VADD [3] justify few-step generation through **explicit latent variable modeling**. However, in this work, the student model itself does not appear to adopt latent variable modeling. Rather, the model being trained seems to directly reuse previously trained models such as SEDD and MDLM, none of which explicitly incorporate latent variable modeling. More precisely, the authors claim $p_\theta(x)=\mathbb{E_{p(z)}}[p_\theta(x_0|z)]$ can simulate the multivariate distribution, but $p(z)$ appears to be the Dirac-Delta distribution of full-masked sequence for SEDD and MDLM. If the prior $p(z)$ differs from my understanding, I apologize to the authors and would be willing to raise my score.
> >
> > Personally, I think this is a high-quality paper. I evaluate it positively for achieving better performance than existing methods, and I also found the response regarding W1 sufficiently convincing. However, if the justification for Theorem 1 is not clarified, I believe the paper’s theoretical contribution would still need further support in its current form. If the authors can provide a detailed and convincing explanation for the validity of Theorem 1, I would be willing to raise my score.
> >
> > [1] Hayakawa et al., "Distillation of Discrete Diffusion through Dimensional Correlations," ICML'25.
> >
> > [3] Xie et al., "Variational Autoencoding Discrete Diffusion with Enhanced Dimensional Correlations Modeling," ICLR'26.

---

> > > ### Author Response · Authors · 2026-04-08
> > >
> > > Thank you for your positive evaluation of our paper. We clarify how latent variable modeling is used in our approach, and its relation to Theorem 3.1 below.
> > >
> > > **Theoretical perspective.** We would like to note that Theorem 3.1 and the inverse distillation framework itself are formulated for an arbitrary parameterized distribution $p_{\theta}$, including distributions parameterized as mixtures of models. The particular parameterization is therefore a practical design choice within the framework. In the paper, we parameterize $p_{\theta}$ using a generator $G_{\theta}: \mathcal{Z} \rightarrow \mathcal{V}$ or $G_{\theta}: \mathcal{Z} \rightarrow \Delta$ for simplex relaxation (see lines 220-231), where $\mathcal{Z}$ is equipped with a tractable prior $p_{\mathcal{Z}}$.
> > >
> > > **Practical scenario used in our implementation.** In the practical setting of our paper, this mixture parameterization is realized through multistep distillation. In lines 299-320, we write that we use as the latent space $\mathcal{Z}$ the same space as the teacher input. For the MDLM/SEDD case, this gives $\mathcal{Z} = \mathcal{V} \times [0,1]$, and for the Duo case it gives $\mathcal{Z} = \Delta \times [0,1]$.
> > >
> > > The remaining question is what the prior distribution $p_{\mathcal{Z}}$ becomes in this multistep setting. To clarify this, consider the multistep distillation objective. During training, we first sample real data $\widetilde{x}\_0 \sim p^\*(x\_0)$, then a time moment $\widetilde{t} \sim p(\tilde{t}) = U[0, 1]$, and then a corrupted state $x\_{\tilde{t}}\sim p\_{\tilde{t}|0}(\cdot|\widetilde{x}\_0)$. Denote the corresponding distribution of pairs $(x_{\tilde{t}}, \tilde{t})$ by
> > > $$
> > >     p^\*(x\_{\tilde{t}}, \tilde{t}) = p(\tilde{t}) \, \mathbb{E}\_{p^\*(\widetilde{x}\_0)} \big[p(x\_{\tilde{t}}|\widetilde{x}\_0)\big].
> > > $$
> > > Then the full multistep objective is
> > >
> > > $$
> > >     \mathbb{E}\_{p\_{\theta}(x\_0|x\_{\tilde{t}}, \tilde{t}) p^\*(x\_{\tilde{t}}, \tilde{t})}  \Big[\mathcal{L}\_{\text{discr.}}(f^\*, x\_0)\Big]
> > >       - \min\_{\widehat{f}} \mathbb{E}\_{p\_{\theta}(x\_0|x\_{\tilde{t}}, \tilde{t}) p^\*(x\_{\tilde{t}}, \tilde{t})} \Big[\mathcal{L}\_{\text{discr.}}(\widehat{f}, x\_0)\Big]
> > > $$
> > > and this is equivalent to the original inverse objective
> > > $$
> > >   \mathbb{E}\_{p\_{\theta}(x\_0)}  \Big[\mathcal{L}\_{\text{discr.}}(f^\*, x\_0)\Big] - \min\_{\widehat{f}}  \mathbb{E}\_{p\_{\theta}(x\_0)} \Big[\mathcal{L}\_{\text{discr.}}(\widehat{f}, x\_0)\Big],
> > > $$
> > > where $p\_{\theta}(x\_0|x\_{\tilde{t}}, \tilde{t})$ is given by the factorized generator $G\_{\theta}$ and $p\_{\theta}(x_0)$ is given by
> > > $$
> > >     p\_{\theta}(x\_0) = \mathbb{E}\_{p^\*(x_{\tilde{t}}, \tilde{t})} [p_{\theta}(x_0|x_{\tilde{t}}, \tilde{t})],
> > > $$
> > > Thus, in this case the latent distribution is exactly $p^\*(x\_{\tilde{t}}, \tilde{t})$, and the latent variable $z$ is the pair $(x\_{\tilde{t}}, \tilde{t})$, i.e. $p_{\mathcal{Z}} = p^\*(x\_{\tilde{t}}, \tilde{t})$.
> > >
> > > **Explicit noise addition.** Since our framework considers general parametrization of $p\_{\theta}$, we consider adding an additional noise channel to the model to extend further its latent space and prior distribution. Namely, we inject additional Gaussian noise into the student. We sample Gaussian noise, then process it with a small MLP network and add it to the embeddings. Thus, we extend latent space to $\mathcal{Z} = \mathcal{V} \times [0,1] \times \mathbb{R}^d$, where $d$ is the dimensionality of gaussian noise vector. Corresponding prior distribution become $p_{\mathcal{Z}} = p^*(x_{\tilde{t}}, t)q(z)$, where $q(z) = \mathcal{N}(0, I_{d})$ is a standard gaussian distribution for vector of size $d$.
> > >
> > > We run the same IDLM-MDLM distillation experiment with an added noise channel. We present the results below (GenPPL ↓, Entorpy ↑):
> > >
> > > |Steps|IDLM-MDLM|IDLM-MDLM with explicit noise|
> > > |---:|---:|---:|
> > > |1|4056.12 (5.87)|4096.54 (5.88)|
> > > |8|79.75 (6.81)|81.61 (5.61)|
> > > |16|32.75 (5.42)|35.76 (5.45)|
> > > |32|20.45 (5.23)|21.03 (5.26)|
> > >
> > > We observe that this expansion of latent space and prior does not lead to improved performance of the model, and hypothesize that the multi-step distillation latent space is already powerful for the considered steps.
> > >
> > > **Concluding remarks.**
> > > Thus, Theorem 3.1 implies a general parametrization of the student distribution, including a mixture of factorized models. Furthermore, we describe in detail the structure of the latent space and the particular prior distribution used in our models in practice. We show that it is essentially more complex than a single Dirac distribution on fully masked sequences. Moreover, to support general type parametrization used in Theorem 3.1, we present additional results of explicit noise addition to our model. Thank you for your interest in these details. We will add these clarifications to the final version of the paper.
> > >
> > > *We therefore respectfully hope the reviewer will consider raising their score in light of these new results and clarifications.*

---

### Decision · Program_Chairs · 2026-04-30

**Decision:**

Accept (regular)

**Comment:**

The paper proposes IDLM, extending inverse distillation to discrete diffusion language models across both absorbing- and uniform-state settings. Its main strengths are a unified formulation, a uniqueness result for the objective, and gradient-stable relaxations that support effective training. Empirically, the method delivers strong few-step results and substantial inference speedups.

The main weaknesses are clarity and scope. In particular, the discussion around Theorem 3.1 should be stated more carefully, with clearer explanation of what is and is not established for multivariate language modeling. The evaluation is also somewhat limited in scale and breadth, though the rebuttal materially strengthened the paper through added ablations, broader comparisons, larger-scale results, conditional-generation experiments, and coverage-oriented analysis.

Overall, this is a solid and novel contribution on accelerating diffusion language models. The final version should sharpen the exposition and qualify the scope of the theorem more explicitly, but the technical contribution and empirical evidence support **accept**.